# An inherited mitochondrial DNA mutation remodels inflammatory cytokine responses in macrophages and in vivo in mice

Eloïse Marques [1], Stephen P. Burr [1], Alva M. Casey [1], Richard J. Stopforth [2,3], Chak Shun Yu [1], Keira Turner [1], Dane M. Wolf[1], Marisa Dilucca[4], Vincent Paupe[1], Suvagata Roy Chowdhury[1], Victoria J. Tyrrell[5], Robbin Kramer [1], Yamini M. Kanse[1], Chinmayi Pednekar[6], Chris A. Powell[1], James B. Stewart [7], Julien Prudent [1], Michael P. Murphy[1], Michal Minczuk[1,8], Valerie B. O'Donnell [5], Clare E. Bryant [4], Patrick F. Chinnery [1,8], Arthur Kaser [2,3], Alexander von Kriegsheim [6] & Dylan G. Ryan [1,9] ✉

Impaired mitochondrial bioenergetics in macrophages promotes hyperinflammatory cytokine responses, but whether inherited mtDNA mutations drive similar phenotypes is unknown. Here, we profiled macrophages harbouring a heteroplasmic mitochondrial tRNA$^{Ala}$ mutation (*m.5019A>G*) to address this question. These macrophages exhibit combined respiratory chain defects, reduced oxidative phosphorylation, disrupted cristae architecture, and compensatory metabolic adaptations in central carbon metabolism. Upon inflammatory activation, *m.5019A>G* macrophages produce elevated type I interferon (IFN), while exhibiting reduced pro-inflammatory cytokines and oxylipins. Mechanistically, suppression of pro-IL-1β and COX2 requires autocrine IFN-β signalling. IFN-β induction is biphasic: an early TLR4-IRF3 driven phase, and a later response involving mitochondrial nucleic acids and the cGAS-STING pathway. In vivo, lipopolysaccharide (LPS) challenge of *m.5019A>G* mice results in elevated type I IFN signalling and exacerbated sickness behaviour. These findings reveal that a pathogenic mtDNA mutation promotes an imbalanced innate immune response, which has potential implications for the progression of pathology in mtDNA disease patients.

Mitochondria are intracellular organelles that act as a nexus for the integration of anabolic and catabolic pathways essential to eukaryotic life[1–3]. They play a central role in cellular bioenergetics as the main producers of ATP via oxidative phosphorylation (OxPhos), as well as in the supply of intermediates for the synthesis of all major biological macromolecules. Mammalian mitochondria contain their own circular chromosome of approximately 16.5 kb, termed mitochondrial DNA (mtDNA)[4]. Importantly, mtDNA encodes 37 genes, including 13 essential subunits of the mitochondrial respiratory chain, 22 tRNAs,

and 2 rRNAs, while the majority of mitochondrial proteins are encoded in the nucleus.

Multiple copies of mtDNA are found per cell, with copy number varying according to cell fate and energetic demands, and are uniparentally inherited through the maternal germline[4,5]. Inherited and somatic mutations in mtDNA give rise to heteroplasmy, the coexistence of one or more variants of mtDNA within a cell[4,6]. Somatic heteroplasmic single-nucleotide variants arise throughout the human lifespan and accumulate sharply after 70 years of age[6]. Since their

---

discovery approximately 30 years ago, inherited mtDNA mutations have emerged as a key driver of primary mitochondrial disease, a group of rare genetic disorders characterised by impairments in mitochondrial bioenergetics affecting ~1 in 5000 of the human population[5,7]. The largest proportion of heteroplasmic mtDNA mutations occurs in genes encoding mitochondrial tRNAs and disrupts intra-mitochondrial translation[5]. Recurrent microbial infection, sepsis and systemic inflammatory response syndrome (SIRS) are commonly observed in mitochondrial disease patients and a major cause of morbidity and mortality[8–14]. Despite this, we do not understand how pathogenic mtDNA mutations impact innate immunity.

Macrophages are essential cells of the innate immune system[15]. Metabolic rewiring underlies their functional plasticity by supporting pathogen clearance, intra- and inter-cellular communication, and the resolution of inflammation[1,16,17]. Mitochondria are central to this metabolic rewiring, serving as vital signalling hubs for the execution of macrophage effector functions following activation[1,18,19]. Previous reports studying mouse models of mitochondrial dysfunction have shown that macrophages drive hyperinflammatory responses leading to pathology[20–24]. These models have typically relied on profiling Polg[D257A] mutant or Ndufs4[-/-] macrophages, which are nuclear-encoded mitochondrial proteins often presenting with severe phenotypes[20–25]. The Polg[D257A] mutation is found in the catalytic subunit of the mtDNA polymerase and impairs proof-reading[26]. This loss of proofreading activity leads to the damage and depletion of mtDNA. However, this does not replicate inherited pathogenic mtDNA mutations found in patients with mtDNA disease. Therefore, it is unclear how heritable pathogenic mtDNA mutations impact inflammatory macrophage activation and inflammation in vivo. Recently developed models of heteroplasmic mtDNA mutations, including the m.5019A>G mouse[27], now enable this gap to be addressed.

Here, we show that primary macrophages harbouring the m.5019A>G mutation exhibit impaired mitochondrial respiration due to combined disruption of respiratory chain complexes I (CI), III (CIII), and IV (CIV), as well as cristae architecture, with no detectable negative effect on complex V (CV). To compensate for these impairments in mitochondrial bioenergetics, macrophages engage in aerobic glycolysis and reductive glutamine metabolism. However, this disrupts inflammatory macrophage activation, increasing type I interferon (IFN) release and inducible nitric oxide synthase (iNOS)-dependent nitric oxide (NO) production, while limiting pro-inflammatory cytokine and oxylipin production. Lipopolysaccharide (LPS) challenge in vivo leads to elevated IFN-β and IFN-α2 in the serum, along with signalling in the kidney and lungs of m.5019A>G mice, indicating a systemic amplification of the IFN response. Together, our data suggest that heteroplasmic mtDNA mutations perturb innate immune responses ex vivo and in vivo, with implications for mtDNA disease patients.

## Results

### Characterisation of heteroplasmic m.5019A>G macrophages
The m.5019A>G mouse contains a point mutation in the mitochondrial tRNA[Ala] gene (mt-Ta) (Fig. 1a), which occurs in the acceptor stem of mt-Ta and prevents charging with its cognate amino acid, alanine[27]. Heteroplasmy proportion was determined by pyrosequencing DNA extracted from ear skin biopsies obtained at weaning. Only mice with a high proportion of the m.5019A>G mutation (between 70% and 87%) were used in this study. Pyrosequencing was performed on bone marrow cells pre-differentiation and primary bone marrow-derived macrophages (BMDMs) post-differentiation, confirming high mutational burdens that remained stable throughout the differentiation process (Fig. 1b). There was no negative impact on macrophage differentiation, as assessed by cell surface expression of F4/80 and proteomic measurements of F4/80, CD11b and CD11c (Supplementary Fig. 1a, b). However, there was a significant decrease in MHC class II protein levels (Supplementary Fig. 1b; right panel). mtDNA copy

number (CN) was comparable between non-stimulated (non-stim) WT and m.5019A>G macrophages; however, following stimulation with LPS, a component of Gram-negative bacterial membranes, m.5019A>G macrophages exhibited a significant increase in mtDNA CN (Fig. 1c).

To determine whether the m.5019A>G mutation had an impact on mitochondrial translation, we performed [35]S-methionine labelling upon inhibition of cytoplasmic protein synthesis. Incorporation of [35]S-methionine into mitochondrial proteins was significantly reduced in non-stim m.5019A>G macrophages compared to WT (Fig. 1d). No decrease in mitochondrial gene expression was observed (Supplementary Fig. 1c, d), confirming the defect occurs at the level of translation. In agreement with impaired mitochondrial translation, there was a significant reduction in basal respiration, spare respiratory capacity (SRC) and ATP-linked oxygen consumption in non-stim m.5019A>G macrophages (Fig. 1e and Supplementary Fig. 1e). However, there was no difference in proton leak or OxPhos coupling efficiency (Supplementary Fig. 1e, f), which shows mitochondrial ATP synthesis can still occur, albeit to a lesser extent. Consistent with a decrease in OxPhos, Coenzyme Q (CoQ) was significantly more oxidised in non-stim m.5019A>G macrophages compared to WT (Fig. 1f).

To identify mitochondrial alterations resulting from the m.5019A>G mutation in the resting state, we performed unbiased data-independent acquisition (DIA) proteomic profiling in parallel with transcriptomic analysis (Supplementary Fig. 1g, h). Over-representation analysis (ORA) of all differentially abundant proteins using gene ontology (GO) cellular compartment terms revealed a significant enrichment in mitochondrial proteins, including respiratory chain complexes and the mitoribosome (Supplementary Fig. 1g). Further analysis demonstrated a marked reduction in the abundance of the nuclear-encoded structural subunits of CI, CIII, and CIV (Fig. 1g) but not CII or CV (Supplementary Fig. 2a, b). These findings are consistent with previous quantitative proteomics analyses of the cerebral cortex and liver of m.5019A>G mice[27]. This selective impairment in CI, CIII and CIV subunits was confirmed using an OxPhos antibody cocktail in non-stim and LPS-stimulated m.5019A>G macrophages (Fig. 1h). The decreased abundance of mtDNA-encoded MT-CO1 in m.5019A>G macrophages further supports a defect in mitochondrial translation. In contrast, mitoribosome 28S and 39S subunits were increased in m.5019A>G macrophages likely as a compensatory adaptation (Supplementary Fig. 2c). Consistent with our proteomics findings, respirometry analysis in permeabilised cells revealed a significant reduction in CI-dependent, but not CII-dependent, respiration in non-stim m.5019A>G macrophages (Supplementary Fig. 2d). Gene set enrichment analysis (GSEA) following transcriptomic profiling also identified a decrease in OxPhos gene expression (Supplementary Fig. 1h). However, direct comparison of transcript and protein levels for individual subunits of CI (Fig. 1i), as well as CIII and CIV (Supplementary Fig. 2e), revealed only modest reductions at the transcriptional level relative to protein abundance. This discrepancy suggests that the loss of nuclear-encoded OxPhos subunits in m.5019A>G macrophages occurs predominantly through post-transcriptional mechanisms. These findings align with a defect in mitochondrial translation, which disrupts mito-nuclear stoichiometry and impairs the assembly of fully functional respiratory chain complexes[28]. Lastly, this impairment in OxPhos coincided with a modest depolarisation of mitochondrial membrane potential (MMP) in m.5019A>G macrophages, when accounting for an increase in mitochondrial mass (Fig. 1j, k and Supplementary Fig. 2f, g).

Despite preserved mtDNA levels and differentiation capacity, the heteroplasmic m.5019A>G mt-Ta mutation impairs mitochondrial translation, leading to selective depletion of respiratory chain complexes, altered mitochondrial membrane potential and decreased OxPhos. This early mitochondrial remodelling establishes a foundation for altered metabolic and immune responses explored in subsequent sections.

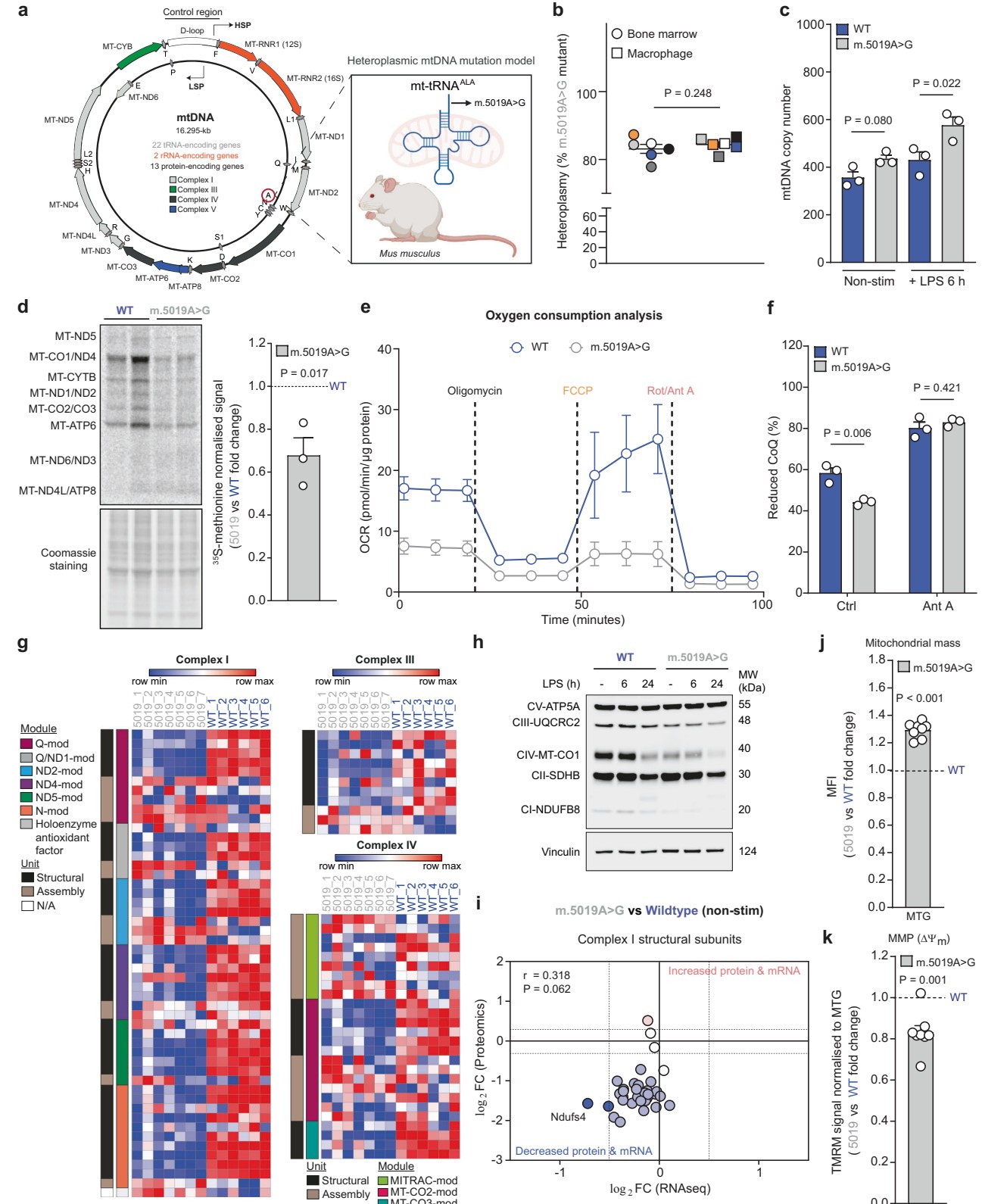

## Glycolytic reprogramming in *m.5019A>G* macrophages

Despite impaired mitochondrial respiration, no changes in whole cell ATP/AMP or ATP/ADP ratios were observed in non-stim *m.5019A>G* macrophages (Supplementary Fig. 3a). This suggested that these macrophages could compensate for the bioenergetic defect. To investigate this further, we analysed our transcriptomic and proteomics datasets and identified a significant enrichment in signatures associated with

HIF-1α signalling, hypoxia, and glycolysis (Fig. 2a, b and Supplementary Fig. 1h). Macrophages engage aerobic glycolysis following inflammatory activation[29–31], which is thought to support cellular ATP synthesis, lactate production by lactate dehydrogenase (LDH), and NAD[+] regeneration in the face of impaired mitochondrial respiration. Measurement of glycolysis using proton efflux rate (PER) confirmed the increase of glycolysis in non-stim *m.5019A>G* macrophages (Fig. 2c, d). Inhibition of

**Fig. 1 | Reduced mitochondrial respiratory chain complexes and oxidative phosphorylation in *m.5019A>G* macrophages. a** Schematic of *m.5019A>G* mt-Ta mutation model. **b** Pyrosequencing results of pre-differentiation bone marrow and post-differentiation bone marrow-derived macrophages (BMDMs) from *m.5019A>G* mice (*n* = 6). Heteroplasmy range between 70% and 87%. Colours indicate matched bone marrow and BMDMs. **c** Mitochondrial DNA (mtDNA) copy number in non-stimulated (non-stim) and lipopolysaccharide (LPS)-stimulated wildtype (WT) and *m.5019A>G* BMDMs (*n* = 3; LPS 6 h). **d** $^{35}$S-methionine labelling and quantification of mitochondrial proteins in non-stim WT and *m.5019A>G* BMDMs (*n* = 3). **e** Seahorse XFe24 oxygen consumption rate (OCR) trace in non-stim WT (*n* = 3) and *m.5019A>G* BMDMs (*n* = 4). **f** Coenzyme Q (CoQ) redox measurements with or without antimycin A (Ant A) in non-stim WT and *m.5019A>G* BMDMs (*n* = 3). **g** Heatmap of all identified complex I (CI), CIII and CIV subunits and assembly factors in non-stim WT (*n* = 6) and *m.5019A>G* (*n* = 7) BMDMs. **h** CV-

ATP5A, CIII-UQCRC2, CIV-MT-COI, CII-SDHB and CI-NDUFB8 protein levels in non-stim and LPS-stimulated WT and *m.5019A>G* BMDMs (*n* = 3; LPS 6 h & 24 h). Representative blot shown. **i** Comparison of $\log_2$FC values of CI structural subunits from proteomics (*n* = 6; WT and *n* = 7; *m.5019A>G*) and RNA sequencing (*n* = 3) data with Pearson *r* correlation and two-tailed statistical analysis applied. **j** Mitochondrial mass (*P* = 0.0000000322) and (**k**) normalised mitochondrial membrane potential (MMP) measurements in non-stim *m.5019A>G* vs WT BMDMs using MitoTracker Green (MTG) and tetramethyl rhodamine methyl ester (TMRM) (*n* = 8). Data are scaled $\log_2$ intensities, $\log_2$FC or mean ± s.e.m. *n* number represents independent biological replicates (mice) from a minimum of two independent experiments. *P*-values calculated using two-tailed Student's *t* test for two group comparisons or multiple two-tailed unpaired *t* tests corrected for multiple comparisons using Benjamini, Krieger and Yekutieli method. **a** Created in BioRender. Dwane, L. (2025) https://BioRender.com/to6x4hj.

CV with oligomycin increased maximum glycolytic capacity in WT macrophages but failed to do so in *m.5019A>G* macrophages, causally linking impaired mitochondrial respiration to increased aerobic glycolysis (Supplementary Fig. 3b). In agreement, oxygen consumption rate (OCR) was also significantly decreased (Supplementary Fig. 3c). Phenotyping of WT and *m.5019A>G* macrophages highlighted a reduction in OCR and concomitant increase in the extracellular acidification rate (ECAR), under non-stim and LPS-stimulated conditions in *m.5019A>G* macrophages (Fig. 2e). Intracellular and extracellular lactate measurements further supported increased glycolysis in *m.5019A>G* macrophages (Supplementary Fig. 3d, e). The ratio of lactate to pyruvate secreted into the cell culture medium (CCM) is a well-established marker of reductive stress and enhanced aerobic glycolysis[32]. Consistent with this, the lactate/pyruvate ratio was significantly elevated in *m.5019A>G* macrophages under both non-stim and LPS-stimulated conditions, with the effect being more pronounced following prolonged LPS exposure (Fig. 2f). This shift was accompanied by an increase in HIF-1α and key glycolytic enzymes, including LDHA and LDHB, further supporting a metabolic bias toward aerobic glycolysis in mutant macrophages (Fig. 2g).

To further dissect the reprogramming of central carbon metabolism, we performed stable isotope-assisted U-$^{13}$C-glucose tracing coupled with liquid chromatography-mass spectrometry (LC-MS) analysis (Fig. 2h). In *m.5019A>G* macrophages, fractional incorporation of glucose-derived carbon into lactate (m + 3) was significantly increased, indicating enhanced glycolytic flux (Fig. 2i and Supplementary Fig. 3f). In contrast, pyruvate (m + 3)-derived labelling of tricarboxylic acid (TCA) cycle intermediates, including citrate (m + 2), isocitrate (m + 2), α-ketoglutarate (α-KG, m + 2), succinate (m + 2), fumarate (m + 2), and malate (m + 2), was reduced (Fig. 2i and Supplementary Fig. 3g), demonstrating impaired entry of glucose-derived carbon into the TCA cycle. Labelling of aspartate (m + 2), which is synthesised in mitochondria from oxaloacetate via the mitochondrial aspartate aminotransferase GOT2, was also decreased, further supporting diminished mitochondrial TCA cycle activity. Together, these adaptations reflect a metabolic state of reductive stress and underscore the ability of *m.5019A>G* macrophages to preserve ATP levels by reprogramming central carbon metabolism under conditions of mitochondrial dysfunction.

## TCA cycle remodelling in *m.5019A>G* macrophages

The TCA cycle undergoes functional remodelling during macrophage activation and is a key regulatory node for the synthesis of immunoregulatory and anti-microbial effectors[29,33–36]. Analysis of TCA cycle metabolite abundance revealed a significant increase in α-KG, fumarate and malate levels in non-stim *m.5019A>G* macrophages relative to WT (Fig. 3a; left panel). α-KG levels were further increased following LPS stimulation, which was accompanied by a significant reduction in succinyl-CoA and aspartate levels. Glutamine anaplerosis is a major source of α-KG in macrophages[37,38]. Under conditions of short-term

glutamine starvation, *m.5019A>G* macrophages exhibited a marked reduction in TCA cycle metabolite abundance both at rest and following activation (Fig. 3a; right panel and Supplementary Fig. 4a), suggesting an increased reliance on glutaminolysis. Glutamine deprivation also reduced aspartate levels in both genotypes (Fig. 3a; right panel), consistent with previous reports demonstrating that glutamine is an important nutrient for aspartate synthesis in macrophages[37].

Succinyl-CoA is synthesised from α-KG in the mitochondrial matrix by the oxoglutarate dehydrogenase complex (OGDHC)[29]. Comparison of the α-KG/succinyl-CoA and α-KG/succinate ratios revealed a significant increase in both non-stim and LPS-stimulated *m.5019A>G* macrophages (Fig. 3b and Supplementary Fig. 4b), indicating reduced flux through OGDHC. OGDHC consists of three core subunits, E1 (OGDH), E2 (DLST), and E3 (DLD)[29]. Proteomic analysis revealed a significant and specific reduction in DLD in non-stim and LPS-stimulated *m.5019A>G* macrophages (Fig. 3c), which was confirmed by western blot (Supplementary Fig. 4c). As DLD is a shared subunit of several mitochondrial dehydrogenase complexes, including the pyruvate dehydrogenase complex (PDHC), it's loss likely contributes to reduced pyruvate entry into the TCA cycle (Fig. 2i and Supplementary Fig. 3g), thereby reinforcing the metabolic shift toward aerobic glycolysis in *m.5019A>G* macrophages.

The TCA cycle can operate in the forward (oxidative) and reverse (reductive) direction[2,39]. Oxidative metabolism supports NADH production for ATP synthesis, while reductive metabolism is associated with NAD(P)H consumption and macromolecule biosynthesis, particularly in respiratory-deficient cancer cells[40,41]. Impairments in CI-dependent respiration and reduced OGDHC levels are known to increase reductive carboxylation in proliferating cells, often triggered by a decline in the NAD$^+$/NADH ratio and resulting reductive stress[40–42]. To investigate whether *m.5019A>G* macrophages engage in reductive metabolism, we performed stable isotope-assisted U-$^{13}$C-glutamine tracing, which enables distinction between oxidative and reductive TCA cycle flux and reflects compartmentalised redox states (Fig. 3d and Supplementary Fig. 4d). The m + 5/m + 3 ratio of α-KG was significantly increased, indicating reduced oxidative metabolism (Supplementary Fig. 4e). Isotopologue analysis revealed increased m + 5 labelling and reduced m + 4 labelling in isocitrate and citrate in non-stim *m.5019A>G* macrophages (Fig. 3e and Supplementary Fig. 4d), which was further enhanced following LPS stimulation. Notably, LPS also increased reductive m + 5 labelling in WT macrophages, consistent with previous findings[43].

Under conditions of OGDHC deficiency and reductive stress, α-KG is converted to 2-hydroxyglutarate (2-HG)[42]. In line with this, 2-HG abundance and m + 5 labelling from glutamine were increased in *m.5019A>G* macrophages (Supplementary Fig. 4f). 2-HG exists as two enantiomers, L-2-HG and D-2-HG, which can only be differentiated by LC-MS after derivitisation[42]. Using this approach, we observed a significant increase in L-2-HG m + 5 labelling in *m.5019A>G* macrophages (Fig. 3f and Supplementary Fig. 4g), further supporting the presence of

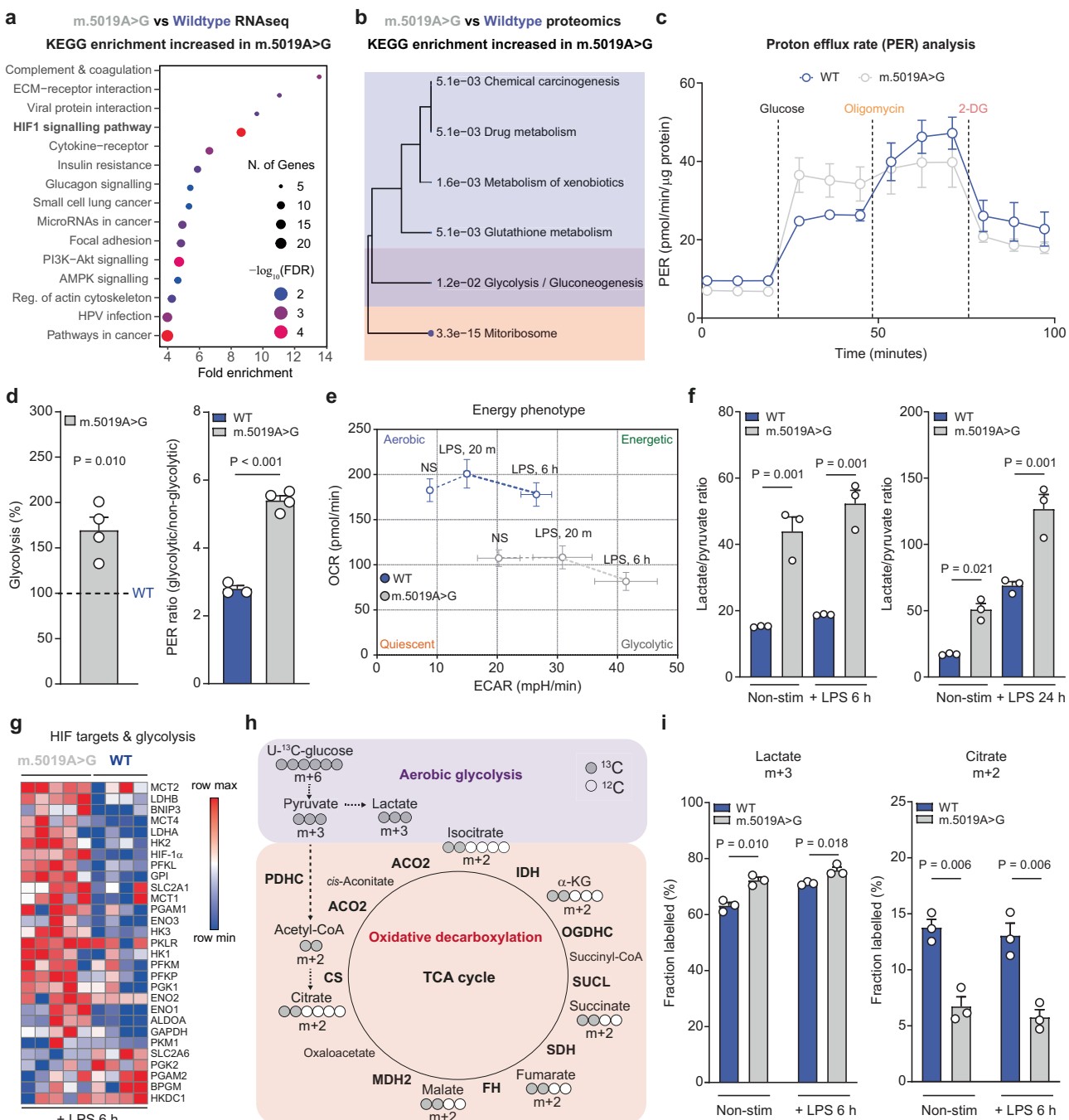

**Fig. 2 | Increased aerobic glycolysis in resting and inflammatory *m.5019A>G* macrophages. a, b** Overrepresentation analysis (ORA) using KEGG terms of all differentially expressed genes from RNA sequencing (*n* = 3) and differentially abundant proteins (*n* = 6; WT and *n* = 7; *m.5019A>G*) increased in non-stimulated (non-stim) *m.5019A>G* vs wildtype (WT) BMDMs. **c, d** Proton efflux rate (PER) measurements in non-stim WT and *m.5019A>G* BMDMs (*n* = 3; WT and *n* = 4; *m.5019A>G*)(*P* = 0.000036). **e** Oxygen consumption rate (OCR) and extracellular acidification rate (ECAR) measurements in non-stim and lipopolysaccharide (LPS)-stimulated WT and *m.5019A>G* BMDMs (*n* = 6; WT and *n* = 7; *m.5019A>G*). **f** Lactate/pyruvate ratio in cell culture medium (CCM) from metabolomics in non-stim and LPS-stimulated WT and *m.5019A>G* BMDMs (*n* = 3; LPS 6 h & 24 h). **g** Heatmap of

hypoxia-inducible factor 1-α (HIF-1α) targets and glycolytic enzymes from proteomics in LPS-stimulated WT and *m.5019A>G* BMDMs (*n* = 4; WT and *n* = 5; *m.5019A>G*; LPS 6 h). **h** Schematic of U-$^{13}$C-glucose tracing into lactate and the tricarboxylic acid (TCA) cycle, indicating the first round labelling pattern. **i** m + 3 labelling in lactate and m + 2 labelling in citrate from U-$^{13}$C-glucose in non-stim and LPS-stimulated WT and *m.5019A>G* BMDMs (*n* = 3; LPS 6 h). Data are scaled log$_2$ intensities or mean ± s.e.m. *n* number represents independent biological replicates (mice) from a minimum of two independent experiments. *P*-values calculated using two-tailed Student's *t* test for two group comparisons or multiple two-tailed unpaired *t* tests corrected for multiple comparisons using Benjamini, Krieger and Yekutieli method.

mitochondrial reductive stress and impaired OGDHC activity. Collectively, these findings confirm that *m.5019A>G* macrophages exhibit increased engagement of reductive glutamine metabolism and synthesis of L-2-HG, indicative of disrupted mitochondrial redox balance and reduced oxidative TCA cycle activity.

In addition, U-$^{13}$C-glucose tracing revealed a significant decrease in m + 3 labelling of citrate, isocitrate, α-KG, succinate, and malate in non-stim *m.5019A>G* macrophages (Fig. 3g, h and Supplementary Fig. 4h). This labelling pattern originates from pyruvate (m + 3) entry into the TCA cycle via pyruvate carboxylase (PC) and is consistent with

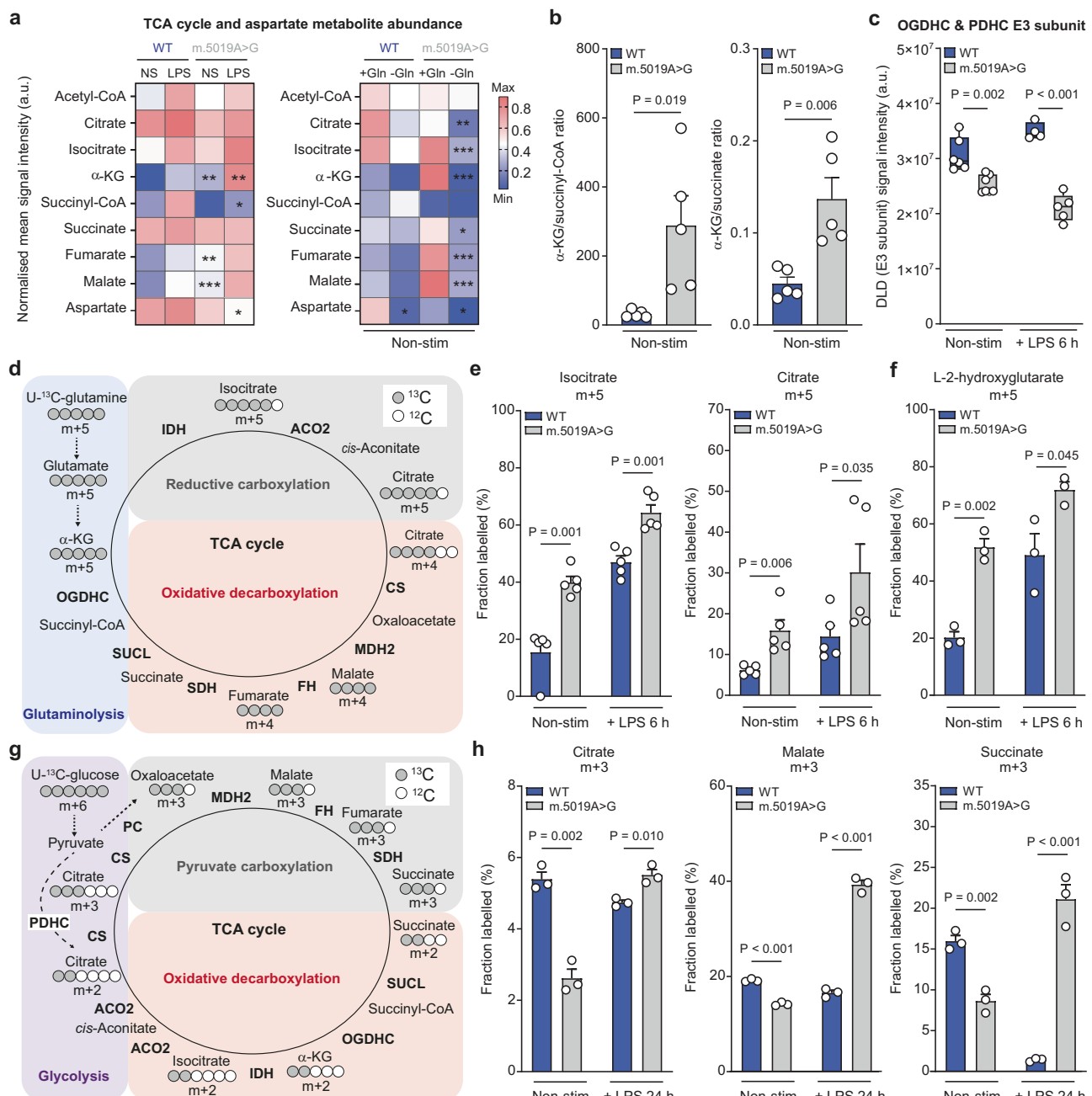

**Fig. 3 | Tricarboxylic acid (TCA) cycle remodelling in *m.5019 A>G* macrophages.**
**a** Heatmap comparing metabolite levels in non-stimulated (non-stim) (α-KG, *P* = 0.001812; Fumarate, *P* = 0.006172; Malate, *P* = 0.000451) and lipopolysaccharide (LPS)-stimulated (α-KG, *P* = 0.005704; Succinyl-CoA, *P* = 0.025485; Aspartate, *P* = 0.047568) wildtype (WT) and *m.5019A>G* BMDMs (*n* = 5; LPS 6 h; left), and comparing metabolite levels in non-stim WT and *m.5019A>G* BMDMs in the presence or absence of glutamine (Gln) (*n* = 3; LPS 4 h; right) (Citrate, *P* = 0.008855; Isocitrate, *P* = 0.000054; α-KG, *P* = 0.0000001; Succinate, *P* = 0.0128505; Fumarate, *P* = 0.0000541; Malate, *P* = 0.0000116; Aspartate, *P* = 0.0297496). **b** α-Ketoglutarate (α-KG)/succinyl-CoA and α-KG/succinate ratio in non-stim WT and *m.5019A>G* BMDMs (*n* = 5). **c** Oxoglutarate dehydrogenase complex (OGDHC) and pyruvate dehydrogenase complex (PDHC) E3 subunit (DLD) levels from proteomics in non-stim WT (*n* = 4), non-stim *m.5019A>G* (*n* = 5), LPS-stimulated WT (*n* = 6; 6 h) and LPS-stimulated *m.5019A>G* BMDMs (*n* = 7; 6 h) (*P* = 0.000047). **d** Schematic of

U-¹³C-glutamine tracing into the tricarboxylic acid (TCA) cycle, indicating oxidative versus reductive labelling patterns. **e** m + 5 labelling from U-¹³C-glutamine in isocitrate and citrate in non-stim and LPS-stimulated WT and *m.5019A>G* BMDMs (*n* = 5; LPS 6 h). **f** m + 5 labelling in L-2-hydroxyglutarate (L-2-HG) from U-¹³C-glutamine in non-stim and LPS-stimulated WT and *m.5019A>G* BMDMs (*n* = 3; LPS 6 h). **g** Schematic of U-¹³C-glucose tracing into the TCA cycle, indicating oxidative versus reductive labelling patterns. **h** m + 3 labelling from U-¹³C-glucose in citrate, malate (*P* = 0.000405; *P* = 0.000405) and succinate in non-stim and LPS-stimulated WT and *m.5019A>G* BMDMs (*n* = 5; LPS 24 h). Data are mean or mean ± s.e.m. *n* number represents independent biological replicates (mice) from a minimum of three independent experiments. *P*-values calculated using two-tailed Student's *t* test for two group comparisons or multiple two-tailed unpaired *t* tests corrected for multiple comparisons using Benjamini, Krieger and Yekutieli method. *** *P* < 0.001 ** *P* < 0.01 * *P* < 0.05.

the switch to aerobic glycolysis in the resting state. However, following prolonged LPS stimulation, *m.5019A>G* macrophages display a marked increase in pyruvate-driven anaplerosis. This is particularly evident for succinate, where there is virtually no glucose-derived labelling of

isocitrate, α-KG and succinate in WT macrophages due to inhibition of aconitase 2 (ACO2)[29], while m + 3-labelled succinate is strongly increased in *m.5019A>G* cells (Fig. 3g, h and Supplementary Fig. 4h). This suggests that the m + 3 labelling in succinate in *m.5019A>G*

macrophages arises from reversal of CII, a metabolic adaptation absent in WT macrophages. Together, these findings reveal that *m.5019A>G* macrophages undergo extensive TCA cycle remodelling and increased engagement of reductive pathways when activated. These adaptations reflect a metabolically reprogrammed state driven by mitochondrial dysfunction and underscore the plasticity of the cellular metabolism in macrophages.

### Inflammatory aspartate-argininosuccinate shunt and NO production in *m.5019A>G* macrophages

Upon LPS stimulation, an inflammatory aspartate-argininosuccinate shunt (AAS) is induced in macrophages, driven by the increased expression of the urea cycle enzyme argininosuccinate synthetase 1 (ASS1) and the synthesis of argininosuccinate from aspartate[34,37,44]. This shunt is required to support the synthesis of arginine in the cytosol by argininosuccinate lyase (ASL), which is subsequently used by iNOS for NO production[34,37]. Both U-$^{13}$C-glutamine and U-$^{13}$C-glucose tracing revealed significantly increased m + 3 labelled argininosuccinate and fumarate in LPS-stimulated *m.5019A>G* macrophages, consistent with their origin from reductively synthesised aspartate (m + 3) (Fig. 4a–d and Supplementary Fig. 5a, b). These findings demonstrate that reductive glutamine and pyruvate carboxylation, metabolic hallmarks of hypoxic and respiratory-deficient cancer cells, are similarly engaged in respiratory-deficient macrophages, where they support the inflammatory AAS. Furthermore, LPS-induced iNOS expression and glutamine-dependent NO production were significantly elevated in *m.5019A>G* macrophages, as measured by western blot, qPCR, and nitrite concentration in the CCM (Fig. 4e–g and Supplementary Fig. 5c). These data indicate that, despite pronounced mitochondrial dysfunction, *m.5019A>G* macrophages retain their capacity for NO synthesis through substantial metabolic rewiring. Given that iNOS is a well-characterised interferon-stimulated gene (ISG)[45–50], with key immunoregulatory and anti-microbial roles, this prompted us to perform detailed immune phenotyping of *m.5019A>G* macrophages.

### Disrupted inflammatory setpoints in *m.5019A>G* macrophages

To determine the impact of the *m.5019A>G* mutation on macrophage function, we performed RNA-seq to identify differentially expressed inflammatory genes (Fig. 5a and Supplementary Fig. 6a). A robust increase in *Ifnb1* expression and IFN-β secretion was observed in LPS-stimulated *m.5019A>G* macrophages (Fig. 5a and Supplementary Fig. 6b), consistent with prior studies reporting enhanced type I IFN responses in macrophages with impaired mitochondrial respiration, including the *Polg*[D257A] mutator model of mitochondrial disease[20,37,51]. To investigate this further, we performed a time-course analysis following LPS stimulation. In agreement with the RNA-seq data, *Ifnb1* expression was significantly increased as early as 1 hour and peaked at 2 h post-LPS in *m.5019A>G* macrophages (Fig. 5b). By 6 hours, expression declined in both genotypes; however, in *m.5019A>G* macrophages, *Ifnb1* levels increased again at 24 h albeit to a lesser extent than the initial peak, an effect not observed in WT cells. Consistent with this, IFN-β release peaks more rapidly and reaches higher concentrations in *m.5019A>G* macrophages and remains significantly higher at 24 h following LPS stimulation (Fig. 5c and Supplementary Fig. 6b). The early increase in IFN-β was accompanied by enhanced phosphorylation of IRF3 at serine 396, an essential activation signal, observed at 1 h post-stimulation (Fig. 5d), indicating amplified TLR4-mediated IRF3 signalling in mutant macrophages. At the later time-points, gene expression and protein levels of several downstream ISGs were significantly higher, including *Nos2*/iNOS (Fig. 4f, g), *Isg15*/ISG15, *Irf7*/IRF7, *Isg20*, and *Cxcl10* (Fig. 5d and Supplementary Fig. 6c, d). These results suggest a biphasic pattern of *Ifnb1* transcriptional regulation in *m.5019A>G* macrophages, likely reflecting distinct temporal phases of upstream signalling that collectively sustain an enhanced type I IFN response.

We subsequently performed oxylipin profiling of the CCM with LC-MS/MS to identify differentially secreted inflammatory lipid mediators (Fig. 5e and Supplementary Fig. 6e) and Olink proteomic profiling to assess cytokine and chemokine release (Fig. 5f) in LPS-stimulated WT and *m.5019A>G* macrophages. Oxylipin profiling quantified six oxylipins in total, with a significant reduction in the cyclooxygenase (COX) products prostaglandin D2 (PGD$_2$), PGE$_2$, and 11-HETE (Fig. 5e), but not COX-independent oxylipins (Supplementary Fig. 6e), in *m.5019A>G* macrophages. Interestingly, Olink profiling identified a significant decrease in IL-1β and IL-6 levels (Fig. 5e and Supplementary Fig. 6f). IFN-β was not included as part of the profiling; however, there was an increase in IFN-α2, which is less abundant in macrophages. Decreases in IL-1β and COX2 were validated using a combination of qPCR and western blot (Fig. 5g and Supplementary Fig. 6g). IL-1β release was also significantly decreased following infection with the gram-negative bacterium *Salmonella typhimurium* (STM) or ATP stimulation in LPS-primed *m.5019A>G* macrophages, while no difference in cell death or bacterial burden was observed (Supplementary Fig. 6h, i). This indicates that restricted IL-1β secretion arises from reduced *Il1b* expression and pro-IL-1β levels, rather than impairment in the NLRC4 or NLRP3 inflammasomes per se. Finally, IL-6 expression and release was modestly impaired, while no significant differences in TNF-α expression or release was observed (Supplementary Fig. 7a–d).

Type I IFN signalling has been reported to antagonise IL-1β production[37,52,53]. Consistent with this, co-treatment of WT macrophages with LPS and recombinant mouse IFN-β (r-mIFN-β) significantly decreased pro-IL-1β and COX2 levels, while increasing *Nos2*/iNOS expression, thereby phenocopying the response observed in LPS-stimulated *m.5019A>G* macrophages (Supplementary Fig. 7e, f). To directly assess the role of autocrine IFN signalling, we pre-treated WT and *m.5019A>G* macrophages with either the JAK inhibitor Ruxolitinib or a monoclonal IFNAR-blocking antibody for 1 hour prior to LPS stimulation (6 or 24 hours) (Fig. 5h–j and Supplementary Fig. 7g). Both treatments reversed the suppression of pro-IL-1β and COX2, and abolished iNOS induction, thereby mechanistically linking autocrine type I IFN-IFNAR-JAK-STAT signalling to the regulation of these inflammatory mediators. In summary, *m.5019A>G* macrophages exhibit an altered inflammatory profile marked by enhanced type I IFN signalling and selective suppression of IL-1β and COX2, with a mild reduction in IL-6 release. This phenotype is partially driven by autocrine IFNAR-JAK-STAT signalling and reveals that mitochondrial dysfunction arising from an inherited mtDNA mutation reshapes inflammatory setpoints through sustained type I IFN activity.

### Mitochondrial network remodelling in *m.5019A>G* macrophages

To understand this imbalance in the innate immune response, we next sought to determine the downstream consequences of the *m.5019A>G* mutation on mitochondrial network morphology. Severe mitochondrial defects often give rise to mitochondrial fission and swelling[54,55]. Given the substantial impairment in mitochondrial respiration in *m.5019A>G* macrophages, we assessed whether this affected mitochondrial morphology. Contrary to expectations, staining of the inner mitochondrial membrane (IMM) protein, Cytochrome c (Cyt c), and the outer mitochondrial membrane protein (OMM), TOM20, revealed an increase in mitochondrial length, junction points and junction points per mitochondrial network in non-stim *m.5019A>G* macrophages, which remained elongated following LPS stimulation for 6 h (Fig. 6a, b and Supplementary Fig. 8a–c). This observation is similar to previous reports of stress-induced mitochondrial hyperfusion (SIMH), a pro-survival adaptation to mild metabolic stress[55,56]. Consistent with a previous study[57], LPS stimulation also promoted mitochondrial elongation in WT macrophages to a similar extent as that observed in non-stim *m.5019A>G* macrophages.

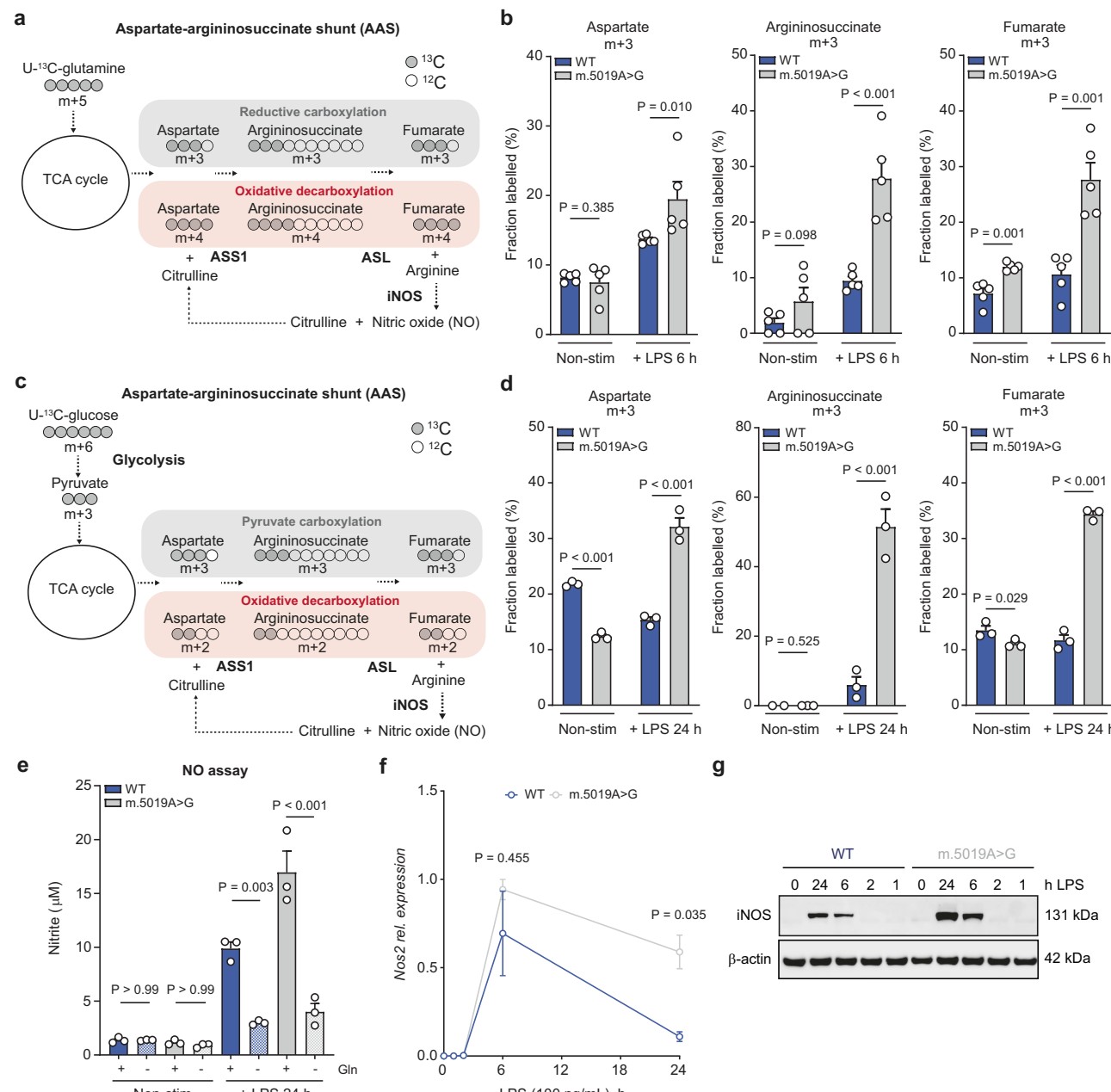

**Fig. 4 | Reductive glutamine and pyruvate carboxylation fuel the AAS and NO in *m.5019 A>G* macrophages. a** Schematic of U-$^{13}$C-glutamine tracing into the aspartate-argininosuccinate shunt (AAS), indicating oxidative versus reductive labelling patterns and NO production. **b** m + 3 labelling from U-$^{13}$C-glutamine in aspartate, argininosuccinate ($P = 0.000026$) and fumarate ($P = 0.000005$) in non-stimulated (non-stim) and lipopolysaccharide (LPS)-stimulated wildtype (WT) and *m.5019A>G* BMDMs ($n = 5$; LPS 6 h). **c** Schematic of U-$^{13}$C-glucose tracing into the AAS indicating oxidative versus reductive labelling patterns and nitric oxide (NO) production. **d** m + 3 labelling from U-$^{13}$C-glucose in aspartate ($P = 0.000070$; $P = 0.000002$), argininosuccinate ($P = 0.000015$) and fumarate ($P = 0.000039$) in non-stim and LPS-stimulated WT and *m.5019A>G* BMDMs ($n = 3$; LPS 24 h). **e** Nitrite

levels in cell culture medium (CCM) in non-stim and LPS-stimulated WT and *m.5019A>G* BMDMs in the presence or absence of glutamine (Gln) ($n = 3$; LPS 24 h) ($P = 0.00000007$). **f, g** *Nos2* expression and inducible nitric oxide synthase (iNOS) protein levels from LPS time course analysis in WT and *m.5019A>G* BMDMs ($n = 3$; LPS 0, 1, 2, 6 & 24 h). Representative blot shown. Data are mean ± s.e.m. *n* number represents independent biological replicates (mice) from a minimum of three independent experiments. *P*-values calculated using multiple two-tailed unpaired *t* tests corrected for multiple comparisons using the Benjamini, Krieger and Yekutieli method or one-way ANOVA corrected for multiple comparisons using the Tukey method.

To examine this phenotype closer, we performed super-resolution microscopy to assess mitochondrial morphology (Fig. 6c, d and Supplementary Fig. 8d, e). Immunofluorescent staining of TOM20 confirmed the mitochondrial elongation phenotype, as determined by a significant decrease in mitochondrial oblate ellipticity and sphericity, in both non-stim and LPS-stimulated *m.5019A>G* macrophages. Interestingly, a significant increase in ATP synthase (CV) puncta (Supplementary Fig. 8e) was observed in

*m.5019A>G* macrophages, likely as a compensatory mechanism for reduced mitochondrial respiration. In addition, we performed co-staining of TOM20 and the main driver of mitochondrial fission, Dynamin-related protein 1 (DRP1). DRP1 is recruited to the OMM to trigger mitochondrial fission, and increased co-localisation of DRP1 and the OMM is an indication of a more punctate/fragmented mitochondrial network[55]. Consistent with the elongation phenotype in *m.5019A>G* macrophages, there was less co-localisation between

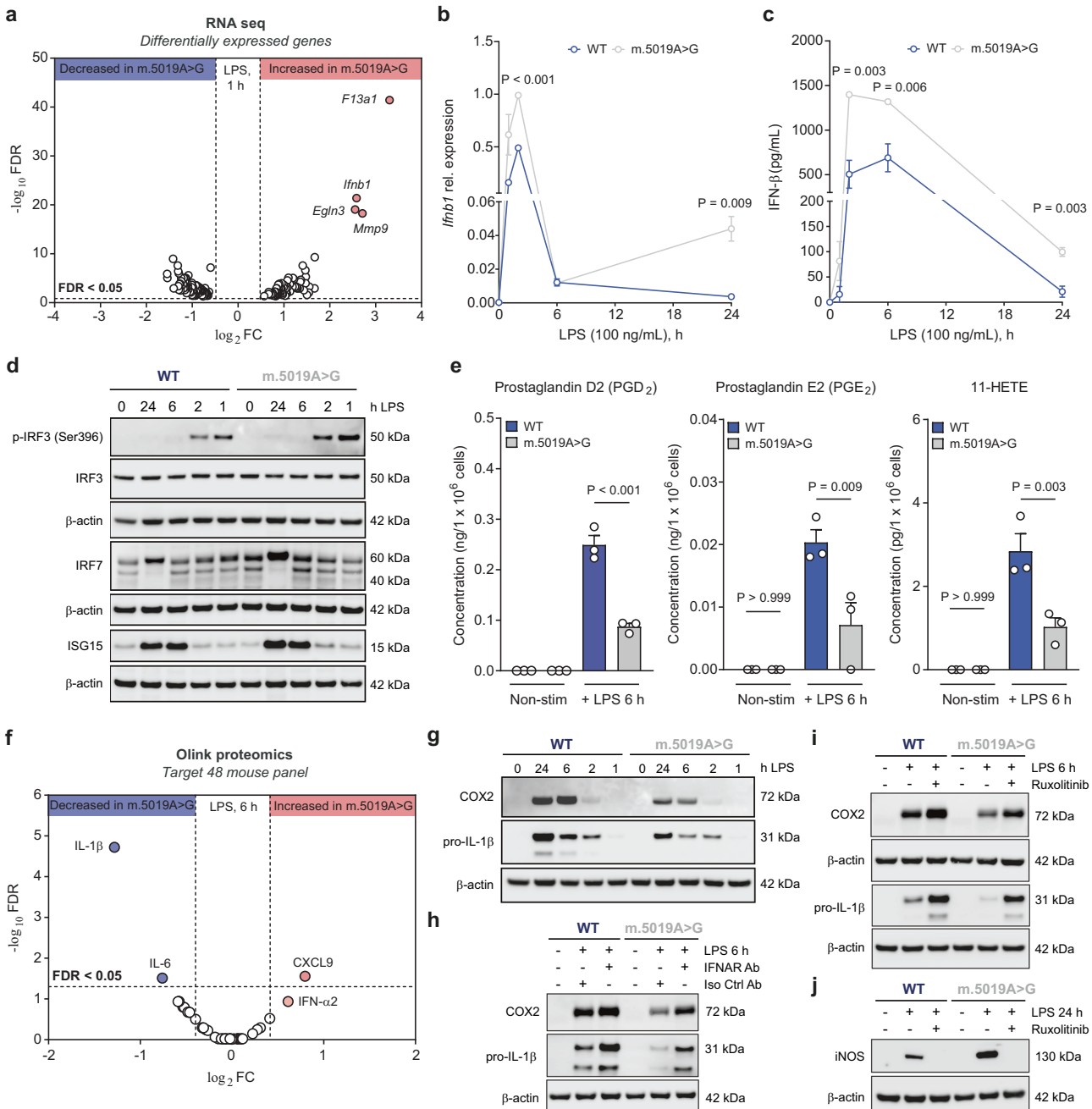

**Fig. 5 | Inflammatory cytokine and oxylipin production is disrupted in *m.5019A>G* macrophages. a** Volcano plot of differentially expressed genes from RNA sequencing in lipopolysaccharide (LPS)-stimulated *m.5019A>G* vs wildtype (WT) BMDMs (*n* = 3; LPS 1 h). **b, c** *Ifnb1* expression and interferon-β (IFN-β) release from LPS time course analysis in WT and *m.5019A>G* BMDMs (*n* = 3; LPS 0, 1, 2, 6 & 24 h) (*P* = 0.000052). **d** Phospho-interferon regulatory factor 3 (IRF3) (ser396), IRF3, IRF7, and interferon-stimulated gene 15 (ISG15) protein levels from LPS time course analysis in WT and *m.5019A>G* BMDMs (*n* = 3; LPS 0, 1, 2, 6 & 24 h). Representative blot shown. **e** Oxylipin profiling of cell culture medium (CCM) in non-stimulated (non-stim) and LPS-stimulated WT and *m.5019A>G* BMDMs (*n* = 3; LPS 6 h) (*P* = 0.00001417). **f** Olink target T48 mouse cytokine and chemokine profiling of CCM in LPS-stimulated *m.5019A > G* vs WT BMDMs (*n* = 3; LPS 6 h). **g** Cyclooxygenase 2 (COX2) and pro-interleukin-1β (pro-IL-1β) protein levels from LPS time course analysis in WT and *m.5019A>G* BMDMs (*n* = 3; LPS 0, 1, 2, 6 & 24 h).

Representative blot shown. **h** COX2 and pro-IL-1β protein levels in non-stim and LPS-stimulated WT and *m.5019A>G* BMDMs treated with an anti-interferon-α/β receptor (IFNAR) monoclonal antibody (Ab) or isotype control Ab (*n* = 3; LPS 6 h). Representative blot shown. **i** COX2 and pro-IL-1β protein levels in non-stim and LPS-stimulated WT and *m.5019A>G* BMDMs treated with Ruxolitinib or vehicle control Ab (*n* = 3; LPS 6 h). Representative blot shown. **j** Inducible nitric oxide synthase (iNOS) protein levels in non-stim and LPS-stimulated WT and *m.5019A>G* BMDMs treated with Ruxolitinib or vehicle control Ab (*n* = 3; LPS 24 h). Representative blot shown. Data are log2FC or mean ± s.e.m. *n* number represents independent biological replicates (mice) from a minimum of three independent experiments. *P*-values calculated using two-tailed Student's *t* test for two group comparisons, multiple two-tailed unpaired *t* tests corrected for multiple comparisons using Benjamini, Krieger and Yekutieli method or one-way ANOVA corrected for multiple comparisons using Tukey method.

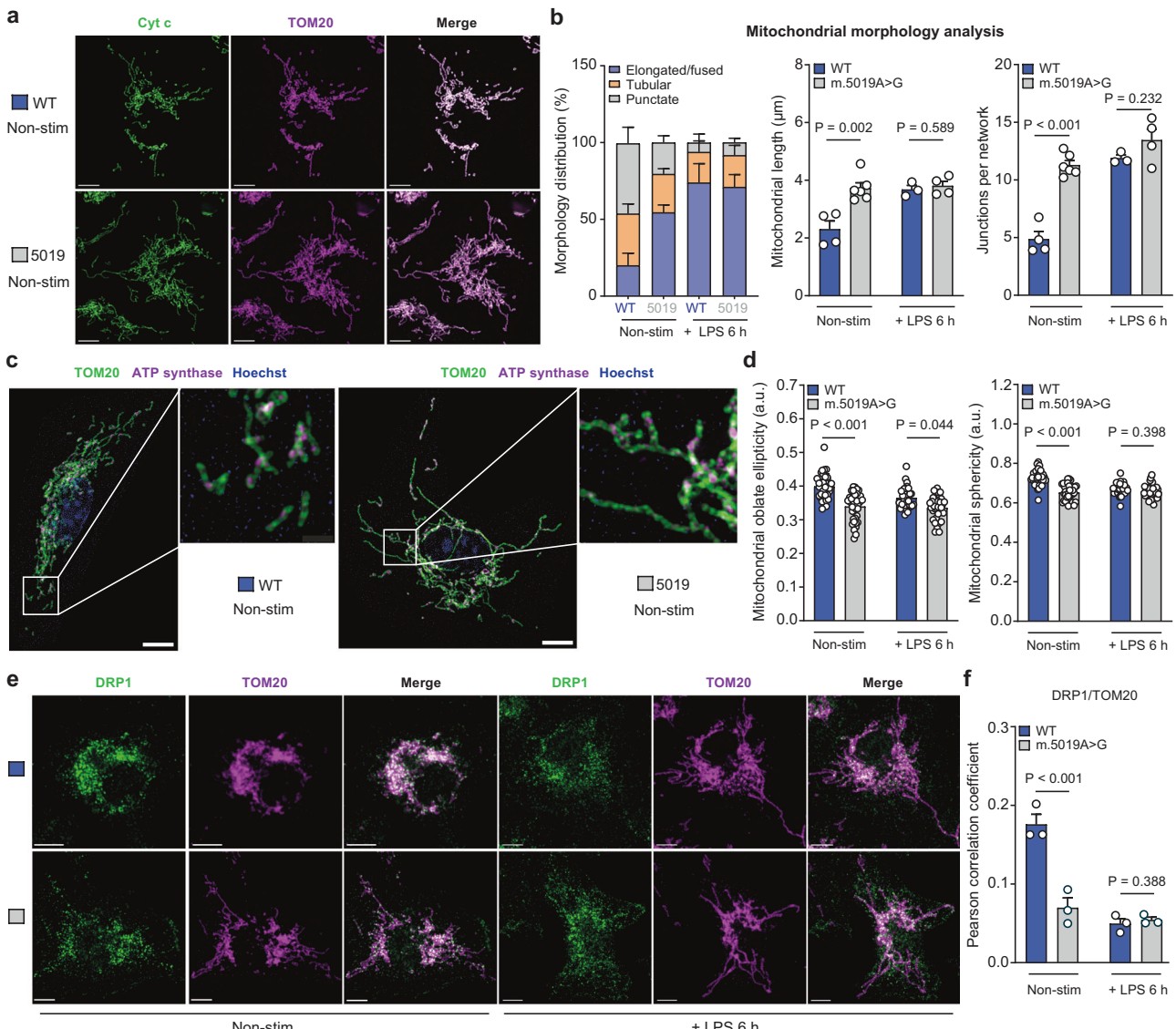

**Fig. 6 | Mitochondrial network remodelling in *m.5019A>G* macrophages.**
**a**, **b** Representative immunofluorescence staining of cytochrome c (Cyt c) and translocase of the outer membrane 20 (TOM20) coupled to confocal microscopy in non-stimulated (non-stim) wildtype (WT) and *m.5019A>G* BMDMs (**a**) and mitochondrial morphology analysis in non-stim WT (*n* = 4), non-stim *m.5019A>G* (*n* = 6), lipopolysaccharide (LPS)-stimulated WT (*n* = 3) and LPS-stimulated *m.5019A>G* (*n* = 4) BMDMs (**b**) (LPS 6 h; minimum of 20 cells analysed per condition per biological replicate) (*P* = 0.000038). Scale bars: 5 μm. **c**, **d** Representative immunofluorescence staining of TOM20 and ATP synthase coupled to super-resolution microscopy in non-stim WT and *m.5019A>G* BMDMs (**c**) and mitochondrial morphology analysis of non-stim WT (*n* = 3), non-stim *m.5019A>G* (*n* = 3), LPS-stimulated WT (*n* = 2) and LPS-stimulated *m.5019A>G* (*n* = 3) BMDMs (**d**) (LPS 6 h;

minimum of 33 cells analysed from independent biological replicates) (*P* = 0.000000183; *P* = 0.0000000003). Scale bars: 5 μm. **e**, **f** Representative immunofluorescence staining of dynamin-related protein 1 (DRP1) and TOM20 coupled to confocal microscopy (**e**) and Pearson *r* correlation analysis (**f**) in non-stim and LPS-stimulated WT and *m.5019A>G* BMDMs (*n* = 3; LPS 6 h; minimum of 20 cells analysed per condition per biological replicate) (*P* = 0.000059). Scale bars: 5 μm. Data are mean ± s.e.m or ± s.d. *n* number represents independent biological replicates (mice) from a minimum of two independent experiments. *P*-values calculated using multiple two-tailed unpaired tests corrected for multiple comparisons using the Holm-Sidak method or one-way ANOVA corrected for multiple comparisons using the Kruskal-Wallis method.

---

DRP1 and the mitochondrial network, which is at similar levels to LPS-stimulated WT macrophages, also exhibiting mitochondrial elongation (Fig. 6e, f).

### Disrupted mitochondrial cristae and nucleic acid release in *m.5019A>G* macrophages

To assess mitochondrial architecture in greater detail, we employed transmission electron microscopy (TEM) (Fig. 7a, b and Supplementary Fig. 9a). Non-stim WT macrophages presented with more punctate electron-dense mitochondria, which elongated following LPS stimulation. In contrast, *m.5019A>G* mitochondria had a greater proportion

of low-density elongated mitochondria with disrupted cristae architecture, consistent with a defect in mitochondrial respiration. This data confirms that the *m.5019A>G* mutation in macrophages leads to a remodelling of the mitochondrial network. Disruption of mitochondrial cristae architecture and respiration has previously been shown to induce type I IFN signalling via the release of mitochondrial nucleic acids[37,54,58], and is consistent with the increase in IFN-β release observed in *m.5019A>G* macrophages. To investigate this, we assessed whether immunostimulatory mtDNA or mtRNA was present in the cytosol of WT or *m.5019A>G* macrophages using a digitonin-based fractionation method coupled to qPCR (Fig. 7c and Supplementary

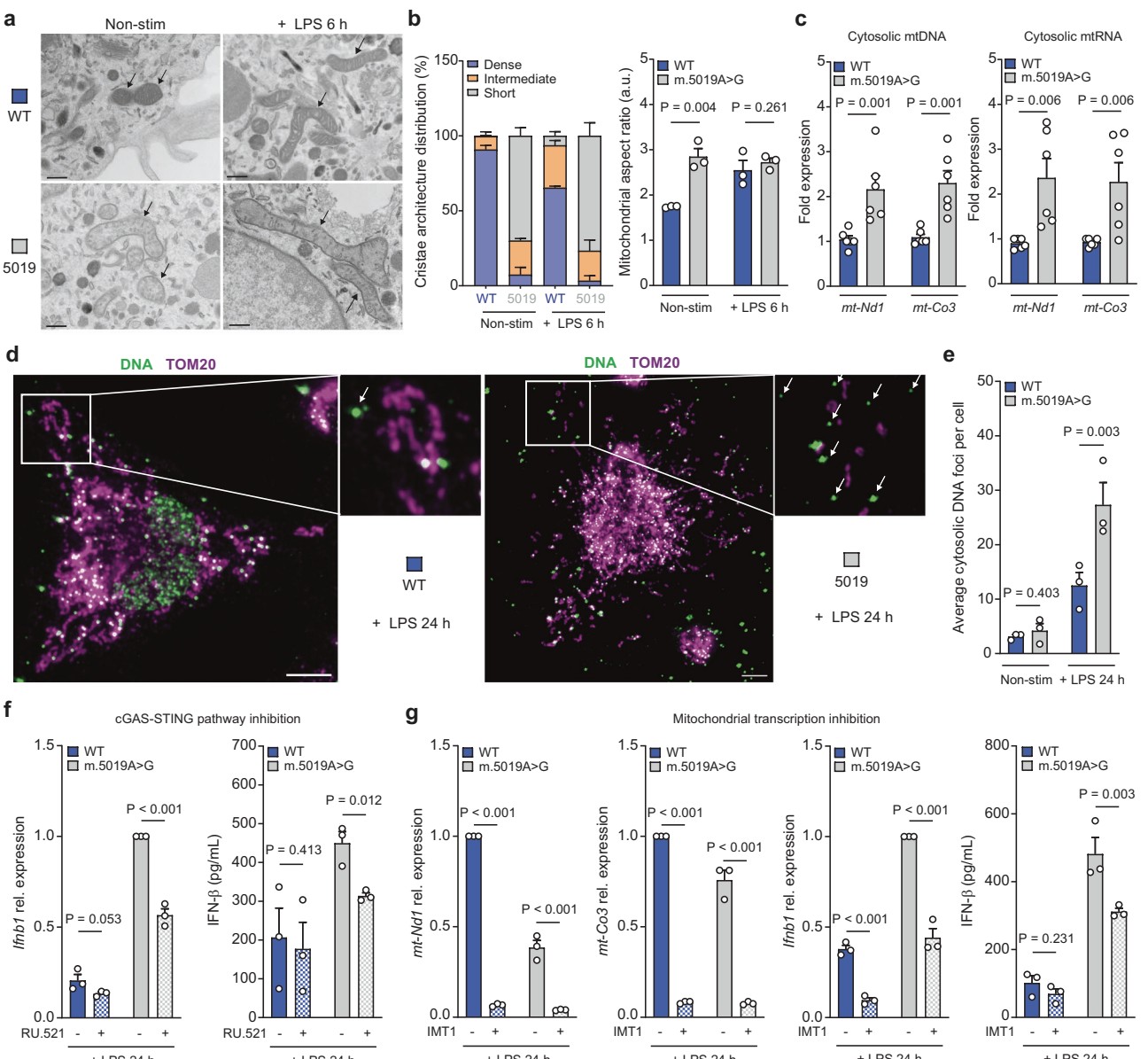

**Fig. 7 | Late phase type I IFN in *m.5019A>G* macrophages is dependent on mitochondrial nucleic acids. a, b** Representative transmission electron microscopy (TEM) images (**a**) and cristae and mitochondrial aspect ratio (length/width) analysis (**b**) of non-stimulated (non-stim) and lipopolysaccharide (LPS)-stimulated wildtype (WT) and *m.5019A>G* BMDMs (*n* = 3; LPS 6 h; mitochondria from a minimum of 9 cells were analysed per condition per biological replicate). Scale bars: 0.5 µm. Black arrows indicate mitochondria. **c** Mitochondrial (mt)DNA and mtRNA levels in cytosolic fraction of non-stim WT and *m.5019A>G* BMDMs (*n* = 6). **d, e** Representative immunofluorescence staining of DNA and translocase of the outer membrane 20 (TOM20) coupled to confocal microscopy in LPS-stimulated WT and *m.5019A>G* BMDMs (**d**) and cytosolic DNA foci quantification in non-stim and LPS-stimulated WT and *m.5019 A > G* BMDMs (**e**) (*n* = 3; LPS 24 h; minimum of 20 cells analysed per condition per biological replicate). White arrows indicate

cytosolic DNA foci. **f** *Ifnb1* expression (LPS 24 h) and interferon-β (IFN-β) release (LPS 6 h) in LPS-stimulated WT and *m.5019 A > G* macrophages pre-treated with cyclic GMP-AMP synthase (cGAS) inhibitor RU.521 or vehicle control (DMSO) for 1 h (*n* = 3) (*P* = 0.000262). **g** *mt-Nd1* (*P* = 0.000000008; *P* = 0.000946773), *mt-Co3* (*P* = 0.000000005; *P* = 0.000236469), *Ifnb1* (*P* = 0.000402; *P* = 0.000402) expression and IFN-β release in LPS-stimulated WT and *m.5019A>G* macrophages pre-treated with inhibitor of mitochondrial transcription 1 (IMT1) or vehicle control (DMSO) for 24 h (*n* = 3; LPS 24 h). Data are mean ± s.e.m. *n* number represents independent biological replicates (mice) from a minimum of three independent experiments. *P*-values calculated using multiple two-tailed unpaired *t* tests corrected for multiple comparisons using the Benjamini, Krieger and Yekutieli method.

Fig. 9b). Levels of both mtDNA and mtRNA were significantly higher in the cytosolic fraction of *m.5019A>G* macrophages. However, this approach has limitations as it is relatively crude and may overestimate cytosolic content, particularly in the context of altered cristae architecture that may render *m.5019A>G* mitochondria more susceptible to permeabilisation.

To more rigorously assess cytosolic nucleic acids, we performed immunofluorescent staining of DNA in intact cells in combination with

TOM20 and analysed the samples by confocal microscopy (Fig. 7d, e and Supplementary Fig. 9c, d). In the resting state, we observed no significant difference in cytosolic DNA foci between *m.5019A>G* and WT macrophages. However, following 24 hours of LPS stimulation, *m.5019A>G* macrophages exhibited a more substantial increase in cytosolic DNA, which has previously been shown to activate the DNA sensing, cGAS-STING pathway[20]. Interestingly, while *m.5019A>G* macrophages typically display a fused mitochondrial network at rest and

after short-term LPS stimulation, we observed notable fragmentation coinciding with increased cytosolic DNA after prolonged LPS treatment (Fig. 7d and Supplementary Fig. 9c). To evaluate whether these cytosolic DNA foci contribute to IFN-β production, we subsequently inhibited the cGAS-STING pathway using RU.521, a well-characterised small molecule inhibitor of cGAS[59]. Inhibition of cGAS selectively reduced late-phase *Ifnb1* expression and IFN-β secretion in *m.5019A>G* macrophages but had no effect on early-phase IFN-β production (Fig. 7f and Supplementary Fig. 9e). However, the inhibition of *Ifnb1* expression by RU.521 was only partial, and we didn't observe a significant decrease in downstream *Isg15* or *Isg20* expression (Supplementary Fig. 9e), suggesting additional factors may also contribute to the late-phase IFN-β response.

To investigate a potential role for mtRNA sensing, we inhibited mtDNA transcription using a recently developed small molecule, IMT1 (Inhibitor of Mitochondrial Transcription)[60]. Pre-treatment of both WT and *m.5019A>G* macrophages with IMT1 for 24 h, prior to LPS stimulation for either 6 or 24 h, led to a substantial reduction in total mtRNA levels (Fig. 7g and Supplementary Fig. 9f). Similar to cGAS inhibition, we observed no significant change in early phase IFN-β release in WT or *m.5019A>G* macrophages (Supplementary Fig. 9f). In contrast, mtRNA depletion resulted in a significant reduction in late phase *Ifnb1* transcript levels, IFN-β secretion, and reduced both *Isg15* and *Isg20* expression in *m.5019A>G* macrophages (Fig. 7g and Supplementary Fig. 9g). mtRNA has previously been shown to activate the double-stranded (dsRNA) sensors, RIG-I and MDA5, in LPS-stimulated macrophages[33,37]. In agreement, late phase *Ifnb1* expression was significantly decreased in WT cells, but this failed to appreciably lower secreted IFN-β levels when compared to *m.5019A>G* macrophages. However, we cannot yet distinguish whether the observed effect in *m.5019A>G* macrophages is due to direct mtRNA sensing or an undefined secondary consequence of impaired mitochondrial gene expression. Taken together, these findings support a biphasic model of *Ifnb1* regulation in *m.5019A>G* macrophages: an early phase driven by enhanced TLR4-IRF3 signalling, and a later phase involving mitochondrial nucleic acids and the cGAS-STING pathway.

**Systemically elevated type I IFN signalling in *m.5019A>G* mice**

To determine whether the innate immune response of *m.5019A>G* mice differs from that of WT mice, we administered a sub-lethal dose of LPS or PBS intraperitoneally and collected serum, kidney and lung tissues for immune profiling and assessment of sickness behaviour (Fig. 8a). Interestingly, circulating IL-17F was the only cytokine significantly elevated in the serum of *m.5019A>G* control mice (Fig. 8b). Consistent with the ex vivo findings in macrophages, *m.5019A>G* mice exhibited a robust and significant increase in serum IFN-α2 and IFN-β levels following LPS challenge (Fig. 8c, d). In contrast, levels of IL-22, IL-5, and the chemokine CCL5 were significantly reduced. Olink proteomic profiling also revealed elevated serum TNF-α, while IL-1β or IL-6 levels were unchanged at this timepoint, highlighting the marked complexity of innate immune signalling in vivo.

In the kidney, *m.5019A>G* mice showed reduced expression of respiratory chain complex subunits, elevated *Ifnb1* expression, and enhanced type I IFN signalling (Fig. 8e, f and Supplementary Fig. 10a), consistent with elevated circulating IFN-α2 and IFN-β levels. This included increased levels of several viral and host dsRNA sensors (PKR, OASL1, OAS1A, DHX58, ADAR, MDA5, SHFL, and MORC2A), the antiviral family of IFIT proteins, and IFI204, which can cooperate with cGAS to sense cytosolic DNA. Notably, even in the absence of LPS, several ISGs were also elevated in *m.5019A>G* control mice (Supplementary Fig. 10a). A similar, though less pronounced, response was observed in the lungs of *m.5019A>G* mice (Supplementary Fig. 10b–d). Finally, *m.5019A>G* mice displayed more severe sickness behaviour compared to WT (Fig. 8g). A summary of the key observations across ex vivo and in vivo *m.5019A>G* macrophages is provided in

Supplementary Fig. 10e. Importantly, this data confirms that an inherited heteroplasmic mtDNA mutation can perturb innate immune responses in macrophages and systemically in vivo.

## Discussion

An increasing number of clinical reports have identified that patients with primary mitochondrial disease are more susceptible to severe infections and sepsis[12–14,61,62]. In parallel, pre-clinical studies have shown that deletion or mutation of nuclear-encoded mitochondrial genes can lead to premature death, heightened sensitivity to endotoxemia, accelerated ageing, and hyperinflammatory macrophage responses[20–23,26,63,64]. A central aim of our study was to determine whether innate immune signalling is perturbed in the context of inherited heteroplasmic mtDNA mutations, which account for the majority of primary mitochondrial disease cases[5]. In line with this, our study demonstrates that biphasic IFN-β expression is elevated in *m.5019A>G* macrophages ex vivo and systemically in *m.5019A>G* mice following an LPS challenge, consistent with reports in *Polg*[D257A] mutator mice[20,51]. In the absence of stimulation, serum levels of IFN-α2 and IFN-β were not appreciably increased in *m.5019A>G* mice. However, elevated levels of circulating IL-17F, along with enhanced type I IFN signalling in lung and kidney tissues, were detected. These findings support the concept that mitochondrial dysfunction resulting from inherited mtDNA mutations perturbs innate immune signalling. Although type I IFNs are critical for anti-viral and anti-bacterial host defence, they can paradoxically drive pathology during infection and autoimmunity[20,65,66]. Notably, IL-17F has also been reported to synergise with type I IFNs to drive systemic autoimmune disease[67]. This aligns with a broader body of evidence indicating that type I IFN production must be tightly controlled to prevent immune-mediated pathology[53].

Supporting this, excessive type I IFN signalling is the hallmark of a set of Mendelian disorders first described in 2011, termed type I interferonopathies[68]. Similarly, increased IFN signatures have been reported in primary mitochondrial disease patients and may serve as novel clinical biomarkers[10,69]. Clinically, these patients exhibit heightened susceptibility to vaccine-preventable infections, and sepsis or SIRS remains a leading cause of morbidity and mortality[8,12,13,61]. Notably, intercurrent infection is reported to trigger neurological symptoms in patients with mtDNA disorders, suggesting that immune dysregulation contributes to disease progression[14]. Given that aberrant IFN signalling contributes to both sepsis[53,65] and neurological dysfunction in type I interferonopathies[70], our findings raise the possibility that inherited mtDNA mutations constitute a potential inborn error of immunity. This may underlie the susceptibility of mitochondrial disease patients to severe infection and immune-driven pathology, with significant implications for disease monitoring and therapeutic intervention.

To our knowledge, the only prior investigation addressing this question in mouse macrophages used the *m.5024C>T mt-Ta* mouse model[71]. Consistent with our observations, *m.5024C>T* macrophages tolerated high mutational burdens without evidence of purifying selection during differentiation, despite exhibiting impaired mitochondrial OxPhos[71]. Reduced MHC class II levels were also observed, aligning with our findings in *m.5019A>G* macrophages. However, that important study predominantly focused on purifying selection in adaptive immune cells and did not investigate how heteroplasmic mtDNA mutations impact innate immune signalling, an essential gap addressed by our work.

Mechanistically, elevated type I IFN responses downstream of mitochondrial dysfunction have previously been linked to cytosolic mtDNA release, while other studies implicate mtRNA release in enhancing IFN-β production in respiratory-deficient macrophages following LPS stimulation[20,33,37]. In line with these findings, *m.5019A>G* macrophages display disrupted cristae architecture, increased

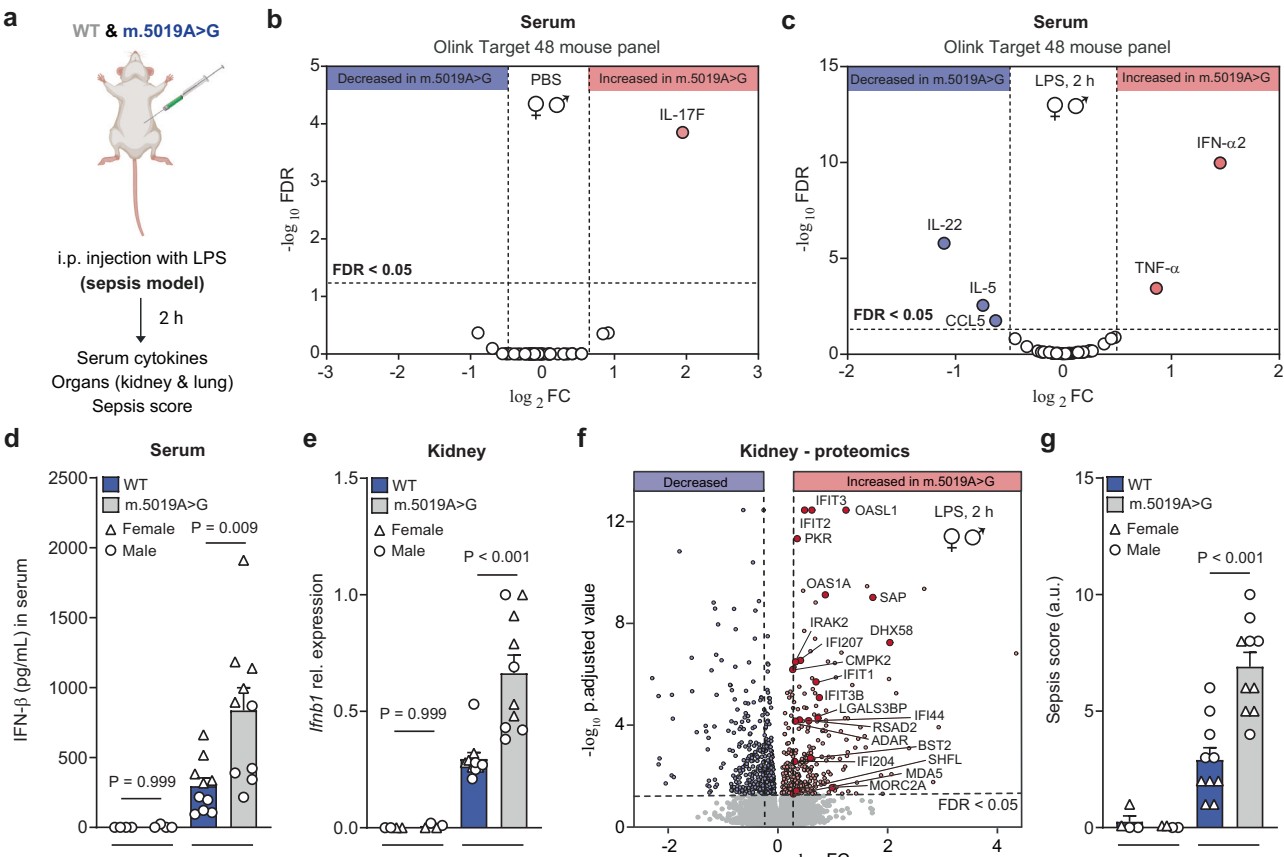

**Fig. 8 | Elevated type I IFN levels in *m.5019A>G* mice following an in vivo LPS challenge. a** Schematic of in vivo lipopolysaccharide (LPS) endotoxemia model experiment. **b, c** Volcano plot of Olink target T48 mouse cytokine and chemokine profiling of *m.5019A>G* mice vs wildtype (WT) mice serum injected intraperitoneally (i.p.) with phosphate-buffered saline (PBS) ($n = 4$; 2 h) or LPS ($n = 10$; 2 h). **d** Interferon-β (IFN-β) levels in serum of WT and *m.5019A>G* mice injected i.p. with PBS ($n = 4$; 2 h) or LPS ($n = 10$; 2 h). **e** *Ifnb1* expression in kidney tissue of *m.5019A>G* mice vs WT mice serum injected i.p. with PBS ($n = 4$; 2 h) or LPS ($n = 10$; 2 h) ($P = 0.0002$). **f** Volcano plot of kidney tissue proteomic analysis of *m.5019A>G* mice

vs WT mice injected i.p. with LPS ($n = 10$; 2 h). **g** Sepsis score of WT and *m.5019A>G* mice injected i.p. with PBS ($n = 4$; 2 h) or LPS ($n = 10$; 2 h) ($P = 0.00001699$). Data are log₂FC or mean ± s.e.m. *n* number represents independent biological replicates (mice) from a minimum of two independent experiments. *P*-values calculated using multiple two-tailed unpaired *t* tests corrected for multiple comparisons using the Benjamini, Krieger and Yekutieli method or one-way ANOVA corrected for multiple comparisons using the Tukey method. **a** Created in BioRender. Dwane, L. (2025) https://BioRender.com/kk5tnv3.

cytosolic DNA foci and elevated IFN-β expression after prolonged LPS stimulation. Notably, this late-phase IFN-β induction is partially attenuated by inhibition of the cGAS-STING pathway, indicating that cytosolic mtDNA sensing contributes to enhanced type I IFN signalling in this context. Similarly, inhibition of mitochondrial transcription, which depletes mtRNA, also reduces late-phase IFN responses. However, it remains unclear whether this reflects direct sensing of mtRNA or a secondary effect of impaired mitochondrial gene expression. In contrast to previous reports, we also observe an early-phase induction of IFN-β that is independent of mitochondrial nucleic acids. This response coincides with increased TLR4-mediated IRF3 phosphorylation, a process driven by TLR4 endocytosis and TRAM-TRIF-dependent signalling from endosomes[72]. These data suggest that mitochondrial dysfunction may modulate TLR4 endocytosis or endosomal signalling dynamics, although this hypothesis warrants further investigation.

Together, our findings underscore the multifaceted nature of type I IFN regulation in macrophages, involving both nucleic acid-dependent and -independent pathways. Interestingly, cytosolic mtDNA and mtRNA levels were elevated in *m.5019A>G* macrophages at rest, as determined by subcellular fractionation, without concurrent increases in macrophage activation. However, imaging failed to corroborate increased cytosolic DNA foci under basal conditions, raising the possibility that fractionation methods may overestimate cytosolic

nucleic acid abundance. Indeed, baseline expression of several ISGs was modestly reduced in *m.5019A>G* macrophages, in contrast to previous reports in *Polg*^D257A mutator macrophages. These findings highlight that the nature and timing of type I IFN signalling can differ depending on the mitochondrial defect, reflecting distinct modes of innate immune engagement. Finally, the upregulation of several viral and host dsRNA sensors, members of the anti-viral family of IFIT proteins, and IFI204, a sensor known to cooperate with cGAS, suggests that these pathways may also contribute to type I IFN induction in vivo, though their specific roles remain to be elucidated.

In addition to altered innate immune signalling, *m.5019A>G* macrophages exhibit pronounced mitochondrial dysfunction and metabolic reprogramming. At baseline, these cells show increased aerobic glycolysis, reduced respiratory chain complexes, and impaired oxidative TCA cycle activity and phosphorylation, consistent with defective mitochondrial translation. A key feature of this metabolic rewiring is a reduction in dihydrolipoamide dehydrogenase (DLD), the E3 subunit shared by multiple mitochondrial dehydrogenase complexes, including PDH and OGDH. Although the regulatory mechanism underlying DLD depletion remains unclear, its loss is accompanied by a marked remodelling of the mitochondrial proteome. Notably, DLD mRNA levels were unchanged in RNA-sequencing analyses, which suggests post-transcriptional regulation. Interestingly, post-

translational modification of OGDHC by fumarate-mediated succination has been reported to reduce enzyme activity in *Ndufs4*[-/-] cells with CI deficiency[73]. Given that fumarate is elevated in *m.5019A>G* macrophages, it is likely that a combination of reduced DLD, increased fumarate, and reductive stress contributes to impaired flux through PDHC and OGDHC. Upon LPS stimulation, *m.5019A>G* macrophages also exhibit elevated iNOS-mediated NO production, which is sustained in part by increased metabolic flexibility and shunting of carbon through the aspartate-argininosuccinate shunt. NO is a known inhibitor of both PDHC and OGDHC, as well as mitochondrial respiration[29,30,74], and has also been shown to promote type I IFN release in macrophages[36]. Thus, the higher levels of NO in *m.5019A>G* macrophages likely feedback to further suppress mitochondrial metabolism and may contribute to enhanced late-phase type I IFN we report following prolonged LPS stimulation. While this link between metabolic stress and IFN signalling remains correlative, it represents an avenue for future investigation.

A wide variety of intracellular bacterial and viral pathogens trigger the release of mtDNA as a danger-associated molecular pattern (DAMP) to activate innate immune response following infection[1]. In this context, one plausible explanation for the increased susceptibility of primary mitochondrial disease patients to severe infections and sepsis is aberrant mitochondrial DAMP signalling driven by mitochondrial dysfunction. However, the immunological consequences of such signalling appear highly dependent on the underlying genetic defect. For example, patients with mitochondrial recessive ataxia syndrome (MIRAS), which is caused by a common European founder mutation in *POLG1*, exhibit compromised antiviral sensing and delayed type I IFN responses. This failure to mount an effective early IFN response enables viral replication, ultimately exacerbating inflammation and disease severity[9]. These observations suggest that both hyperactive and insufficient type I IFN signalling can be deleterious in mitochondrial disease, paralleling patterns seen in monogenic inborn errors of immunity[53]. In conclusion, our heteroplasmic mtDNA mutation model provides a valuable platform to dissect the mechanistic links between inherited mtDNA mutations, type I IFN signalling, infection susceptibility and immune-mediated disease. These insights will be important in guiding future efforts to identify therapeutic strategies for mitigating infection-related morbidity and systemic inflammation in mitochondrial disease patients, and perhaps also more common inflammatory and autoimmune conditions.

## Methods

### Animals
All research conducted in this study complies with the relevant ethical regulations, followed ARRIVE guidelines, and procedures were approved by the University of Cambridge Animal Welfare and Ethical Review Body (AWERB) Committee. All mouse experiments and breeding were carried out in accordance with the UK Animals (Scientific Procedures) Act, 1986 (Home Office PPL no. PP1740969) and EU Directive 2010/63/EU. Wildtype (WT) mice were purchased from Charles River Laboratories, UK. The *m.5019A>G* mouse strain (Allele symbol: mt-T[am2Jbst], MGI ID: 6860509) was generously provided from the colony of Patrick F. Chinnery. The mice were provided through an MTA with James. B Stewart. All mice used in this study were on the C57BL/6 J background. Both females and males were used in adherence with 3 R's principles, ARRIVE 2.0 guidelines and due to the overall lack of binary sex differences observed in mitochondrial biology[75]. WT and *m.5019A>G* mice were age and sex matched for all experiments. Mice were kept in individually-ventilated cages (Tecniplast) at 20–24 °C, 45–65% humidity, with a 12 h light-dark cycle and with ad libitum access to SAFE 105 universal diet (Safe Diets) and water.

### Bone marrow-derived macrophages (BMDMs)
Mice aged 8–33 weeks old were euthanised by cervical dislocation, and death was confirmed by exsanguination. Bone marrow was harvested from the tibia and fibula. Cells were pelleted by centrifugation at 500 x *g* for 5 mins and red blood cells were lysed using a hypotonic red blood cell lysis buffer (Abcam, ab204733). Cells were pelleted by centrifugation at 500 x *g* for 5 mins and filtered through a 70 μm nylon mesh. Obtained cells were differentiated in DMEM (Gibco) containing L-929 (ATCC) conditioned medium (20%), Foetal Bovine Serum (10%) (FBS, Gibco A5670701) and penicillin-streptomycin (1%) (Gibco, 15070-063) for 6 days. Media was changed on day 4 and topped up on day 5. On day 6, BMDMs were scraped and counted using trypan blue, and then plated in DMEM with L929-conditioned medium (10%), FBS (10%) and penicillin-streptomycin (1%). BMDMs were plated at $0.5 \times 10^6$ cells/well (1 mL total volume) in 12-well cell culture plates (150628, ThermoFisher Scientific) and left overnight to adhere at 37 °C in a de-humidified incubator (21% $O_2$, 5% $CO_2$), unless otherwise stated. BMDMs were activated with LPS from *Escherichia coli*, serotype EH100 (Alexis), used at a final concentration of 100 ng/mL in all experiments, unless otherwise stated. Where stated, pre-treatment of BMDMs with Ruxolitinib (HY-50856; MedChemExpress; 1 μM) or an anti-IFNAR monoclonal blocking antibody (MAR1-5A3; ThermoFisher Scientific; 20 μg/mL) prior to LPS stimulation were used to inhibit IFNAR signalling. Mouse IgG Isotype antibody (ThermoFisher Scientific; 20 μg/mL) was used as a control for anti-IFNAR antibody experiments. Pre-treatment of BMDMs with IMT1 (HY-134539; MedChemExpress; 10 μM) for 24 h or RU.521 (HY-114180; MedChemExpress; 10 μM) for 1 h prior to LPS stimulation were used to inhibit mitochondrial transcription and the cGAS-STING pathway, respectively. ATP (Sigma; 5 mM) treatment of LPS-primed macrophages was used to activate the NLRP3 inflammasome, where indicated. WT macrophages were co-treated with recombinant mouse IFN-β (r-mIFN-β) (8234-MB; R&D systems; 20 ng/mL) and LPS, where indicated.

### Flow cytometry analysis
BMDMs were plated at $1 \times 10^6$ cells/well in 6-well cell culture plates (2 mL total volume) and left to adhere overnight at 37 °C in a de-humidified incubator (21% $O_2$, 5% $CO_2$). For macrophage differentiation, F4/80 cell surface marker (Invitrogen, 12-4801-80) was used according to the manufacturer's instructions. For mitochondrial mass and membrane potential measurements, cells were incubated with MitoTracker Green FM (M7514, ThermoFisher Scientific; 200 nM) or Tetramethyl rhodamine Methyl Ester Perchlorate (TMRM, 11550796, Invitrogen; 20 nM) according to the manufacturer's instructions. Cells were then washed twice with PBS, scraped and resuspended in DMEM with FBS (1%). Flow cytometry analysis was performed on a BD Fortessa flow cytometer and further analysed using FlowJo software.

### mtDNA heteroplasmy measurements
At weaning, ear skin biopsies were taken from all heteroplasmic *m.5019A>G* mice to allow each animal to be assigned a reference heteroplasmy value. DNA was extracted from ear skin biopsy samples, bone marrow and BMDMs using the DNeasy Blood & Tissue kit (69504, Qiagen) according to the manufacturer's protocol and quantified using a NanoDrop spectrophotometer (ThermoFisher Scientific). Heteroplasmy measurements were determined by pyrosequencing using the Q48 Autoprep or Q24 vacuum workstation systems (Qiagen) as previously described[27]. In brief, a section of the mitochondrial genome containing the mutation of interest was amplified from approximately 25 ng of DNA using the Pyromark PCR kit (978703, Qiagen) and primers (Integrated DNA technologies (IDT)) designed according to the manufacturer's instructions. Following PCR, 3 μL of Pyromark magnetic beads (974203, Qiagen) were loaded with 10 μL of biotinylated PCR product onto Pyromark Q48 disks (974901, Qiagen). Disks were

run on a Q48 Pyromark sequencer (Qiagen) using an allele quantification assay using the following dispensation order: TG/AAGGAC/TTGTAAG using a sequencing primer (Supplementary Table 1). Heteroplasmy was then called using Pyromark® analysis software (Qiagen) and reported as the percentage of mutant base present in the sample.

## Mitochondrial DNA copy number and levels measurement

Mitochondrial DNA (mtDNA) copy number was quantified by digital droplet PCR (ddPCR). DNA was isolated using DNeasy Blood & Tissue Kit (69504, Qiagen) according to the manufacturer's instructions. The DNA concentration of each sample was measured using a Nanodrop ND-8000 UV-visible spectrophotometer (ThermoFisher) and then adjusted to 10 ng/µL.

For ddPCR, a mixture of 1 µL DNA and 21 µL reaction mixture was made up and placed in a 96-well ddPCR plate (Bio-Rad). The plate was then sealed, vortexed for 20 secs, centrifuged with a pulse spin and placed on a chilled block (4 °C) (Bio-Rad). Droplets were generated using a QX200 AutoDG droplet generator with corresponding automated droplet-generating oil (1864110, Bio-Rad), DG32 automated droplet generator cartridges (1864108, Bio-Rad) and pipette tips for the AutoDG system (1864120, Bio-Rad). Each item was placed in the appropriate place in the instrument, the plate was unsealed and placed on the chilled block, in the droplet generator. Upon completion, the plate was removed and sealed with foil (1814040, Bio-Rad) using a plate sealer (Bio-Rad) set at 180 °C for 5 s. The plate was stored on ice until the next step. ddPCR was run on a C1000 Touch Thermal Cycler (BioRad) programmed to 95 °C for 30 s and 60 °C for 1 min repeated for 40 cycles followed by 90 °C for 10 mins. Temperature change ramp rates were set at 2 °C/sec. The plate was stored at 4 °C until analysis. Droplets were read by a QX200 Droplet Reader (Bio-Rad) using the following experimental settings: absolute quantification, rare event detection, copy number variation and supermix for probes with no dUTP. The results were analysed using QuantaSoft analysis software (Bio-Rad), and the average mtDNA copy number (HEX probe) was normalised to the nuclear DNA copy number (FAM probe). Primers and probes are in Supplementary Table 2.

## Extracellular flux analysis (Seahorse)

BMDMs were seeded at $0.2 \times 10^6$ cells/well in a Seahorse XFe24 well plate and left to adhere overnight at 37 °C in a de-humidified incubator (21% $O_2$, 5% $CO_2$). Oxygen consumption rate (OCR) and Extracellular acidification rate (ECAR) were analysed using an Energy Phenotype test, Seahorse XF Mito Stress test (103015-100, Agilent) or Seahorse XF Glycolysis Stress test (103020-100, Agilent) according to the manufacturer's instructions. In brief, a utility plate containing XF Calibrant fluid (100840-000, Agilent), together with the sensor cartridge, was placed in a $CO_2$-free incubator at 37 °C overnight. The following day, the medium was replaced with 500 µL of XF DMEM medium pH 7.4 (103575-100, Agilent) supplemented with glucose (10 mM), glutamine (2 mM), and sodium pyruvate (1 mM) for the Energy Phenotype and Seahorse XF Mito Stress test or glutamine (2 mM) for the Seahorse XF Glycolysis Stress test. The cell culture plate was then placed in a $CO_2$-free incubator at 37 °C for 45–60 mins prior to analysis on Seahorse XFe24 Analyser (Agilent). Seahorse XF Mito Stress Test Kit inhibitors (2 µM Oligomycin A, 1.0 µM FCCP, 0.5 µM Rotenone and 0.5 µM Antimycin A, final well concentration), Seahorse XF Glycolysis Stress Test Kit (10 mM glucose, 2 µM Oligomycin A and 50 mM 2-deoxyglucose (2-DG), final well concentration) and Energy Phenotype (200 ng/mL LPS, final well concentration) were added to the appropriate ports of the injector plate. After calibration of the utility plate and sensor cartridge, the cell culture plate was analysed on the Seahorse XF24 analyser using default test settings. All results were acquired with Wave software (Agilent) and analysed with Seahorse XF test report generators. For Seahorse XF Glycolysis Stress test, ECAR was converted to proton efflux rate (PER) according to the manufacturer's instructions. For

Seahorse XF Mito Stress test and XF Glycolysis Stress test, protein concentration was determined at experimental endpoints for normalisation of OCR and PER data to total protein (µg) using a Pierce BCA Protein Assay Kit (23225, ThermoFisher Scientific).

## Respirometry analysis (Oroboros system)

Oxygen consumption rate (OCR) in BMDMs was measured using an Oxygraph-2k high-resolution respirometer (Oroboros Instrument, Innsbruck, Austria) in 2 mL glass chambers at a constant temperature of 37 °C and stirrer speed of 750 rpm. Oxygen flux ($JO_2$), which is directly proportional to OCR, was continuously recorded with a 2 sec sampling rate using DatLab software 6.1 (Oroboros Instruments, Austria). Calibration at air saturation was carried out every day prior to experimentation, and all data were corrected for background instrumental $JO_2$ in accordance with the manufacturer's instructions. BMDMs were counted directly before resuspending in mitochondrial respiration medium (MIRO5) at $1 \times 10^6$ cells/mL. WT and *m.5019A>G* BMDMs were then added to individual Oxygraph-2k chambers to enable paired comparisons and allowed to reach a stable baseline OCR prior to stimulations. All reagents injected into the chambers (Digitonin, Malate, Glutamate, ADP, Cytochrome c, Succinate, FCCP and Rotenone) were warmed to room temperature prior to addition. Digitonin (14952-100mg-CAY, Cayman Chemical) was used to permeabilise the plasma membrane. Malate (M1000, Sigma) and glutamate (G1626, Sigma) were injected to determine complex I (CI) leak activity. ADP (A2754, Sigma) was subsequently injected to determine CI-dependent OxPhos. Cytochrome c (C7752, Sigma) was injected to ensure the integrity of the mitochondrial membrane. Succinate (S2378, Sigma) was injected to determine maximum OxPhos. FCCP (15218, Cambridge Bioscience) was injected incrementally to increase the concentration to that required to dissipate membrane potential to assess maximum electron transfer capacity (ETC). Rotenone (HY-B1756, Cambridge Bioscience) was subsequently injected to inhibit CI activity to determine CII ETC. The flux control ratio for CI was assessed from CI OxPhos and maximum OxPhos, while for CII was assessed from CII ETC and maximum ETC.

## $^{35}$S-methionine labelling of mitochondrial translation products

BMDMs were plated at $0.5 \times 10^6$ cells/mL (2 mL total volume) in a 6-well dish and left to adhere overnight at 37 °C in a de-humidified incubator (21% $O_2$, 5% $CO_2$). In order to label newly synthesised mitochondrial proteins, the previously published protocol was used[76]. Briefly, cells were incubated in methionine/cysteine-free medium for 10 mins before being replaced with methionine/cysteine-free medium containing 10% dialysed FCS and emetine dihydrochloride (100 µg/ml) to inhibit cytosolic translation. Following a 20 min incubation, 120 µCi/ml of [$^{35}$S]-methionine was added, and the cells were incubated for 30 min. After washing with PBS, cells were lysed, and 30 µg of protein was loaded on 10–20% Tris-glycine SDS-PAGE gels. Coomassie staining for total protein was also performed. Dried gels were visualised with a PhosphorImager system. The $^{35}$S-methionine signal intensity was subsequently normalised to the Coomassie staining to assess mitochondrial translation. Uncropped images can be found in the Source Data file.

## Confocal microscopy

BMDMs were plated at $0.25 \times 10^6$ cells/well (0.5 mL total volume) on coverslips in a 24-well plate and left to adhere overnight at 37 °C in a de-humidified incubator (21% $O_2$, 5% $CO_2$). Cells were stimulated as indicated. Cells were subsequently washed three times with PBS and fixed with paraformaldehyde (PFA) (4%) for 15 mins at room temperature. After washing, cells were permeabilised and non-specific binding was blocked with 0.1% Triton X-100, 3% BSA in PBS for 30 mins. Incubation with primary antibodies for TOM20 (11802-1AP, Proteintech), Cytochrome c (556432, BD Biosciences), DRP1 (BD

Biosciences) or DNA (Merck) was performed overnight at 4 °C in blocking buffer, followed by 3 washes with PBS. Incubation with Alexa Fluor 488 or 568 secondary antibodies (ThermoFisher Scientific) was performed for 30 mins at room temperature, followed by 3 washes with PBS. The slides were mounted using ProLong Gold Antifade Mountant (P36982, Invitrogen). All images were acquired using a Zyla 4.2 PLUS sCMOS camera attached to an Andor DragonFly 500 confocal spinning disk mounted on a Nikon Eclipse TiE microscope using a CFI Plan Apochromat lambda 100X oil immersion objective and using the Fusion user interface (Andor). Seven stacks of 0.2 μm were acquired using appropriate lasers. All images from the same experiment were acquired with the same parameters, including exposure time and laser intensities. Images were compiled by "max projection" and analysed with Fiji ImageJ (NIH). Mitochondrial morphology percentage per cell were done as previously described[77]. In brief, mitochondrial morphology was classified as fragmented/punctate, tubular or elongated, and the distribution was represented in a bar graph. Mitochondrial length quantification was done using a published Mito-Morphology macro and Fiji ImageJ[78]. In brief, one cell was defined as the region of interest for each analysis. Mitochondria within each cell were extracted into grayscale, inverted to show mitochondria-specific fluorescence as black pixels, and thresholded to optimally resolve individual mitochondria. The mean area/perimeter ratio was employed as an index of mitochondrial interconnectivity, with inverse circularity used as a measure of mitochondrial elongation and manually quantified. A macrophage-validated Fiji ImageJ macro, mitoMAPR[57,79], was also used for the assessment of branching morphology, including mitochondrial number, length, network structure, junction points, and junctions per network. In brief, this approach involves selecting a standardised region of interest (ROI; 15 × 15 μm) in the peripheral cytoplasm, where mitochondrial networks are more readily visualised. This ROI selection reduces the number of mitochondrial objects and networks per cell, enhancing the reliability and comparability of the branching analysis across cells. Co-localisation analysis between DRP1 and TOM20 was performed using a macro written for Fiji ImageJ[80], in an automated manner. This involved selecting three regions of interest per cell (ROI; 10 × 10 μm) in the peripheral cytoplasm. Cytosolic DNA foci were quantified manually using Fiji ImageJ. At least 20 cells per mouse per condition were analysed in each experiment. Antibodies used are listed in Supplementary Table 3.

## Super-resolution microscopy

BMDMs were plated at $0.5 \times 10^6$ cells/mL (0.5 mL total volume) on coverslips in a 24-well plate and left to adhere overnight at 37 °C in a de-humidified incubator (21% $O_2$, 5% $CO_2$). Following indicated treatments, BMDMs were fixed in PFA (5%), pH 7.4 for 15 min at 37 °C and then washed three times with PBS. After blocking with FBS (10%) for 30 mins, macrophages were incubated with ATP Synthase (MAB3494, Merck) and TOM20 (ab232589, Abcam) primary antibodies at 1:500 in FBS (5%) overnight at 4 °C. Cells were then washed with FBS (5%) three times and incubated with fluorescent secondary antibodies at 1:1000 for 1 h at room temperature with shaking, protected from light, and then washed three times with PBS. Cells were then incubated with Hoechst 33342 (1:500 in PBS) for 1 h at room temperature, washed three times with PBS, mounted onto glass slides using mounting medium (ProLong Diamond), left to dry for 12 h at room temperature, and then stored at 4 °C until imaging. Fixed cell super-resolution images for analysis of mitochondrial morphology in 3D were obtained with the Zeiss Elyra7 lattice SIM, using the Plan-Apochromat 63x/1.4 Oil DIC M27 objective with 15 phases and 0.091 μm intervals. Images were acquired with 20 ms exposure time with 405 nm (20.0%), 488 nm (4.0%), and 561 nm (6.0%) lasers. Standard deconvolution was performed in Zen Black. The "standard deconvolution" function in Zen black is a default selection to convert the raw image, which contains the moiré-like diffraction patterns, into a super-resolution image. SR-

Lattice SIM reconstruction: Raw phase images were reconstructed in ZEISS ZEN Black (v 2.3) using the 3D Lattice SIM module with a *linear SIM / generalised Wiener* approach (implementation proprietary to Zeiss). User-exposed settings were phases = 15, angles = 5, drift correction = on, OTF = system-measured beads, Wiener-regularisation = 0.003, noise/background suppression = default, apodisation = 0.85. The reconstructed volumes ($0.031 \times 0.031 \times 0.091$ μm voxels, 16-bit) were exported as TIFF stacks for analysis in Imaris. Reconstruction quality was verified by Fourier-space order separation and bead-based resolution cheques. For the analysis of mitochondrial morphology in 3D and quantification of ATP synthase puncta from the obtained SIM images, individual cells were cropped in ImageJ (Fiji, NIH). Using Imaris 10.1.0, objects in separate channels were segmented and rendered in 3D with the Surfaces function, as detailed in the Imaris reference manual. Segmentation setup included smoothing with Surfaces detail of 0.0986 μm, and background subtraction (Local Contrast) of 0.370 and 0.2 μm were used for TOM20 and ATP Synthase channels, respectively. Mitochondria typically have diameters between 1 and 0.2 μm. Machine Learning Segmentation was used for Hoechst channel with smoothing and Surfaces detail of 1 μm. Manual thresholding was used for TOM20 and ATP Synthase channels. For ATP Synthase channel, Split touching Objects (Region Growing) was enabled with an Intensity Based Seed Points Diameter of 0.2 μm. A surfaces filter was then applied to select objects above 10 voxels. 1 voxel = x*y*z = 0.0313 μm * 0.0313 μm * 0.0909 μm = 0.0000891 μm³. Small mitochondria are unlikely to be smaller than a sphere with a diameter of 0.2 μm, equal to a volume of $4.19 \times 10^{-3}$ μm³ = 0.00419 μm³. Therefore, objects excluded by this filter were approximately 5 times smaller than a very small mitochondrion. This means that any objects removed were likely to be background fluorescence and should not be included in the values for measuring mitochondrial shape (oblate ellipticity and sphericity). An ellipsoid is a type of quadratic that is a higher dimension analogue of an ellipse. The equation of a standard ellipsoid in an x-y-z Cartesian coordinate system is:

$$\frac{x^2}{a^2} + \frac{y^2}{b^2} + \frac{z^2}{c^2} = 1 \tag{1}$$

where a, b and c (the length of the three semi-axes) are fixed positive real numbers determining the shape of the Ellipsoid.

$$e_{oblate} = \frac{2b^2}{b^2 + c^2} \times \left(1 - \frac{2a^2}{b^2 + c^2}\right) \tag{2}$$

$e_{oblate}$ = oblate Ellipsoid, where greater oblate ellipticity indicates that the object is more flattened.

Surface – Sphericity

Sphericity is a measure of how spherical an object is and defined by Wadell in 1932. $\psi$ of a particle is the ratio of the surface area of a sphere (with the volume as the given particle) to the surface area of the particle:

$$\psi = \frac{\pi^{\frac{1}{3}}(6V_p)^{\frac{2}{3}}}{A_p} \tag{3}$$

Object statistics from Imaris surfaces were then recorded in Microsoft Excel. Antibodies used are listed in Supplementary Table 3.

## Transmission electron microscopy (TEM)

BMDMs were plated at $1 \times 10^6$ cells/mL (2 mL total volume) in a 35 mm dish with plastic cover slips and left to adhere overnight at 37 °C in a de-humidified incubator (21% $O_2$, 5% $CO_2$). After stimulation, cells were washed with PBS and then fixed in fixation buffer (2% glutaraldehyde, 2% formaldehyde) in 50 mM sodium cacodylate pH 7.4 containing 2 mM calcium chloride for 2 h at room temperature. After 2 h, cells

were kept in fixation buffer at 4 °C overnight. Cells were subsequently washed three times with washing buffer (50 mM sodium cacodylate pH 7.4) and samples were osmicated (1% osmium tetroxide, 1.5% potassium ferricyanide and 50 mM sodium cacodylate pH 7.4) for 3 days at 4 °C. After washes with deionised water, samples were treated with 0.1% (w/v) thiocarbohydrazide in deionised water for 20 mins at room temperature in the dark. Samples were then washed in deionised water and osmicated a second time for 1 h at room temperature (2% osmium tetroxide in deionised water). After washes with deionised water, samples were blockstained with uranyl acetate (2% uranyl acetate in 50 mM maleate pH 5.5) for 3 days at 4 °C. Then, samples were washed in deionised water and dehydrated in a graded series of ethanol (50%/70%/95%/100%/100% dry) and 100% dry acetonitrile, three times for each at least 5 mins. Samples were then infiltrated with a 50/50 mixture of 100% dry acetonitrile/Quetol resin (without BDMA) overnight, followed by three days in 100% Quetol (without BDMA). Then, samples were infiltrated for five days in 100% Quetol resin with BDMA exchanging the resin each day. The dishes were filled with resin to the rim, covered with a sheet of Aclar and cured at 60 °C for three days. After curing, the Aclar sheets were removed, and small blocks were cut from the dishes using a hacksaw. Thin sections (~70 nm) were prepared using an ultramicrotome (Leica Ultracut E). Resin blocks were oriented with the cell-side towards the knife and sections were collected on bare 300 mesh copper grids immediately when reaching the cell monolayer. Samples were then imaged with a Tecnai G2 TEM (FEI/ThermoFisher Scientific) run at 200 keV using a 20 μm objective aperture to improve contrast. Images were acquired using an ORCA HR high resolution CCD camera (Advanced Microscopy Techniques Corp, Danvers USA). Mitochondrial cristae architecture was assessed semi-quantitatively and mitochondrial aspect ratio (length/width) was measured quantitatively using Fiji (NIH). At least 12 images per mouse per condition were analysed in each experiment.

## Proteomic analysis

BMDMs at $0.5 \times 10^6$ cells/mL (5 mL total volume) were plated onto 6-cm dishes and left to adhere overnight at 37 °C in a de-humidified incubator (21% $O_2$, 5% $CO_2$) before being treated as indicated. At the experimental endpoint, cells were washed with PBS on ice, scraped and centrifuged at 300 x $g$ for 5 mins at 4 °C and frozen at −80 °C. Kidney and lung tissue were disrupted in 100 mM Tris-HCL, pH 8.5 5% SDS with a homogeniser (Precellys) prior to heating at 95 °C for 10 mins and subsequently frozen at −70 °C. The lysates were added to a KingFisher deep well plate prepared for an 8 h digest protocol on the Kingfisher Flex. Lysate was added to Plate G with MagReSyn HILIC beads and ACN is added to 70% final concentration. Plates D, E, and F are filled with 95% ACN and Plates B and C are filled with 70% EtOH. The digest buffer (1 μg/ml MS grade trypsin in 50 mM triethylammonium bicarbonate) is added to Plate A. Alternatively, cell pellets were lysed, reduced and alkylated in 50 μl of 6 M Gu-HCl, 200 mM Tris-HCl pH 8.5, 10 mM TCEP, 15 mM chloroacetamide by probe sonication and heating to 95 °C for 5 mins. Protein concentration was measured by a Bradford assay and initially digested with LysC (Wako) with an enzyme to substrate ratio of 1/200 for 4 h at 37 °C. Subsequently, the samples were diluted 10-fold with water and digested with porcine trypsin (Promega) at 37 °C overnight. Samples were acidified to 1% TFA, cleared by centrifugation (16,000 × $g$ at RT) and ~ 20 μg of the sample was desalted using a Stage-tip. Eluted peptides were lyophilised, resuspended in 0.1% TFA/water, and the peptide concentration was measured by A280 on a nanodrop instrument (Thermo). The sample was diluted to 2 μg/ 5 μl for subsequent analysis.

The tryptic peptides were analysed on a Fusion Lumos mass spectrometer connected to an Ultimate Ultra3000 chromatography system (both Thermo Scientific, Germany) incorporating an autosampler. 2 μg of de-salted peptides were loaded onto a 50 cm emitter packed with 25 cm Aurora columns (IonOptics, Australia) using a

RSLC-nano uHPLC systems connected to a Fusion Lumos mass spectrometer (both Thermo, UK). Peptides were separated by a 140 min linear gradient from 5% to 30% acetonitrile, 0.5% acetic acid. The mass spectrometer was operated in DIA mode, acquiring a MS 350-1650 Da at 120k resolution, followed by MS/MS on 45 windows with 0.5 Da overlap (200–2000 Da) at 30k with a NCE setting of 28.

Raw files were analysed and quantified by searching against the Uniprot *Mus Musculus* database using DIA-NN 1.8 (https://github.com/vdemichev/DiaNN). Library-free search was selected, and the precursor ion spectra were generated from the FASTA file using the deep learning option. Default settings were used throughout, apart from using "Robust LC (high precision)". In brief, Carbamidomethylation was specified as fixed modification while acetylation of protein N-termini was specified as variable. Peptide length was set to minimum 7 amino acids, precursor FDR was set to 1%. Subsequently, missing values were replaced by a normal distribution (1.8 π shifted with a distribution of 0.3 π) in order to allow the following statistical analysis. Protein-wise linear models combined with empirical Bayes statistics are used for the differential expression analyses. The Bioconductor package limma was used to carry out the analysis using LFQ-Analyst an R based online tool[81]. Heatmaps were generated using Morpheus software from the Broad Institute.

## RNA sequencing

BMDMs were plated at $0.5 \times 10^6$ cells/mL (2 mL total volume) in a 6-well dish and left to adhere overnight at 37 °C in a de-humidified incubator (21% $O_2$, 5% $CO_2$) before being treated as indicated. RNA isolation was carried using RNeasy Plus kit (74136, Qiagen) following manufacturer's suggestions and eluted RNA was purified using RNA Clean & Concentrator Kits (Zymo Research). RNA-seq sample libraries were prepared by Cambridge Genomic Services (CGS) using TruSeq Stranded mRNA (Illumina) following the manufacturer's description. For the sequencing, the NextSeq 75 cycle high output kit (Illumina) was used and samples spiked in with 1% PhiX. The samples were run using the NextSeq 500 sequencer (Illumina). Differential Gene Expression Analysis was done using the counted reads and the R package edgeR version 3.26.5 (R version 3.6.1) for the pairwise comparisons.

## Metabolomic analysis

BMDMs were plated at $0.5 \times 10^6$ cells/mL (1 mL total volume) in a 12-well dish and left to adhere overnight at 37 °C in a de-humidified incubator (21% $O_2$, 5% $CO_2$), before being treated as indicated. For stable isotope-assisted tracing experiments, glutamine free-DMEM (Gibco™) was supplemented with U-$^{13}$C-glutamine (Cambridge Isotope Laboratories) or glucose free-DMEM (Gibco™) was supplemented with U-$^{13}$C-glucose (Sigma) and replaced complete DMEM at the experimental start point. For the glucose- and glutamine-free experiments, DMEM without glucose and DMEM without glutamine were replaced at the experimental start point prior to being treated. Metabolite extraction buffer (MES) (methanol/acetonitrile/water, 50:30:20 v/v/v) was added (0.25 mL per $0.5 \times 10^6$ cells) and samples were incubated for 15 mins on dry-ice. The resulting suspension was transferred to ice-cold microcentrifuge tubes. For cell culture medium (CCM) analysis, 50 μL was added to 300 μL of MES. Samples were agitated for 20 mins at 4 °C in a thermomixer and then incubated at −20 °C for 1 h. Samples were centrifuged at maximum speed for 10 min at 4 °C. The supernatant was transferred into a new tube and centrifuged again at maximum speed for 10 min at 4 °C. The supernatant was transferred to autosampler MS vials and stored at −80 °C. Metabolite extracts were dried using an SC210A SpeedVac vacuum centrifuge (ThermoFisher Scientific) and reconstituted in an appropriate LC sample buffer, as detailed below. When necessary, further internal standards were added to the LC sample buffer: universal $^{15}$N$^{13}$C amino acid mix, succinate $^{13}$C$_4$; AMP $^{15}$N$_2$$^{13}$C$_{10}$; ATP $^{15}$N$_5$$^{13}$C$_{10}$; putrescine D8; dopamine D4. Internal standards were not included in the LC sample buffer for

[13]C labelling experiments. A Q Exactive Plus Orbitrap coupled to a Vanquish Horizon ultra-high performance liquid chromatography system (ThermoFisher Scientific) was used for LC-MS analysis. Several customised LC methods were used to separate aqueous metabolites of interest as previously published[82].

## HILIC method

Samples were reconstituted in 7:3 acetonitrile: water and analysed using a bridged ethylene hybrid (BEH) amide hydrophilic interaction liquid chromatography (HILIC) approach for the highly polar aqueous metabolites. The LC column used was the Acquity Premier BEH amide column (150 × 2.1 mm, 1.7 μm, Waters, cat 186009506). Mobile phase (A) was 100 mM ammonium carbonate, and mobile phase (B) was acetonitrile, with 1:1 water:acetonitrile being used for the needle wash. The flow rate was 0.5 mL/min and the injection volume was 5 μL. The following linear gradient was used: 20% (A) for 1.5 mins, followed by an increase to 60% (A) over 3.5 mins, a hold at 60% (A) for 1 min, a return to 20% (A) over 0.1 min and column re-equilibration for 3.9 mins. The total run time was 10 mins.

**C18pfp method**. An ACE Excel C18-PFP column (150 × 2.1 mm, 2.0 μm, Avantor, cat EXL-1010-1502U) was used to separate TCA cycle intermediates and amino acids. Samples were reconstituted in 10 mM ammonium acetate. For positive ion mode, mobile phase (A) was water with 10 mM ammonium formate and 0.1% formic acid, and mobile phase (B) was acetonitrile with 0.1% formic acid. For negative ion mode, mobile phase (A) was water with 0.1% formic acid, and mobile phase (B) was acetonitrile with 0.1% formic acid. The flow rate was 0.5 mL/min and the injection volume was 3-3.5 μL. The needle wash used was 1:1 water:acetonitrile. The following gradient was used: 0% (B) for 1.6 mins, followed by a linear gradient to 30% (B) over 2.4 mins, a further increase to 90% (B) over 0.5 mins, a hold at 90% (B) for 0.5 mins, a return to 0% (B) over 0.1 min, and re-equilibration for 1.4 mins. The total run time was 6.5 mins.

**BEH C18 anion exchange (high mass) method**. An Atlantis Premier NEH C18 AX column (Waters, cat 186009368) was used for the separation of higher mass species. Samples were reconstituted in 10 mM ammonium acetate prior to injection. Mobile phase (A) was 10 mM ammonium acetate, and mobile phase (B) was 9:1 ACN:H2O, 10 mM ammonium acetate, 0.1% ammonia. The flow rate was 0.4 mL/ min and the injection volume was 5 μL. The needle wash used was 1:1 water:acetonitrile. The following gradient was used: 0% (B) for 1.6 min, followed by a linear gradient to 30% (B) over 1.9 minutes, a further increase to 90% (B) over 1 min, a hold at 90% B for 0.5 mins, a return to 0% B over 0.1 min, with re-equilibration for 2.4 mins. The total run time was 6.5 mins.

Source parameters used for the orbitrap were an auxiliary gas heater temperature of 438 °C, a capillary temperature of 269 °C, an ion spray voltage of 3.5 kV and a sheath gas, auxiliary gas and sweep gas of 53, 14 and 3 arbitrary units respectively with an S-lens radio frequency of 85%. A full scan of 500–1000 m/z was used at a resolution of 70,000 ppm in positive ion mode.

To measure L-and D-enantiomers of 2-HG, diacetyl-L-tartaric anhydride- (DATAN, Merck, cat 336040050)-derivatised metabolite extracts from U-[13]C-glutamine labelling experiments were separated on an Acquity Premier HSS T3 column (1.8 μm, 2.1 × 100 mm, Waters, cat 186009468), as previously described[42]. The mobile phases were: (A) 1.5 mM ammonium formate (to pH 3.6 with formic acid) and (B) acetonitrile with 0.1% formic acid. The flow rate was 0.4 ml/min and the injection volume 5 μL. The column gradient used was: 3 to 5% (B) over 2.6 mins, followed by 5 to 95% (B) over 0.4 min, a hold at 95% (B) for 1.7 min, a return to 3% (B) over 1.3 mins and re-equilibration at 3% (B) for 6 mins, for a total run time of 12 mins.

Source parameters used for the Q Exactive Orbitrap were identical to the C18pfp method. Parallel reaction monitoring (PRM) of the following 6 transitions was performed in negative ion mode using a collision energy of 20: 363.0569 (2-HG M + 0 + DATAN) to 147.0299 (2-HG M + 0); 364.0603 (2-HG M + 1 + DATAN) to 148.0333 (2-HG M + 1); 365.0636 (2-HG M + 2 + DATAN) to 149.0366 (2-HG M + 2); 366.0670 (2-HG M + 3 + DATAN) to 150.0400 (2-HG M + 3); 367.0703 (2-HG M + 4 + DATAN) to 151.0433 (2-HG M + 4); and 368.0737 (2-HG M + 5 + DATAN) to 152.0467 (2-HG M + 5).

LC-MS data were analysed with a targeted approach using ThermoFisher Scientific Xcalibur. Peak areas were normalised to an appropriate internal standard. Fractional incorporation (%) of individual isotopomers was calculated from the sum of all isotopomers. For L- and D- 2-HG, the m/z ratios of 147.0299 (2-HG M + 0), 148.0333 (2-HG M + 1), 149.0366 (2-HG M + 2), 150.0400 (2-HG M + 3), 151.0433 (2-HG M + 4) and 152.0467 (2-HG M + 5) were extracted in Xcalibur prior to integration of L- and D-peaks and subsequent calculation of fractional incorporation.

## Coenzyme Q (CoQ) extraction and analysis

BMDMs were plated at $0.5 \times 10^6$ cells/mL (2 mL total volume) in a 6-well dish and left to adhere overnight at 37 °C in a de-humidified incubator (21% $O_2$, 5% $CO_2$). The following day, cells were treated with vehicle (ethanol) control or antimycin A (Ant A) (5 μM, Sigma) as a positive control for 5 mins prior to being washed twice in cold PBS and then scraped into 200 μL PBS, added to a mixture of 300 μL acidified methanol (0.1% (w/v) HCl) and 250 μL hexane, and vortexed to extract CoQ, as previously described[57,83]. In brief, the hexane phase was separated by centrifugation at $17,000 \times g$ for 5 mins. The hexane layer was transferred in mass spectrometry (MS) vials and dried under $N_2$. The crude residue was resuspended in methanol containing ammonium formate (2 mM), overlaid with argon and analysed by LC-MS/MS using a I-class Acquity LC system attached to a Xevo TQ-S triple quadrupole mass spectrometer (Waters). Samples were stored in a refrigerated autosampler and 2 μL of sample was injected into a 15 μL flowthrough needle and separated by RP-HPLC at 45 °C using an Acquity column (Waters). The isocratic mobile phase was ammonium formate (2 mM) in methanol used at 0.8 mL/min over 5 mins. The mass spectrometer was operated in positive ion mode with multiple reaction monitoring. The following settings were used for electrospray ionisation in positive ion mode: capillary voltage – 1.7 kV; cone voltage – 30 V; ion source temperature – 100 °C; collision energy – 22 V. Nitrogen and argon were used as curtain and collision gases, respectively. Transitions used for quantification were: $UQ_9$, 812.9 > 197.2; $UQ_9H_2$, 814.9 > 197.2. Samples were quantified using MassLynx 4.1 software to determine the peak of area for $UQ_9$ and $UQ_9H_2$.

## ATP, ADP and AMP measurements

BMDMs were plated at $0.5 \times 10^6$ cells/mL (2 mL total volume) in a 6-well dish and left to adhere overnight at 37 °C in a de-humidified incubator (21% $O_2$, 5% $CO_2$). The extraction buffer used was 4:1 MeOH: water with 5 μM ATP-[13]C (710695, Merck) as the internal standard (IS). Keeping the plate on ice, cells were quickly washed twice in PBS, 1 mL of extraction buffer was added and then scraped, transferred to a microcentrifuge tube and stored at – 80 °C for 1 h. A 0.2 μm PTFE filter was used to remove the protein in the sample, and diluted 10-fold using the extraction buffer with IS prior to injection on a HPLC column (Shimadzu) mass spectrometer (Waters). An Atlantis Premier BEH Z-HILIC 1.7 μm, 2.1 × 150 mm (Waters) was used. Mobile phase A was 15 mM ammonium acetate with 0.1% ammonium, and mobile phase B was 100% MeOH. The flow rate was 0.2 mL/min, and the injection volume was 5 μL. Peak areas were normalised to an appropriate internal standard. ATP, ADP and AMP concentrations in the samples were determined using a standard curve and the ratios assessed.

## Oxylipin analysis

BMDMs at $1 \times 10^6$ cells/mL (2 mL total volume) were plated onto 6-cm dishes and left to adhere overnight at 37 °C in a de-humidified incubator (21% $O_2$, 5% $CO_2$). Medium was replaced with phenol red-free DMEM (Gibco$^T$) before being treated as indicated. Cell culture supernatant was removed and stored at −80 °C prior to extraction. Cell culture supernatant samples were spiked with 2.1-2.9 ng 12-HETE-d8, 15-HETE-d8, LTB4-d4 and PGE$_2$-d4 standards (Cayman Chemical) prior to extraction and downstream processing as previously described[84]. Lipids were extracted by adding a 2.5 mL solvent mixture (1 M acetic acid/isopropanol/hexane; 2:30:30, v/v/v) to 1 mL of cells or cell culture supernatant in a glass extraction vial and vortexed for 1 min. 2.5 mL hexane was added to samples after vortexing for 1 min, and tubes were centrifuged at $500 \times g$ for 5 mins at 4 °C to recover lipids in the upper hexane layer (aqueous phase), which was transferred to a new tube. Aqueous samples were re-extracted as above by addition of 2.5 mL hexane, and upper layers were combined. Lipid extraction from the lower aqueous layer was then completed according to the Bligh and Dyer technique using sequential additions of methanol, chloroform and water, and the lower layer was recovered following centrifugation as above and combined with the upper layers from the first stage of extraction. Solvent was dried under vacuum and lipid extract was reconstituted in 100 μL HPLC grade methanol. Lipids were separated by liquid chromatography using a gradient of 30–100% Acetonitrile: Methanol – 80:15 + 0.1% Acetic acid over 20 mins on an Eclipse Plus C18 Column (Agilent) and analysed on a Sciex QTRAP 7500 LC-MS/MS system. Source conditions: TEM 475 °C, IS −2500, GS1 40, GS2 60, CUR 40. Lipids were detected using MRM monitoring with the following parent to daughter ion transitions: PGD$_2$ [M-H] – 351.2/271.1, PGE$_2$ [M-H] – 351.2/271.1, 11-HETE [M-H] – 319.2/167.1, 8,9-DiHETrE [M-H] – 337.2/127.1, 14,15-DiHETrE [M-H] – 337.2/207.1, 17,18-DiHETE [M-H] – 335.2/247.1. Deuterated internal standards were monitored using parent to daughter ion transitions of 12-HETE-d8 [M-H] – 327.2/184.1, PGE$_2$-d4 [M-H] – 355.2/275.1, 15-HETE-d8 [M-H] – 327.2/226.1, LTB4-d4 [M-H] – 339.2/197.1. Chromatographic peaks were integrated using Sciex OS 3.3.0 software (Sciex). Peaks were only selected when their intensity exceeded a 5:1 signal-to-noise ratio with at least 7 data points across the peak. The ratio of analyte peak areas to internal standard were taken and lipids quantified using a standard curve made up and run at the same time as the samples.

## Western blot

BMDMs were plated at $0.5 \times 10^6$ cells/mL (1 mL total volume) in a 12-well dish and left to adhere overnight at 37 °C in a de-humidified incubator (21% $O_2$, 5% $CO_2$), before being treated as indicated. Cells were scraped in lysis buffer (Pierce RIPA Lysis and Extraction buffer, 89900, ThermoFisher Scientific) supplemented with 1X Protease Inhibitor Cocktail (ab271306, Abcam) and 5 μL/mL Benzonase Nuclease (E1014-5KU, Merck). Total protein concentrations in samples were measured using a Pierce BCA Protein Assay Kit (23227, ThermoFisher Scientific) and diluted to the same concentration as required. Samples were denatured and reduced for SDS-PAGE in 4X Bolt LDS Sample Buffer (B0007, Invitrogen) and 10X Bolt Sample Reducing Agent (B0009, Invitrogen) at 95 °C for 5 mins. For total OxPhos assessment, samples were left at RT for 20 mins instead of heating. Equal protein amounts were loaded and separated on Bolt 4–12% Bis-Tris Plus Mini Protein Gels 4–12% (NW04125BOX, Invitrogen). After SDS-PAGE, proteins were semi dry-transferred to nitrocellulose membranes (IB23001, ThermoFisher Scientific) for 12 mins at 20 V. Membranes were blocked in 5% BSA or milk in TBS-Tween 0.1% for 1 h at RT and subsequently incubated with primary antibodies diluted in 5% BSA or milk in TBS-Tween 0.1% overnight at 4 °C. After washing membranes three times with TBS-Tween 0.1%, membranes were incubated with a secondary antibody solution in 5% BSA or milk for 30 mins at RT. Lastly, membranes were washed three times with TBS-Tween 0.1% and imaged using an Amersham Imager 680 and the SignalFire Plus ECL Reagents (6883, Cell Signalling). Membranes were exposed for different durations to ensure proper visualisation of all proteins. Blots were subsequently analysed using Image Lab software (Bio-Rad). Uncropped blots can be found in the Source Data file. Antibodies used are listed in Supplementary Table 3.

## Nitrite assay

BMDMs were plated at $0.5 \times 10^6$ cells/mL (1 mL total volume) in a 12-well dish and left to adhere overnight at 37 °C in a de-humidified incubator (21% $O_2$, 5% $CO_2$), before being treated as indicated. Nitric oxide production was assessed by measuring extracellular nitrite levels using the Griess assay (G2930, Promega) according to the manufacturer's instructions. In brief, 50 μL extracellular media from each sample was placed in a 96-well plate in duplicate. An equal volume of sulfanilamide solution was added to each well and left for 5–10 mins at room temperature (RT), protected from light. Then 50 μL N-(1-naphthyl) ethylenediamine (NED) solution was added to each well and incubated for 5–10 mins at RT, protected from light. Absorbance was measured at 540 nm, and nitrite levels were quantified against a standard curve made up using 0.1 M stock of sodium nitrite.

## Extracellular lactate measurements using Lactate-Glo™ Assay

BMDMs were plated at $0.5 \times 10^6$ cells/mL (1 mL total volume) in a 12-well dish and left to adhere overnight at 37 °C in a de-humidified incubator (21% $O_2$, 5% $CO_2$). The medium was changed prior to being treated as indicated. Extracellular lactate concentration in the cell culture medium was determined using a Lactate-Glo$^T$ Assay (J5021, Promega), according to the manufacturer's instructions. In brief, 50 μL extracellular media from each sample was placed in a 96-well plate in duplicate. An equal volume of lactate detection reagent was added to each well and left to incubate for 60 mins at RT. Luminescence was subsequently recorded using a plate-reading luminometer.

## Digitonin fractionation

BMDMs were plated at $1 \times 10^6$ cells/mL (2 mL total volume) in a 6-well dish and left to adhere overnight at 37 °C in a de-humidified incubator (21% $O_2$, 5% $CO_2$). Cells were subject to digitonin fractionation as previously described[33]. In brief, cells were washed once with room-temperature PBS, scraped on ice into ice-cold PBS and pelleted at 500 g for 5 mins at 4 °C. The supernatant was discarded, and the pellet was resuspended in 400 μL of extraction buffer (150 mM NaCl, 50 mM HEPES, pH 7.4, and 25 μg.ml$^{-1}$ digitonin). Samples were rotated at 4 °C for 10 mins before being centrifuged at $2000 \times g$ at 4 °C for 5 mins. The resulting supernatant constituted the cytosolic fraction, from which RNA and DNA were isolated using the RNeasy® Plus kit (74136, Qiagen) or the DNeasy Blood & Tissue kit (69504, Qiagen). Alternatively, the cytosolic fraction was concentrated with Strataclean resin (Agilent) and analysed by western blot. The pellet containing membrane-bound organelles was lysed for analysis by western blotting. To detect mtRNA and mtDNA in the cytosol, qPCR was performed using primers specific for *mt-Nd1* and *mt-Co3* on cDNA reverse-transcribed from cytosolic RNA (mtRNA) and on DNA isolated from the cytosolic fraction (mtDNA).

## Olink target T48 mouse cytokine and chemokine panel

BMDMs at $0.5 \times 10^6$ cells/mL (5 mL total volume) were plated onto 6-cm dishes and left to adhere overnight at 37 °C in a de-humidified incubator (21% $O_2$, 5% $CO_2$). Medium was replaced with DMEM (Gibco) before being treated as indicated. Cell culture medium was removed, centrifuged at $300 \times g$ for 5 mins to remove any cell debris, before an aliquot was taken and transferred to a microcentrifuge tube and stored at −80 °C. Proteins were measured using the Olink® Target 48 Cytokine Panel (Olink Proteomics AB, Uppsala, Sweden), which employs Proximity Extension Assay (PEA) technology to simultaneously analyse

43 analytes with only 1 μL of each sample. In this method, pairs of oligonucleotide-labelled antibody probes bind to their respective target proteins and when in close proximity, the oligonucleotides hybridise in a pair-wise manner. The addition of DNA polymerase triggers proximity-dependent DNA polymerisation, generating a unique PCR target sequence. The resulting DNA sequence is subsequently detected and quantified using a microfluidic real-time PCR instrument (Olink Q100 machine). Samples were run in the Stratified Medicine Core Laboratory (SMCL) NGS Hub, Department of Medical Genetics, University of Cambridge. Three internal controls are added to each sample, the incubation control, the extension control and the detection control. The extension control is used for the data normalisation of each sample, but is not used as a quality control measure. The incubation control and the detection control are used to monitor the quality of assay performance, as well as the performance of individual samples. Three external controls are included in the kit and analysed on each sample plate, negative control, sample control and calibrator. Negative controls are analysed in duplicates and sample control and calibrator are analysed in triplicates on each plate. The calibrator is used for data normalisation, and both calibrator and sample control are used to monitor the quality of assay performance. Only data from runs that meet these quality control criteria are reported. Data is provided as normalised protein expression (NPX). For more detailed information, see panel-specific validation data and the Olink NPX Signature manual available at the Olink website (www.olink.com).

## Enzyme-linked immunosorbent assay (ELISA)

BMDMs were plated at $0.5 \times 10^6$ cells/mL (1 mL total volume) in a 12-well dish and left to adhere overnight at 37 °C in a de-humidified incubator (21% $O_2$, 5% $CO_2$) prior to being treated as indicated. Concentrations of secreted IFN-β, IL-6, IL-1β and TNF-α were determined using the corresponding ELISA kits (IFN-β DY8234, IL-6 DY406, IL-1β DY401, and TNF-α DY410, R&D systems). In brief, MaxiSorp Nunc-Immuno 96-well plates (439454, ThermoFisher) were incubated overnight at RT with a capture antibody. After three washes with 0.05% Tween 20 (P1379, Sigma) in PBS, the plates were blocked for an hour with 1% BSA in PBS. Incubation with the sample or standards was performed at 4 °C overnight after three washes with 1% BSA in PBS. Experiments were conducted using two technical replicates of 50 μL of culture media. For IL-6 and TNF-α, the culture media was diluted in reagent diluent 1:2 and 1:5, respectively. Subsequently, plates were washed and incubated with the appropriate detection antibody for two hours at RT. After washing, a working solution of 50 μL streptavidin-HRP was added to the plates for 20 mins, and the plate was covered from direct light. After washing, 50 μL substrate solution (1:1 mixture of $H_2O_2$ and tetramethylbenzidine (R&D Systems) was added to all wells and left to incubate for approximately 20 mins. The reaction was stopped using 25 μL 2 M $H_2SO_4$, and optical density measurements were collected at 450 nm on a SpectraMax Plus 384 microplate reader. Concentrations were calculated using the corresponding standard curve after accounting for the dilution of the sample in the assay when necessary.

## RT-qPCR

BMDMs were plated at $0.5 \times 10^6$ cells/mL (1 mL total volume) in a 12-well dish and left to adhere overnight at 37 °C in a de-humidified incubator (21% $O_2$, 5% $CO_2$) prior to being treated as indicated. Kidney and lung tissue from in vivo experiments were disrupted with a homogeniser (Precellys) in RNA extraction buffer. mRNA expression was quantified by quantitative RT-qPCR. RNA extraction was performed as described in the manufacturer's protocol (RNeasy Plus Mini Kits 74136, Qiagen). cDNA was synthesised with the High-Capacity cDNA Reverse Transcription kit (ThermoFisher Scientific). The reaction was performed in a MicroAmp optical 96-well reaction plate (N8010560, Applied Biosystems) in a 96-well QuantStudio 3 PCR

machine with PowerUp SYBR Green Master Mix (A25741, Applied Biosystems) using primers listed in Supplementary Table 4. *Rps18* was used as a normalisation control. Oligonucleotide primers were designed using NCBI Primer-Blast and custom-synthesised by Merck (Scale: 0.05 μmole, purification: desalted, modifications: unmodified, format: dry).

## Salmonella typhimurium (STM) infection assay

Cells were primed for 4 h with LPS from *E. coli* serotype EH100 (Alexis) at a concentration of 100 ng/mL. *Salmonella enterica* serovar Typhimurium (*S.* Typhimurium) strain SL1344 was grown in low-salt lysogeny broth (LB) medium (ThermoFisher Scientific) at 37 °C in an orbital shaker. Overnight *Salmonella* cultures were sub-cultured 1:10 in fresh LB and grown until mid-exponential phase ($OD_{600}$ = 0.8–1.2). Prior to infection, bacteria were pelleted by centrifugation (10,900 x *g*, 1.5 mins), washed, and resuspended in Opti-MEM (Gibco). Bacteria were added to confluent cells in 96-well plates (BMDMs, $5 \times 10^4$ cells/well) at different multiplicities of infection (MOIs), as described in the figure legends. Plates were then incubated for 30 mins at 37 °C. Non-internalized bacteria were then removed by washing cells with pre-warmed medium. Cells were incubated with Opti-MEM containing 10 μg/ml of gentamycin to eliminate any remaining extracellular bacteria. At specified time points postinfection (p.i.), cells were either processed for Colony-Forming Unit (CFU) analysis, Lactate Dehydrogenase Cytotoxicity assay (LDH), or ELISA. To determine CFU, infected cells were gently washed with PBS and lysed with water containing 0.2% Triton X-100 at the indicated time points. Bacteria were then serially diluted and plated onto LB agar. Cell death was quantified by measuring LDH release into the supernatant, using the LDH cytotoxicity detection kit (CytoTox 96 Non-Radioactive Cytotoxicity Assay, Promega). To normalise for spontaneous cell lysis, the percentage of cell death was calculated as follows: ($LDH_{sample}$ − $LDH_{negative\ control}$)/($LDH_{positive\ control}$ − $LDH_{negative\ control}$) × 100. The level of IL-1β release was measured by ELISA (IL-1β DY401, R&D systems).

## Endotoxemia model

Both male and female WT and *m.5019A>G* mice (8–12 weeks old) were used for these experiments. Mice were injected intraperitoneally with PBS or Ultrapure LPS from *Escherichia coli* O55:B5 (2.5 mg/kg, Sigma). After 2 h, mice were euthanised in a $CO_2$ chamber and blood was collected from the Vena Cava. Blood was centrifuged at 300 x *g* for 10 mins at 4 °C, and the serum was collected for analysis using an Olink target T48 mouse cytokine and chemokine panel and an IFN-β ELISA (IFN-β DY8234, R&D systems). Lung and kidney tissue was subsequently removed and snap frozen using liquid nitrogen ($LN_2$). The pain response was also assessed using the NC3Rs grimace scale for mice, with 0 = not present, 1 = moderately present and 2 = obviously present. A behavioural analysis of the mice post-injection with PBS or LPS was performed to generate a sepsis score, using a method adapted from ref. 85. In brief, the movement behaviour, flight reaction and body position were assessed and assigned a score in a blinded manner from video recordings. This also included an in-person assessment of the pain response as a parameter using the NC3Rs grimace scale for mice. The sum of these scores constituted the sepsis score.

## Quantification and statistical analysis

Details of all statistical analyses performed can be found in the figure legends. Data were expressed as mean ± standard error of the mean (s.e.m) or standard deviation (s.d.), unless stated otherwise. Graphpad Prism 10.4.2 was used to calculate statistics in plots using appropriate statistical tests depending on the data, including two-tailed unpaired *t* test, one-way ANOVA and multiple *t* tests. Adjusted *p*-values were assessed using appropriate correction methods, such as the Kruskal-Wallis method, Benjamini, Krieger and Yekutieli method, Tukey method, and Holm-Sidak tests. Scaling of features (metabolites or

proteins) was used for heatmap generation from metabolomics. For proteomics data, protein signal intensity was converted to a $log_2$ scale and biological replicates were grouped by experimental condition. Protein-wise linear models combined with empirical Bayes statistics were used for the differential expression analyses. The Bioconductor package limma was used to carry out the analysis using LFQ-Analyst an R based online tool[81]. Data were visualised using a Volcano plot, which shows the $log_2$ fold change (FC) on the $x$-axis and the $-log_{10}$ adjusted $p$-value on the $y$-axis. The cut-off for significant hits in the proteomics, Olink mouse T48 panel and RNA seq were set to false discovery rate (FDR) < 0.05. Overrepresentation analysis (ORA) of differentially abundant proteins and differentially expressed genes were determined using ShinyGO[86]. GSEA analysis of RNA seq was performed using the Broad Institutes GSEA 4.1.0[87]. Sample sizes were determined on the basis of previous experiments using similar methodologies. All depicted data points are biological replicates taken from distinct samples (mice), unless stated otherwise. Each figure consists of a minimum of 3 biological replicates, as indicated in the figure legends, from multiple independent experiments.

## Reporting summary

Further information on research design is available in the Nature Portfolio Reporting Summary linked to this article.

## Data availability

The RNA-sequencing datasets generated in this study have been deposited in Dryad under accession https://doi.org/10.5061/dryad.ksn02v7fn and in ArrayExpress under accession code **E-MTAB-15591** (https://www.ebi.ac.uk/biostudies/arrayexpress/studies?query=E-MTAB-15591). The proteomics datasets have been deposited in PRIDE under accession code **PXD060284** (https://www.ebi.ac.uk/pride/archive/projects/PXD060284). The raw lipidomics dataset has been deposited in the Cardiff University Research Data Repository (https://doi.org/10.17035/cardiff.30074749), together with the Oxylipin scheduled multiple reaction monitoring (sMRM) method report (https://doi.org/10.5281/zenodo.17078883). The raw metabolomics datasets have been deposited in MassIVE under accession code **MSV000099126** (https://doi.org/10.25345/C58C9RH3G). All other data are available in the article and its Supplementary files or from the corresponding author upon request. Source data are provided in this paper.

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

## Acknowledgements

We thank Alexandra Karcanias, Julien Bauer and Stephanie Wenlock of Cambridge Genomic Services (CGS), Department of Pathology, University of Cambridge, for RNA sequencing and bioinformatic analysis services. We thank the Stratified Medicine Core Laboratory (SMCL) NGS Hub, Department of Medical Genetics, University of Cambridge, for the Olink Services. We thank the Cambridge Advanced Imaging Centre (CAIC), University of Cambridge, for the assistance with Transmission Electron Microscopy (TEM). We thank Reiner Schulte and the staff of the Cambridge Institute for Medical Research (CIMR) flow cytometry core facility for training and support for flow cytometry analysis. We thank Alice Sowton of the Kunji lab for Oroboros training in the Medical Research Council (MRC) Mitochondrial Biology Unit (MRC MBU). We thank Jelle van den Ameele and members of the Murphy lab in the MRC MBU for helpful discussions and feedback. D.G.R. was supported by funding from the UKRI MRC (MC_UU_00028) and a Wellcome Trust-Academy of Medical Sciences (AMS) springboard grant (G123514). J.P. was supported by the UKRI MRC (MC_UU_00028/5). C.A.P. and M.M. were supported by core funding from UKRI MRC (MC_UU_00028/3) and the UKRI MRC award (MC_PC_21046) to the National Mouse Genetics Network Mitochondria Cluster (MitoCluster). A.K. was supported by a Wellcome Trust investigator award (222497/Z/21/Z) and Leona M & Harry B Helmsley Charitable Trust (R-2408-07256). A.v.K and C.P. were supported by funding from the Wellcome Trust (Multiuser Equipment 208402/Z/17) and a UKRI MRC fellowship (MR/X01293X/1).

## Author contributions

D.G.R. conceptualised the project. E.M. and D.G.R. were the lead experimentalists, designed all experiments, analysed and visualised the data and co-wrote the paper with input from all authors. S.P.B. and P.F.C. assisted with pyrosequencing, mtDNA copy number assessment and provided important intellectual input and assistance with the *m.5019A>G* mouse model. A.M.C., C.S.Y. and M.P.M. assisted with targeted LC-MS analysis for CoQ and ATP, ADP and AMP. R.J.S. and A.K. assisted with metabolomics and stable isotope-assisted tracing. K.T., Y.M.K. and E.M. assisted with mouse breeding, colony management, tissue acquisition and in vivo experiments using the *m.5019A>G* mice. A.M.C. assisted with blinded sepsis scoring. D.M.W. assisted with immunofluorescence, super-resolution microscopy and image analysis. S.R.C., V.P. and J.P. assisted with confocal microscopy analysis. M.D and C.E.B. assisted with the *Salmonella typhimurium* infection experiments. V.J.T. and V.B.O.D. assisted with the oxylipin profiling. R.K. assisted with metabolomic sample preparation, IMT1 optimisation and Griess assay. C.A.P. and M.M. assisted with ³⁵S-methionine labelling experiments. J.B.S. developed the *m.5019A>G* mouse model and provided input. C.P. and A.v.K. assisted with the proteomics. J.P., M.P.M., M.M., V.B.O.D., C.E.B., A.K., P.F.C. and A.v.K oversaw a portion of the research. D.G.R. obtained the funding, lead the study and oversaw the research programme.

## Competing interests

The authors declare no competing interests.

## Additional information

[1]MRC Mitochondrial Biology Unit, School of Clinical Medicine, University of Cambridge, Cambridge Biomedical Campus, Cambridge, UK. [2]Cambridge Institute of Therapeutic Immunology and Infectious Disease, Jeffrey Cheah Biomedical Centre, University of Cambridge, Cambridge, UK. [3]Division of Gastroenterology and Hepatology, Department of Medicine, University of Cambridge, Addenbrooke's Hospital, Cambridge, UK. [4]Department of Medicine, Addenbrooke's hospital, Cambridge Biomedical Campus, Cambridge, UK. [5]Division of Infection and Immunity, School of Medicine, Cardiff University, Cardiff, UK. [6]Cancer Research UK Centre, Institute of Genetics and Cancer, University of Edinburgh, Edinburgh, UK. [7]Biosciences Institute, Faculty of Medical Sciences, Newcastle University, Newcastle upon Tyne, UK. [8]Department of Clinical Neurosciences, School of Clinical Medicine, University of Cambridge, Cambridge Biomedical Campus, Cambridge, UK. [9]School of Biochemistry and Immunology, Trinity Biomedical Sciences Institute, Trinity College, Dublin 2, Ireland. ✉e-mail: RYAND67@tcd.ie

