## [Transparent Peer Review file · Nature Communications]

An inherited mitochondrial DNA mutation remodels inflammatory cytokine responses in macrophages and in vivo

Corresponding Author: Dr Dylan Ryan

Version 0:

Reviewer comments:

Reviewer #1

(Remarks to the Author)

The manuscript by Marques et al shows that the m.5019A>G mutation is associated with mitochondrial defects and remodeling of IFN-I release by macrophages. The authors utilized several state-of-the-art technologies to address their questions and this is highly appreciated by the reviewer. The manuscript is well written, logically organized and the scientific issue is well-identified and clearly formulated. There some major and minor issues to address.

Major:

The authors stated that the key finding of their study was that IFN-I release is elevated in m.5019A>G macrophages, providing evidence to support the idea that mitochondrial dysfunction arising from inherited mtDNA mutations perturb innate immune system signaling. Which is the mechanistic link between mitochondrial dysfunction and innate immune signaling perturbations? Is the inflammasome involved? Several studies reported that mtDNA can be released from mitochondria to the cytosol and trigger innate immune response, as it is sensed by the cGAS-MITA/STING-inflammasome pathway, resulting in the induced expression of type I interferon and other effector genes. I suggest the authors to explore this route, or other routes, to identify the molecular mechanism(s) linking mitochondrial dysfunction and IFN-I release. mtDNA can be visualised in the cytosol by super resolution microscopy, a technique which is familiar to the authors. Is mtDNA released? Is the cGAS-MITA/STING pathways activated? Is NRLP3 inflammasome activated?

On the same way, what about the signaling pathways activated by the release of mtRNA?

In Figures 1-3, data have been mainly reported for LPS-stimulated macrophages, whereas in Figure 4 data have been reported on non-stimulated macrophages. Is there a reason for this? Considering that after stimulation with LPS, mitochondrial mass increased in m.5019A>G macrophages (fig. 1j), did mitochondrial complex I, III and IV also increase after stimulation?

In figure 4, the abundance of nuclear-encoded structural subunits of mitochondrial complexes is reported. I would couple figure 4 with figure 1, somehow, because both of them represents complementary descriptions of mitochondrial phenotype and biology of m.5019A>G macrophages compared to WT mice macrophages.

The authors stated that a key focus of their study was to understand whether the innate immune response of m.5019A>G mice was distinct from WT mice. To investigate this they injected a sub-lethal dose of LPS intraperitoneally to analyse cytokine and chemokines expression. However, innate immune response is determined both by soluble factor, like cytokines/chemokines and complement, and by cellular mediators, like monocytes, NK cells, neutrophils. Did the authors had the chance to measure these cells, and their subsets, in m.5019A>G mice vs WT mice?

I would suggest the authors to ass a final graphical abstract, reporting all changes described in m.5019A>G macrophages.

Minor:

Glycolysis stress test could not be appropriate while Glycolytic Rate assay would provide a more accurate measure of glycolysis in live cells as it provides a more precise measurement of extracellular acidification specifically due to glycolysis by subtracting out mitochondrial sources of acidification, as well as reporting the data in standard units (pmol/min).

What about ECAR in stimulated macrophages?

Lines 202-208: figures should be reported in the corrected order.

Reviewer #2

(Remarks to the Author)

In this manuscript, Marques et al. explore innate immune responses in a mouse model that is heteroplasmic for the m.5019A>G mtDNA mutation, which impacts the mitochondrial tRNA Ala gene (mt-Ta) and prevents charging with its cognate amino acid, alanine. Macrophages from mice carrying >70% m.5019A>G exhibited interesting alterations in innate immune responses, including increased IFN β release, reduced pro-inflammatory cytokine production, lower inflammatory oxylipin generation, and elevated NO production. They also report impaired OXPHOS, increased glycolysis, TCA remodeling, and mitochondrial structural alterations, as well as increased release of mitochondrial nucleic acids, in m.5019A>G macrophages. Overall, this is an interesting study that adds to the growing literature detailing how mitochondrial dysfunction shapes innate immunity. The findings presented overlap recently reported data from other mouse models of mitochondrial dysfunction/disease but differ in other aspects, revealing that the impacts of different mitochondrial mutations/stressors on innate immune function are complex and likely to be unique across mouse models and classes of mitochondrial diseases. The methods, data, and analysis appear solid and the findings are logical; however, there are some gaps in the mechanisms presented. Additional experiments are needed to fill these gaps and help clarify the unique immune aspects of this particular model.

Major concerns:

- 1) IFN β secretion is clearly increased in m.5019A>G BMDMs relative to WT BMDMs after 6 hrs of LPS. Ifnb1 transcripts are also increased 1h after addition of LPS (Fig. 2). However, Fig. 4 GSEA shows that the IFN-I response is down in resting m.5019A>G BMDMs. These findings are seemingly at odds, especially since there is also elevated cytosolic mtRNA and mtDNA at rest and after LPS stimulation in m.5019A>G BMDMs. The authors should more clearly examine IFN-I responses at rest and after stimulation over a kinetic time course experiment, 0-24 hours sampling throughout, to be sure the static timepoints shown do not over or underestimate IFN-I signaling. Preferably, these experiments should analyze protein levels of ISGs and IFN-I signaling factors to solidify findings from RNA-seq and proteomics, but also shed light on the underlying mechanisms. For example, is IFN β secretion increased after LPS because of a change in signal transduction, or do metabolic alterations underlie this increase in m.5019A>G BMDMs? Also, are nucleic acid sensors involved in potentiating IFN-I signaling upon LPS challenge? The authors can use cGAS/STING inhibitors or knockdown cytosolic RNA sensors to determine if there is signal integration between TLR4 and these cytosolic innate immune sensors.
- 2) Can the authors comment how they are able to measure IL-1 β secretion from BMDMs stimulated with only LPS for 6h (Olink assay, Fig. 2d)? This is not a phenotype consistent with the literature, as pro-IL-1 β cleavage and mature IL-1 β secretion in BMDMs requires a second signal and activation of an inflammasome. It would be helpful to run LPS priming + inflammasome stimulations here (ATP for NLRP3, dsDNA for AIM2) to determine how the m.5019A>G BMDMs respond. As presented, the data are not convincing thorough to conclude that IL-1 β secretion is truly impaired. And if indeed IL-1 β secretion is down in m.5019A>G BMDMs, there needs to be convincing controls run to exclude other inflammasome impairments. IL-6 assessments should also be broadened to examine multiple timepoints.
- 3) The paper suggests that enhanced IFN β responses underlie the reduced expression of IL-1 β , IL-6, and COX-dependent oxylipin production in m.5019A>G BMDMs. To solidify this claim, the authors should use IFNAR blocking Abs in m.5019A>G BMDMs to show that a reduction of IFN β signaling is sufficient to release the break on these pro-inflammatory cytokines. Likewise, the authors suggest that higher IFN β release contributes to higher NO production in m.5019A>G BMDMs. Experiments testing whether IFNAR blockade is sufficient to suppress higher NO production in m.5019A>G BMDMs should be performed.
- 4) A reduction in OGDH E3 subunit expression seems to be very important for metabolic reprogramming in m.5019A>G BMDMs. Reduced expression of OGDH E3 subunit should be confirmed by western blot. It is unclear how the m.5019A>G mutation reduces OGDH E3 subunit expression. The authors suggest that higher NO levels in m.5019A>G BMDMs inhibit OGDHC. Experiments testing whether inhibiting NO production (via an iNOS inhibitor) can alleviate OGDHC suppression and rescue TCA remodeling should be added.
- 5) The LPS sepsis data provided in Fig. 7 also leaves some open questions. IFN β is clearly up in the serum of all m.5019A>G mice after LPS. Pro-inflammatory cytokines are also trending up (although there is a mention of IL-6 and IL-1 β being up in females, but the graph in Fig. 7d doesn't have either cytokine labeled). These results are opposite from those shown in the in vitro BMDM studies. How can the authors explain these findings? Is this a timepoint issue (2hs is relatively early for cytokines to accumulate in the LPS sepsis model). Also, can more evidence be provided to highlight the hyper-production of IFN β and IFN responses in general in tissues from m.5019A>G mice? Is there an upregulation of interferon stimulated genes in the m.5019A.G mouse tissues basally or during LPS challenge? Are there changes in iNOS levels, HIF, or COX2-dependent oxylipins in vivo after LPS?

Minor concerns:

- 1) The TMRM data in Extended data Fig 1i is contradictory to the TMRM data in Fig 1i.
- 2) The representative images from WT and m.5019A>G BMDMs in Fig 3a and Fig 3c are not very different. The mitochondria in m.5019A>G BMDMs in Extended data figure 3d are even more fragmented than

WT.

3) The image analysis of mitochondrial networks in Fig 3a-d and Extended data fig 3a should be described in the methods section instead of simply relying on citations for support.

Reviewer #3

(Remarks to the Author)

This study reports the impact of a mitochondrial tRNA mutation (m.5019A>G) on macrophage function. Leading to decreased mitochondrial translation and OXPHOS defects (CI, CIII, CIV), this mutation stimulates mitochondrial fusion, reductive glutamine metabolism and glycolysis.

QUESTIONS AND REMARKS

1. Do patients with a (homoplasmic) m.5019A>G mutation display altered inflammation-linked immune responses (SIRS)? Is there anything known?
2. Line 70 states "complex I subunit", whereas line 80 states: CI, CIII and CIV. Please define the abbreviation first, and then use it.
3. Please avoid the commercial term "Mitostress test". Perhaps use "OCR" or "oxygen consumption analysis". Similarly for the term "Glycolysis stress test". I wonder how maximal OCR can be determined using the Seahorse system and a single FCCP addition? Was this strategy also used for the Oroboros analysis? It also is unclear to me for which figure(s) the Oroboros system was used.
4. The titles of the (Extended data) figures do not always properly reflect their contents.
5. In the Results section, figure data is not always explained (e.g. what is Ant A in Figure 1g and what does the obtained data mean)?
6. To avoid confusion, please use the terms "depolarized" (less negative deltaPSI) and "hyperpolarized" (more negative deltaPSI).
7. More details are needed on the flowcytometry analyses. How was the gating performed, etc.? How does the TMRM flowcytometry assay work? TMRM accumulates in the cytosol (plasma membrane potential-dependent) and in the mitochondrial matrix (deltaPSI-dependent). With flowcytometry one measures the integrated (total) signal of each cell. How can specific statements be made on deltaPSI? Perhaps JC-1 could be used here (ratiometric)?
8. What were the dilution factors of the used antibodies?
9. Please provide magnifications of the four conditions in Figure 3A to illustrate the mitochondrial morphology changes. It is unclear whether the applied image analysis macros were specifically designed for BMDMs? If not, they should be properly validated first. Related to this, the number of mitochondrial objects per cell is only 15-20 and does not change (Extended data figure 3a). This number appears very low? It should be explained more clearly if/how the changes in various parameters are compatible with the conclusion that mitochondria are more elongated.
10. Along the same lines, for the SIM data, how are "oblate ellipticity" and "sphericity" exactly defined (equation?), and why was the aspect ratio not calculated (a prime measure of mitochondrial elongation)? How does the image processing ("standard deconvolution") and applied pipelines (e.g. "surfaces filter") affect the obtained numbers (no refs are provided on this strategy in the M&M)?
11. Did the TOMM20 staining (circular objects; Fig. 3C) interfere with the image quantification?
12. Please state exactly what "n" means for each figure (biological replicates, number of cells analyzed?). Add the total number of cells analyzed when missing.
13. Please state exactly whether SEM or SD was used for each graph (e.g. Fig. 6c).
14. Please add the protein names to the proteomics data in Figure 4C.
15. The used ATP, ADP and AMP analyses measure total cellular content. They are less informative on the free levels of these molecules, so may confound ratio calculations.
16. In the discussion it is stated that cristae architecture is disrupted (line 388). However, no quantitative data is presented?

Version 1:

Reviewer comments:

Reviewer #1

(Remarks to the Author)

The authors have thoroughly addressed all the concerns raised by the reviewer. Their revisions have significantly improved the clarity and quality of the manuscript

Reviewer #2

(Remarks to the Author)

I commend the authors for their thorough response to all reviewers' critiques. This is an interesting and impactful study and it is ready for publication without delay.

Reviewer #3

(Remarks to the Author)

I thank the authors for properly answering most of my remarks. However, I still have some remaining suggestions/questions:

Previous remark 10: Detailed information on the SIM data analysis needs to be added to the manuscript.

Previous remark 15: My question was not addressed. I meant that the used assay measures "total" amounts (or cellular content). Within a cell, the free levels of ATP, ADP and AMP are the levels that are most relevant. However, within a cell a substantial fraction of ATP, ADP and AMP are protein-bound. This means that quantification of total amounts is less informative on the free amounts.

Previous remark 7: The provided explanation is still not clear to me. I'm perfectly aware of how TMRM (should) work(s). TMRM distributes according to the potential of the plasmamembrane (~-70 mV) and $\Delta\psi$ (~-120mV). Given the Nernst equation, this will lead to TMRM accumulation in the mitochondrial matrix. This is all fine. However, flowcytometry measures the integrated (sum) fluorescence signal for each cell (so cytosol + mitochondria). This is the reason why Mitotracker Green is useful for normalization, since the mitochondrial content might change, thereby artificially suggesting that $\Delta\psi$ is changing. However, if TMRM is lost from the mitochondria (i.e. when $\Delta\psi$ depolarizes), it will show up in the cytosol. If the TMRM staining was performed correctly (i.e. there are no quenching-related phenomena occurring), this means that the total integrated (sum) fluorescence cellular signal of TMRM cannot change. I.e. the fluorescent molecules are just moving from the mitochondria to the cytosol. Therefore, the fluorescence signal cannot change, unless the plasmamembrane potential is depolarizing, which allows the TMRM to leave the cell. I guess the above needs to be considered in the interpretation of the results?

Version 2:

Reviewer comments:

Reviewer #3

(Remarks to the Author)

Reviewer #1 (Remarks to the Author)

The manuscript by Marques et al shows that the m.5019A>G mutation is associated with mitochondrial defects and remodeling of IFN-I release by macrophages. The authors utilized several state-of-the-art technologies to address their questions and this is highly appreciated by the reviewer. The manuscript is well written, logically organized and the scientific issue is well-identified and clearly formulated. There some major and minor issues to address.

We thank the reviewer for their appreciation of our experimental approach and highly positive appraisal of our manuscript. The thorough analysis of the manuscript by the reviewer has enabled us to significantly improve the work, strengthen the central findings and provide improved mechanistic insight, as detailed below.

Major:

The authors stated that the key finding of their study was that IFN-I release is elevated in m.5019A>G macrophages, providing evidence to support the idea that mitochondrial dysfunction arising from inherited mtDNA mutations perturb innate immune system signaling. Which is the mechanistic link between mitochondrial dysfunction and innate immune signaling perturbations? Is the inflammasome involved? Several studies reported that mtDNA can be released from mitochondria to the cytosol and trigger innate immune response, as it is sensed by the cGAS-MITA/STING-inflammasome pathway, resulting in the induced expression of type I interferon and other effector genes. I suggest the authors to explore this route, or other routes, to identify the molecular mechanism(s) linking mitochondrial dysfunction and IFN-I release. mtDNA can be visualised in the cytosol by super resolution microscopy, a technique which is familiar to the authors. Is mtDNA released? Is the cGAS-MITA/STING pathways activated? Is NLRP3 inflammasome activated? On the same way, what about the signaling pathways activated by the release of mtRNA?

We thank the reviewer for this thoughtful comment, which was also raised by Reviewer #2. This is indeed a complex and multifaceted issue that is of significant interest to us.

To begin, while the hypothesis that the inflammasome may be involved is certainly intriguing, we believe it is unlikely that the NLRP3 inflammasome plays a central role in the phenotype we describe. This is primarily because the majority of our experiments were not conducted under conditions typically required for canonical inflammasome activation—that is, LPS priming followed by a secondary stimulus such as ATP or nigericin. In our study, LPS was used as a single stimulus, which is generally insufficient to robustly activate the NLRP3 inflammasome. However, in response to Reviewer #2's request, we have now measured IL-1 β release following LPS priming and ATP stimulation to directly assess inflammasome activation. To avoid redundancy, we kindly refer you to our detailed response to this specific point in our reply to Reviewer #2.

To gain a more detailed understanding of the signalling cascades underlying elevated IFN- β production in m.5019A>G macrophages, we performed a time-course analysis of the type I IFN response following LPS stimulation. In line with our RNA-seq data (**Figure 5a**), *Ifnb1* expression is significantly increased at 1 hour and peaks at 2 hours post-LPS in m.5019A>G macrophages compared to WT controls, as measured by qPCR (**Figure 5b**). At 6 hours, *Ifnb1* expression declines in both genotypes; however, in m.5019A>G macrophages, expression levels increase again by 24 hours, an effect not observed in WT cells. These findings suggest a biphasic pattern of *Ifnb1* transcriptional regulation in m.5019A>G macrophages.

Consistent with this, IFN- β protein release peaks more rapidly and reaches higher concentrations in m.5019A>G macrophages (**Figure 5c**) and remains significantly elevated at 24 hours following LPS stimulation. The early increase in *Ifnb1* transcription is accompanied by enhanced phosphorylation of IRF3 at serine 396, an essential activation signal, observed at 1- and 2-hours post-stimulation (**Figure 5d**), indicating amplified TLR4-mediated IRF3 signalling in mutant macrophages. Furthermore, expression and protein levels of multiple downstream type I IFN-stimulated genes (ISGs) are significantly elevated (**Figures**

4f, 4g, 5d; Extended data figure 6c and 6d). These include *Nos2/iNOS*, *Isg15/ISG15*, *Irf7/IRF7*, *Isg20*, and *Cxcl10*. Collectively, these new data reinforce our conclusion that type I IFN signalling is elevated in *m.5019A>G* macrophages and point to enhanced TLR4-IRF3 pathway activation as a key mechanistic driver. This new data is shown below.

Figure 5:

Figure 4:

Extended data figure 6:

Extended data figure 6:

To address your comment regarding the potential role of mitochondrial nucleic acid signalling, we would like to first note that in our initial submission we reported increased levels of both mtDNA and mtRNA in the cytosol of *m.5019A>G* macrophages compared to WT (Figure 7c). This was assessed using a digitonin-based fractionation protocol. However, we acknowledge that this approach has limitations: it is relatively crude and may overestimate cytosolic content, particularly in the context of altered cristae architecture, which may render *m.5019A>G* mitochondria more susceptible to permeabilization.

To more rigorously assess cytosolic nucleic acids, we have now performed immunofluorescent staining of intact cells for DNA in combination with mitochondrial markers and analysed the samples by confocal microscopy (Figure 7d and 7e; Extended data figure 9d). In the resting state, we observed no difference in cytosolic DNA foci between *m.5019A>G* and WT macrophages. However, following 24 hours of LPS stimulation, *m.5019A>G* macrophages exhibited a significant increase in cytosolic DNA foci relative to WT.

To evaluate whether these cytosolic DNA foci contribute to IFN- β production, we inhibited the cGAS-STING pathway using RU.521, a well-characterised small molecule inhibitor of cGAS (PMID: 28963528). Inhibition of cGAS significantly reduced *Ifnb1* expression and IFN- β secretion in *m.5019A>G* macrophages at the 24-hour timepoint but had no effect on the early phase of IFN- β production (Figure 7f; Extended data figure 9e). Interestingly, while *m.5019A>G* macrophages typically display a fused mitochondrial network at rest and after short-term LPS stimulation, we observed pronounced mitochondrial fragmentation after 24 hours of LPS treatment (Extended data figure 9c). Taken together, these findings support a biphasic model of *Ifnb1* regulation in *m.5019A>G* macrophages: an early phase driven by enhanced TLR4-IRF3 signalling, and a later phase involving mitochondrial nucleic acid release and cGAS-STING activation.

Figure 7:

Figure 7:**Extended data figure 9:****Extended data figure 9:**
The inhibition of *Ifnb1* expression by RU.521 was partial and we did not observe a significant decrease in *Isg15* or *Isg20* expression (**Extended data figure 9e**), suggesting that additional factors may contribute to the late-phase IFN- β response. To investigate a potential role for mtRNA, we inhibited mitochondrial transcription using a recently developed small molecule, IMT1 (Inhibitor of Mitochondrial Transcription; PMID: 33328633). We first characterised the effects of IMT1 on mitochondrial transcription in non-stimulated WT macrophages, performing a time- and dose-dependent analysis. This confirmed a significant reduction in mtRNA levels, which was most pronounced after 24 hours of treatment (**data not included in the revised manuscript; see below**).

For rebuttal:

Based on these findings, we pre-treated both WT and *m.5019A>G* macrophages with 10 μM IMT1 for 24 hours prior to LPS stimulation for either 6 or 24 hours (Figure 7g; Extended data figure 9f and 9g). Despite the substantial reduction in mtRNA, we observed no significant change in IFN-β release at 6 hours in *m.5019A>G* macrophages (Extended data figure 9f). This suggests that mtRNA sensing does not contribute to the early phase of enhanced *Ifnb1* expression following TLR4 activation. However, at 24 hours post-LPS, IMT1 treatment resulted in a significant reduction in both *Ifnb1* transcript levels, IFN-β secretion and reduced *Isg15* and *Isg20* expression (Figure 7g; Extended data figure 9g). These findings indicate that mitochondrial gene expression also supports sustained IFN-β production in the late phase, though the precise mechanism remains unclear. We attempted to knock down known sensors of double-stranded RNA, RIG-I and MDA5, using RNAi to assess whether direct sensing of mtRNA mediates this effect. However, we were unable to achieve efficient knockdown of these targets due to technical limitations. Consequently, we cannot yet distinguish whether the observed effect is due to direct mtRNA sensing or a secondary consequence of impaired mitochondrial gene expression, which we discuss in the revised manuscript. We hope the reviewer agrees that a detailed mechanistic investigation of this and other pathways is beyond the scope of the current study but represents an exciting direction for future work.

Figure 7:

Extended Data Figure 9:

In Figures 1-3, data have been mainly reported for LPS-stimulated macrophages, whereas in Figure 4 data have been reported on non-stimulated macrophages. Is there a reason for this? Considering that after stimulation with LPS, mitochondrial mass increased in *m.5019A>G* macrophages (fig. 1j), did mitochondrial complex I, III and IV also increase after stimulation? In figure 4, the abundance of nuclear-encoded structural subunits of mitochondrial complexes is reported. I would couple figure 4 with figure 1, somehow, because both of them represents complementary descriptions of mitochondrial phenotype and biology of *m.5019A>G* macrophages compared to WT mice macrophages.

We thank the reviewer for their comment and would like to address both points together, as they have directly informed a restructuring of the manuscript. We would first like to clarify a minor misunderstanding regarding the data presented in the original figures. The data in the original Figure 1 and Figure 4 were derived from non-stimulated WT and *m.5019A>G* macrophages, with the exception of Figure 1c, which included both non-stimulated and LPS-stimulated conditions. The original Figure 3 included both non-stimulated and LPS-stimulated conditions. In response to the reviewer's suggestion, we have now restructured **Figure 1** to more clearly present the data (**see below**), and the associated **Extended data figures 1** and **2**. The revised figures combines the original datasets along with our proteomic and transcriptomic analyses, allowing for a more integrated and coherent depiction of the basal state in *m.5019A>G* macrophages. We believe this new organisation improves the clarity of the manuscript and more effectively sets the stage for the subsequent analyses.

Figure 1: Editorial note: Cartoon schematic in panel a is Created in BioRender. Dwane, L. (2025) <https://BioRender.com/to6x4hj>

The increase in mitochondrial mass (**Figure 1j**) was observed in non-stimulated *m.5019A>G* macrophages compared to WT. Despite this increase, we detect a significant depletion in structural subunits of respiratory chain complexes I (CI), III (CIII), and IV (CIV) in *m.5019A>G* cells under basal conditions (**Figure 1g**; **Extended Data Figure 1g, 1h, and 2e**). To further support these findings, we have now included new data using a commercial OXPHOS antibody cocktail, which reveals reduced levels of CI subunit NDUF8, CIII subunit UQCRC2, and the mtDNA-encoded CIV subunit MT-CO1 in *m.5019A>G* macrophages. This reduction is evident both under non-stimulated conditions and following LPS stimulation for 6 and 24 hours (**Figure 1h**). However, we also observe a compensatory increase in other mitochondrial proteins, notably components of the mitochondrial ribosome, in the resting state (**Extended Data Figure 2c**). These results suggest that although mitochondrial mass is modestly elevated, there is a selective depletion of key respiratory chain components in *m.5019A>G* macrophages at rest and in the activated state.

Extended data figure 1:

Extended data figure 2:

The authors stated that a key focus of their study was to understand whether the innate immune response of *m.5019A>G* mice was distinct from WT mice. To investigate this they injected a sub-lethal dose of LPS intraperitoneally to analyse cytokine and chemokines expression. However, innate immune response is

determined both by soluble factor, like cytokines/chemokines and complement, and by cellular mediators, like monocytes, NK cells, neutrophils. Did the authors had the chance to measure these cells, and their subsets, in *m.5019A>G* mice vs WT mice?

We thank the reviewer for this thoughtful comment and fully agree that cellular mediators play a critical role in shaping the innate immune response. However, the complexity of our *m.5019A>G* mtDNA mutation model presents logistical challenges that have limited the extent of *in vivo* experimentation within a reasonable timeframe. Specifically, we frequently face difficulties in breeding sufficient numbers of mice with high mutational burden and matched age, which are essential for robust *in vivo* analyses. Given these constraints, our *in vivo* study was designed to prioritise the assessment of soluble mediators of the innate immune response, key drivers of systemic inflammation, while also allowing for the collection of lung and kidney tissue. To ensure consistent and time-sensitive sampling of serum and organs and behavioural analysis, we were unable to include protocols for the analysis of cellular immune populations during these initial studies.

We have now expanded our *in vivo* analyses to include tissue-specific assessments alongside circulating cytokines, as requested by Reviewer #2. Notably, we observe significantly elevated *Ifnb1* expression in both lung and kidney tissue from *m.5019A>G* mice, consistent with the increase in circulating IFN- β levels (**Figure 8e; Extended Data Figure 10b**). Moreover, proteomic analysis of these tissues reveals a significant enrichment of type I IFN ISGs (**Figure 8f; Extended Data Figures 10a, 10c and 10d**). In the kidney, this includes several key viral and host dsRNA sensors (PKR, OASL1, OAS1A, DHX58, ADAR, MDA5, SHFL, and MORC2A), as well as the anti-viral family of IFIT proteins, and IFI204, which can co-operate with cGAS to sense cytosolic DNA. We acknowledge the importance of directly analysing innate immune cell populations *in vivo* and fully agree that such an investigation would provide valuable mechanistic insight. However, we believe this is beyond the scope of the current study and represents a logical and exciting direction for future work. We are grateful to the reviewer for highlighting this important avenue of investigation.

Figure 8:

Extended data figure 10:

I would suggest the authors to add a final graphical abstract, reporting all changes described in m.5019A>G macrophages.

We thank the reviewer for this great suggestion and have now included a graphical abstract summarising the findings. This is now included in **Extended data figure 10e**.

Extended data figure 10: Editorial note: Cartoon schematic below is Created in BioRender. Dwane, L. (2025) <https://BioRender.com/d1gscu7>

Minor:

Glycolysis stress test could not be appropriate while Glycolytic Rate assay would provide a more accurate measure of glycolysis in live cells as it provides a more precise measurement of extracellular acidification specifically due to glycolysis by subtracting out mitochondrial sources of acidification, as well as reporting the data in standard units (pmol/min).

What about ECAR in stimulated macrophages?

We thank the reviewer for this suggestion and for highlighting to report the data in standard units. We have restructured the manuscript, and this data is now a component of **Figure 2** and **Extended data figure 3**. We now provide further analysis of the proton efflux rate (PER) (pmol/min/ μ g protein) calculated from the ECAR measurements obtained in the glycolysis stress test, according to the manufacturer's instructions. This analysis reflects the findings of the reported ECAR measurements, with a significant increase in PER in non-stimulated *m.5019A>G* macrophages relative to WT following injection of glucose. This has now replaced the ECAR measurements (**Figure 2c and 2d**). While not precisely ruling out some contribution of mitochondrial sources of acidification, it shows significantly enhanced PER arising from glucose catabolism compared to non-glycolytic PER. In addition, we also include the oxygen consumption rate (OCR) measurements from the assay, which highlight a substantial reduction in *m.5019A>G* macrophages basally (**Extended data figure 3c**). Simultaneously, OCR rates decrease further following glucose injection.

Figure 2:

Extended figure 3c:

As the glycolytic rate assay indirectly measures glycolysis-mediated lactate production, we have now performed $U\text{-}^{13}\text{C}$ -glucose tracing in non-stimulated and LPS-stimulated WT and *m.5019A>G* macrophages as a more precise approach. Specifically, we have measured lactate and pyruvate released into the cell culture medium (CCM) and observe significantly increased lactate/pyruvate ratios (an established proxy of reductive stress - PMID: 31932725) in *m.5019A>G* macrophages both at rest and after 6 h and 24 h LPS timepoints (**Figure 2f**). We also observe robust $m+3$ labelling of intracellular pyruvate and lactate from $U\text{-}^{13}\text{C}$ -glucose, which is increased in *m.5019A>G* macrophages relative to WT (**Figure 2i; Extended data figure 3f**). Our analysis of the TCA cycle using both $U\text{-}^{13}\text{C}$ -glutamine and $U^{13}\text{C}$ -glucose isotope-assisted tracing demonstrate reduced oxidative TCA cycle flux and increased reductive carboxylation from glutamine, which would limit mitochondrial CO_2 production and its contribution to PER/ECAR in *m.5019A>G* macrophages. Overall, our data supports a decrease in mitochondrial respiration, reduced oxidative TCA cycle activity, and

a compensatory increase in aerobic glycolysis in *m.5019A>G* macrophages, which we hope the reviewer will find satisfactory.

Figure 3:

Figure 2:

Extended data figure 3f:

Lines 202-208: figures should be reported in the corrected order.

We have now restructured the manuscripts and reported the figure panels in the correct order, thank you for this astute observation.

We thank the reviewer for their thorough evaluation of our manuscript. We believe that the additional clarifications and new data have substantially strengthened the study.

Reviewer #2 (Remarks to the Author)

In this manuscript, Marques et al. explore innate immune responses in a mouse model that is heteroplasmic for the m.5019A>G mtDNA mutation, which impacts the mitochondrial tRNA Ala gene (mt-Ta) and prevents charging with its cognate amino acid, alanine. Macrophages from mice carrying >70% m.5019A>G exhibited interesting alterations in innate immune responses, including increased IFN β release, reduced pro-inflammatory cytokine production, lower inflammatory oxylipin generation, and elevated NO production. They also report impaired OXPHOS, increased glycolysis, TCA remodeling, and mitochondrial structural alterations, as well as increased release of mitochondrial nucleic acids, in m.5019A>G macrophages. Overall, this is an interesting study that adds to the growing literature detailing how mitochondrial dysfunction shapes innate immunity. The findings presented overlap recently reported data from other mouse models of mitochondrial dysfunction/disease but differ in other aspects, revealing that the impacts of different mitochondrial mutations/stressors on innate immune function are complex and likely to be unique across mouse models and classes of mitochondrial diseases. The methods, data, and analysis appear solid and the findings are logical; however, there are some gaps in the mechanisms presented. Additional experiments are needed to fill these gaps and help clarify the unique immune aspects of this particular model.

We thank the reviewer for their positive appraisal of our scientific approach and for recognising our study as a valuable contribution to the growing field of mitochondria and innate immunity. We also greatly appreciate the reviewer's thorough and constructive analysis of the manuscript, which has enabled us to substantially improve the quality and clarity of the work, as detailed below.

Major concerns:

1) IFN β secretion is clearly increased in m.5019A>G BMDMs relative to WT BMDMs after 6 hrs of LPS. *Ifnb1* transcripts are also increased 1h after addition of LPS (Fig. 2). However, Fig. 4 GSEA shows that the IFN-I response is down in resting m.5019A>G BMDMs. These findings are seemingly at odds, especially since there is also elevated cytosolic mtRNA and mtDNA at rest and after LPS stimulation in m.5019A>G BMDMs. The authors should more clearly examine IFN-I responses at rest and after stimulation over a kinetic time course experiment, 0-24 hours sampling throughout, to be sure the static timepoints shown do not over or underestimate IFN-I signaling. Preferably, these experiments should analyze protein levels of ISGs and IFN-I signaling factors to solidify findings from RNA-seq and proteomics, but also shed light on the underlying mechanisms. For example, is IFN β secretion increased after LPS because of a change in signal transduction, or do metabolic alterations underlie this increase in m.5019A>G BMDMs? Also, are nucleic acid sensors involved in potentiating IFN-I signalling upon LPS challenge? The authors can use cGAS/STING inhibitors or knockdown cytosolic RNA sensors to determine if there is signal integration between TLR4 and these cytosolic innate immune sensors.

We thank the reviewer for this thoughtful comment, which was also raised by Reviewer #1. To gain a more detailed understanding of the signalling cascades underlying elevated IFN- β production in m.5019A>G macrophages, we performed a time-course analysis of the type I IFN response following LPS stimulation. In line with our RNA-seq data (**Figure 5a**), *Ifnb1* expression is significantly increased at 1 hour and peaks at 2 hours post-LPS in m.5019A>G macrophages compared to WT controls, as measured by qPCR (**Figure 5b**). At 6 hours, *Ifnb1* expression declines in both genotypes; however, in m.5019A>G macrophages, expression levels increase again by 24 hours, an effect not observed in WT cells. These findings suggest a biphasic pattern of *Ifnb1* transcriptional regulation in m.5019A>G macrophages.

Consistent with this, IFN- β protein release peaks more rapidly and reaches higher concentrations in m.5019A>G macrophages (**Figure 5c**) and remains significantly elevated at 24 hours following LPS stimulation. The early increase in *Ifnb1* transcription is accompanied by enhanced phosphorylation of IRF3 at serine 396, an essential activation signal, observed at 1- and 2-hours post-stimulation (**Figure 5d**), indicating amplified TLR4-mediated IRF3 signalling in mutant macrophages.

Furthermore, expression and protein levels of multiple downstream type I IFN-stimulated genes (ISGs) are significantly elevated (**Figures 4f, 4g, 5d; Extended data figure 6c and 6d**). These include *Nos2/iNOS*, *Isg15/ISG15*, *Irf7/IRF7*, *Isg20*, and *Cxcl10*. Collectively, these new data reinforce our conclusion that type I IFN signalling is elevated in *m.5019A>G* macrophages and point to enhanced TLR4-IRF3 pathway activation as a key mechanistic driver. Importantly, we did not observe any evidence of basal IFN- β release in the absence of TLR4 stimulation. The basal decrease in ISG expression observed in our RNA sequencing analysis of *m.5019A>G* macrophages is modest in magnitude and primarily driven by a small subset of genes. Notably, this reduction was not reflected in our proteomics data.

Figure 5:

Figure 4:

Extended data figure 6:

Extended data figure 6:

To address your comment regarding the potential role of mitochondrial nucleic acid signalling, we would like to first note that in our initial submission we reported increased levels of both mtDNA and mtRNA in the cytosol of *m.5019A>G* macrophages compared to WT (**Figure 7c**). This was assessed using a digitonin-based fractionation protocol. However, we acknowledge that this approach has limitations: it is relatively crude and may overestimate cytosolic content, particularly in the context of altered cristae architecture, which may render *m.5019A>G* mitochondria more susceptible to permeabilization.

To more rigorously assess cytosolic nucleic acids, we have now performed immunofluorescent staining of intact cells for DNA in combination with mitochondrial markers and analysed the samples by confocal microscopy (**Figure 7d and 7e**; **Extended data figure 9d**). In the resting state, we observed no difference in cytosolic DNA foci between *m.5019A>G* and WT macrophages. However, following 24 hours of LPS stimulation, *m.5019A>G* macrophages exhibited a significant increase in cytosolic DNA foci relative to WT.

To evaluate whether these cytosolic DNA foci contribute to IFN- β production, we inhibited the cGAS–MITA/STING pathway using RU.521, a well-characterised small molecule inhibitor of cGAS (PMID: 28963528). Inhibition of cGAS significantly reduced *Ifnb1* expression and IFN- β secretion in *m.5019A>G* macrophages at the 24-hour timepoint but had no effect on the early phase of IFN- β production (**Figure 7f**; **Extended data figure 9e**). Interestingly, while *m.5019A>G* macrophages typically display a hyperfused mitochondrial network at rest and after short-term LPS stimulation, we observed pronounced mitochondrial fragmentation after 24 hours of LPS treatment (**Extended data figure 9c**). Taken together, these findings support a biphasic model of *Ifnb1* regulation in *m.5019A>G* macrophages: an early phase driven by enhanced TLR4-IRF3 signalling, and a later phase involving mitochondrial nucleic acid release and cGAS–STING activation.

Figure 7:

Figure 7:**Extended data figure 9:****Extended data figure 9:**
The inhibition of *Ifnb1* expression by RU.521 was partial and we did not observe a significant decrease in *Isg15* or *Isg20* expression (**Extended data figure 9e**), suggesting that additional factors may contribute to the late-phase IFN- β response. To investigate a potential role for mitochondrial RNA, we inhibited mitochondrial transcription using a recently developed small molecule, IMT1 (Inhibitor of Mitochondrial Transcription; PMID: 33328633). We first characterised the effects of IMT1 on mitochondrial transcription in non-stimulated WT macrophages, performing a time- and dose-dependent analysis. This confirmed a significant reduction in mtRNA levels, which was most pronounced after 24 hours of treatment (**data not included in the revised manuscript; see below**).

For rebuttal:

Based on these findings, we pre-treated both WT and *m.5019A>G* macrophages with 10 μ M IMT1 for 24 hours prior to LPS stimulation for either 6 or 24 hours (**Figure 7g**; **Extended data figure 9f and 9g**). Despite the substantial reduction in mtRNA, we observed no significant change in IFN- β release at 6 hours in *m.5019A>G* macrophages (**Extended data figure 9f**). This suggests that mtRNA sensing does not contribute to the early phase of enhanced *Ifnb1* expression following TLR4 activation. However, at 24 hours post-LPS, IMT1 treatment resulted in a significant reduction in both *Ifnb1* transcript levels, IFN- β secretion and reduced *Isg15* and *Isg20* expression (**Figure 7g**; **Extended data figure 9g**). These findings indicate that mitochondrial gene expression also supports sustained IFN- β production in the late phase, though the precise mechanism remains unclear. We attempted to knock down known sensors of double-stranded RNA, RIG-I and MDA5, using RNAi to assess whether direct sensing of mtRNA mediates this effect. However, we were unable to achieve efficient knockdown of these targets due to technical limitations. Consequently, we cannot yet distinguish whether the observed effect is due to direct mtRNA sensing or a secondary consequence of impaired mitochondrial gene expression, which we discuss in the revised manuscript. We hope the reviewer agrees that a detailed mechanistic investigation of this and other pathways is beyond the scope of the current study but represents an exciting direction for future work.

Figure 7:

Extended Data Figure 9:

2) Can the authors comment how they are able to measure IL-1 β secretion from BMDMs stimulated with only LPS for 6h (Olink assay, Fig. 2d)? This is not a phenotype consistent with the literature, as pro-IL-1 β cleavage and mature IL-1 β secretion in BMDMs requires a second signal and activation of an inflammasome. It would be helpful to run LPS priming + inflammasome stimulations here (ATP for NLRP3, dsDNA for AIM2) to determine how the *m.5019A>G* BMDMs respond. As presented, the data are not convincing thorough to conclude that IL-1 β secretion is truly impaired. And if indeed IL-1 β secretion is down in *m.5019A>G* BMDMs,

there needs to be convincing controls run to exclude other inflammasome impairments. IL-6 assessments should also be broadened to examine multiple timepoints.

We thank the reviewer for their comment. We were indeed initially surprised to detect IL-1 β in the Olink assay in the absence of a second signal for inflammasome activation. However, upon closer examination, this observation became more understandable. Previous studies have reported IL-1 β detection from BMDMs without inflammasome activation, although typically at later timepoints (≥ 24 h) (PMID: 32019928; PMID: 32259027). We further investigated the IL-1 β levels quantified by the Olink assay and found that the detected concentrations were extremely low—below 1.8 pg/mL (**Extended Data Figure 6f**). This level is beneath the detection threshold of all commercial ELISA kits we are aware of and thus would not be reliably observed using standard assays at this timepoint. The high sensitivity of the Olink platform likely accounts for this detection.

Extended data figure 6:

This finding suggested several possible explanations. On one hand, it may reflect a very low level of NLRP3 inflammasome activity or alternative proteolytic cleavage of pro-IL-1 β occurring in the absence of a classical second signal. Alternatively, it could represent a small amount of pro-IL-1 β released extracellularly following LPS stimulation. As the Olink assay detects both pro- and cleaved forms of IL-1 β , it is difficult to determine which explanation is most likely.

Nevertheless, this observation prompted us to more systematically investigate *Il1b* transcript levels and pro-IL-1 β protein expression in WT and *m.5019A>G* macrophages following LPS stimulation (**Extended Data Figure 6g**). These data clearly show a reduction in *Il1b* mRNA and pro-IL-1 β protein in *m.5019A>G* macrophages compared to WT. In response to the reviewer's request, we also performed a time-course analysis of pro-IL-1 β , COX2 and IL-6 levels, which further supports our conclusion that these inflammatory mediators are consistently reduced in *m.5019A>G* macrophages at all examined timepoints (**Figure 5g; Extended data figure 7b**). During this time-course, we also noted the appearance of a faint, lower molecular weight band in WT macrophages after LPS stimulation, which becomes more prominent over time as pro-IL-1 β accumulates. This band likely represents a low-abundance IL-1 β cleavage product. Notably, this band is not always clearly detectable in *m.5019A>G* macrophages. Taken together, these findings help explain the ability of the Olink assay to detect IL-1 β and are consistent with a mild level of LPS-induced IL-1 β release, likely driven by basal pro-IL-1 β release or limited cleavage, primarily in WT cells.

Figure 5:**Extended data figure 7b**
To further support our findings, we employed a *Salmonella typhimurium* (STM) infection model, which activates both the NLRC4 and NLRP3 inflammasomes (PMID: 24803432). In this context, we observed reduced IL-1 β release from *m.5019A>G* macrophages, with minimal differences in bacterial colony-forming units or LDH release (**Extended Data Figure 6h**). These results suggest that the diminished IL-1 β production stems from lower pro-IL-1 β availability rather than an intrinsic defect in inflammasome activation or cell viability. In response to the reviewer's request, we have now also performed a canonical NLRP3 inflammasome assay using ATP as a second signal following LPS priming (**Extended Data Figure 6i**). Consistent with our *Salmonella* findings, IL-1 β release was significantly reduced in *m.5019A>G* macrophages compared to WT at both 3- and 6-hours post-treatment. Collectively, these results indicate that the reduced IL-1 β output in *m.5019A>G* macrophages is primarily due to lower *Il1b* transcription and reduced pro-IL-1 β levels, rather than impaired inflammasome signalling *per se*. Similar results have recently been reported in *Polg*^{D275A} mutant macrophages (PMID: 38798587). We have now incorporated this new data into the revised manuscript and hope the reviewer finds this response satisfactory.

Extended data figure 6i:
3) The paper suggests that enhanced IFN β responses underlie the reduced expression of IL-1 β , IL-6, and COX-dependent oxylipin production in *m.5019A>G* BMDMs. To solidify this claim, the authors should use IFNAR blocking Abs in *m.5019A>G* BMDMs to show that a reduction of IFN β signalling is sufficient to release the brake on these pro-inflammatory cytokines. Likewise, the authors suggest that higher IFN β release contributes to higher NO production in *m.5019A>G* BMDMs. Experiments testing whether IFNAR blockade is sufficient to suppress higher NO production in *m.5019A>G* BMDMs should be performed.

We thank the reviewer for their comment. The essential role of IFN- β and IFNAR signalling in promoting iNOS expression and nitric oxide (NO) production in LPS-stimulated macrophages is well established (PMID:

10733102; 11310846; 11602590; 11896392; 16912041; 22717332; 26146080; 30450098; 34777348; doi: <https://doi.org/10.1101/2025.03.10.642106>). This has been demonstrated using a range of pharmacological and genetic approaches, including JAK inhibition (e.g., with Ruxolitinib), and studies employing TRIF^{-/-}, STAT1^{-/-}, IFN-β^{-/-}, and IFNAR^{-/-} macrophages, as well as anti-type I IFN neutralising antibodies. Collectively, these studies highlight a critical role for autocrine type I IFN signalling in the induction of iNOS and NO following LPS stimulation. Conversely, supplementation with exogenous IFN-β alongside LPS accelerates iNOS expression and restores its induction in IFN-β^{-/-} macrophages (PMID: 16912041; 30450098). In this context, IFN-β can also substitute for IFN-γ in supporting iNOS induction. These findings reinforce the link between type I IFN signalling and iNOS.

We have confirmed this effect of exogenous IFN-β on WT macrophages, which induced iNOS and reduced both COX2 and pro-IL-1β levels (**Extended data figure 7e and 7f**). We did not observe a decrease in IL-6 secretion with IFN-β treatment (data not shown) and so is less likely to be involved in this modest reduction. Given *Nos2* is an established ISG, and both iNOS and NO are significantly increased in *m.5019A>G* macrophages, along with elevated IFN-β, It's highly likely that this observation is connected. To directly test this, we pre-treated *m.5019A>G* macrophages with either the JAK inhibitor Ruxolitinib or an IFNAR-blocking antibody for 1 hour prior to LPS stimulation for 6 or 24 hours (**Figure 5h, 5i and 5j; Extended data figure 7g**). Both treatments abolished iNOS induction and reversed the suppression of pro-IL-1β and COX2, thereby mechanistically linking type I IFN signalling to the regulation of these inflammatory mediators in *m.5019A>G* macrophages.

Extended data figure 7:

Figure 5:

Extended data figure 7:

g

4) A reduction in OGDH E3 subunit expression seems to be very important for metabolic reprogramming in *m.5019A>G* BMDMs. Reduced expression of OGDH E3 subunit should be confirmed by western blot. It is unclear how the *m.5019A>G* mutation reduces OGDH E3 subunit expression. The authors suggest that higher NO levels in *m.5019A>G* BMDMs inhibit OGDHC. Experiments testing whether inhibiting NO production (via an iNOS inhibitor) can alleviate OGDHC suppression and rescue TCA remodeling should be added.

We thank the reviewer for their comment and would like to highlight that dihydrolipoamide dehydrogenase (DLD), the E3 subunit, is a shared enzymatic component of several mitochondrial dehydrogenase complexes. These include the pyruvate dehydrogenase complex (PDHC), the 2-oxoglutarate dehydrogenase complex (OGDHC), the branched-chain α -keto acid dehydrogenase complex, the α -aminoadipate dehydrogenase complex, and the glycine cleavage system. Given this shared role, a reduction in DLD levels would be expected to impair the function of all these complexes. Functionally, reduced DLD expression would limit the entry of pyruvate into the TCA cycle via PDHC, thereby supporting a shift toward aerobic glycolysis. Simultaneously, it would constrain oxidative TCA cycle activity through inhibition of OGDHC. Thus, the observed reduction in DLD levels is consistent with the broader metabolic rewiring we report in *m.5019A>G* macrophages. We have now confirmed this by performing U- ^{13}C -glucose tracing into pyruvate, lactate and TCA cycle intermediates (also see response to Reviewer #1). This data shows there is increased flux of carbon from glucose into lactate ($m+3$) and reduced entry of pyruvate ($m+3$) into citrate ($m+2$) and downstream TCA cycle metabolites in *m.5019A>G* macrophages following 6 hours of LPS stimulation (**Figure 2h and 2i; Extended data figure 3f and 3g**). There is also a reduction in $m+4$ and $m+6$ citrate, and $m+4$ labelling across all other intermediates indicative of reduced oxidative TCA cycle activity, consistent with our U- ^{13}C -glutamine tracing.

Figure 2:

Extended data figure 3:

Extended data figure 3:

Excitingly, this data has also revealed that a significant proportion of glucose carbons enter the TCA cycle and the aspartate-argininosuccinate shunt via pyruvate carboxylation (m+3 labelling) in *m.5019A>G* macrophages following 24 hours of LPS stimulation (**Figure 3g and 3h; Extended data figure 4h; Figure 4c and 4d; Extended data figure 5b**). Furthermore, this U-¹³C-glucose tracing confirms the breakpoint of aconitase (ACO2) previously reported in LPS-stimulated macrophages (PMID: 32019928) with virtually no labelling of isocitrate or α-KG after 24 h of LPS. As such, the m+3 labelled succinate in *m.5019A>G* macrophages observed after 24 h is likely due to a reversal of complex II/SDH activity, highlighting two additional metabolic adaptations to an inherited mtDNA mutation in inflammatory macrophages. As per your request, we have also confirmed reduced levels of DLD by Western blot (**Extended data figure 4c**).

Figure 3:

Extended data figure 4:

Figure 4:

Extended data figure 5:

It was not our intention to suggest that induction of iNOS and NO is sufficient to explain metabolic remodelling in *m.5019A>G* macrophages. Indeed, this data shows that DLD levels are already reduced in *m.5019A>G* prior to LPS-mediated NO production. While it is unclear precisely how this subunit is regulated in this context, it is a key component of several mitochondrial metabolic enzymes and accompanied by marked remodelling of the mitochondrial proteome, as we have described within the manuscript. We found no changes in the gene expression level of DLD E3 subunit in our RNA sequencing (**data not included in the revised manuscript; see below**), which suggests this regulation occurs post-transcriptionally.

For rebuttal:

It is worth mentioning that post-translational modification of OGDHC by fumarate-mediated succination reduces its activity in *Ndufs4*^{-/-} cells with Complex I deficiency (PMID: 37883842). Given fumarate is elevated in *m.5019A>G* macrophages, a variety of factors, including reduced DLD, increased fumarate, and reductive stress likely contribute to reduced flux through PDHC and OGDHC. NO is also an established inhibitor of both of these complexes and OxPhos in macrophages, which we wanted to acknowledge. It is a reasonable assumption that given our observations of elevated NO in *m.5019A>G* macrophages that this will feedback to further limit PDHC and OGDHC activity following prolonged LPS stimulation based on the published literature (PMID: 32019928; PMID: 27732846). However, inhibiting iNOS alone is unlikely to reverse the observed metabolic effect of impaired mitochondrial respiration and increased aerobic glycolysis in *m.5019A>G* macrophages prior to stimulation. We have now extended our discussion section to more broadly discuss the metabolic changes observed in response to this mtDNA mutation.

5) The LPS sepsis data provided in Fig. 7 also leaves some open questions. IFN β is clearly up in the serum of all *m.5019A>G* mice after LPS. Pro-inflammatory cytokines are also trending up (although there is a

mention of IL-6 and IL-1 β being up in females, but the graph in Fig. 7d doesn't have either cytokine labelled). These results are opposite from those shown in the *in vitro* BMDM studies. How can the authors explain these findings? Is this a timepoint issue (2hs is relatively early for cytokines to accumulate in the LPS sepsis model). Also, can more evidence be provided to highlight the hyper-production of IFN β and IFN responses in general in tissues from *m.5019A>G* mice? Is there an upregulation of interferon stimulated genes in the *m.5019A.G* mouse tissues basally or during LPS challenge? Are there changes in iNOS levels, HIF, or COX2-dependent oxylipins *in vivo* after LPS?

We thank the reviewer for their comment. We would like to point out a minor misunderstanding of the reading of the *in vivo* data. In the *in vivo* LPS sepsis model, we observed no statistically significant differences in circulating IL-6 and IL-1 β levels between WT and *m.5019A>G* mice at this timepoint. As such, these cytokines were not labelled on the graph as they weren't significantly different, and we only labelled the significant findings. We have now combined the Olink *in vivo* data from male and female mice (**Figure 8a, 8b and 8c**). Interestingly, this revealed that circulating IL-17F is significantly elevated in *m.5019A>G* macrophages in our PBS controls. This is an exciting observation as IL-17F is an inflammatory cytokine and together with type I IFNs can amplify autoimmune and inflammatory responses, such as in systemic lupus erythematosus (SLE) (PMID: 22949326). As before, IFN- α 2 and TNF- α are significantly increased in *m.5019A>G* mice in our LPS sepsis model.

Figure 8: Editorial note: Panel a below is Created in BioRender. Dwane, L. (2025) <https://BioRender.com/kk5tnv3>

The 2 h timepoint does represent an acute timepoint but it is sufficient to drive a potent increase in circulating cytokines and is not an uncommon timepoint to examine, with circulating TNF- α and other cytokines often peaking at this early stage (e.g. PMID: 27667687; PMID: 27667687) and drives sickness behaviour in mice, as we have documented (**Figure 8g**). The lack of significant differences in IL-1 β and IL-6 between WT and *m.5019A>G* mice is likely to be complex but could be due to the acute timepoint as you mention. IFN- β is known to inhibit IL-1 β , which often contributes to the immunosuppressive capacity of IFN- β during infection (PMID: 23842752, PMID: 22195750; PMID: 23580529; PMID: 23580528; PMID: 16549598; PMID: 23935205). However, the *in vivo* situation is further complicated by the fact that IRF3, IRF7 and type I IFNs can also promote IL-1 β production *in vivo* following intraperitoneal LPS injection (PMID: 32373120). As such, IFN- β can play both agonistic and antagonistic roles in relation to IL-1 β . It will require a larger, focused study to assess longer timepoints and fully dissect the kinetics and dynamics of the inflammatory response in sepsis models. Unfortunately, our current animal licence does not allow us to perform a prolonged challenge, but this will certainly form part of future studies once we can obtain approval, a somewhat lengthy process, which we hope the reviewer can appreciate. Another factor to consider is the *m.5019A>G* mutation is found throughout the entire mouse, as is the case in patients with heritable mtDNA diseases, and not solely restricted to macrophage populations. Therefore, the contribution of other cells to the immune response adds an additional layer of complexity to disentangle.

In line with your request, we have now expanded our *in vivo* analyses to include tissue-specific assessments alongside circulating cytokines. Notably, we observe significantly elevated *Ifnb1* expression in both lung and kidney tissue from *m.5019A>G* mice, consistent with the increase in circulating IFN- β levels (**Figure 8e**; **Extended Data Figure 10b**). Moreover, proteomic analysis of these tissues reveals a significant enrichment of type I IFN ISGs (**Figure 8f**; **Extended Data Figures 10a, 10c and 10d**). In the kidney, this includes several key sensors of viral double-stranded RNA, such as PKR, OASL1, OAS1A, DHX58, ADAR, MDA5, SHFL, and MORC2A, as well as the anti-viral IFIT family of proteins, which can bind both host and viral RNAs to restrict replication, and IFI204, which can co-operate with cGAS to sense cytosolic DNA. This new data strengthens our central findings. There are many exciting directions we aim to pursue, which is impossible to address in the span of a single manuscript. We see open questions as a positive sign that we're just scratching the surface of our scientific inquiries into how mtDNA mutations can impact the inflammatory response *in vivo*.

Figure 8:

Extended data figure 10:

Minor concerns:

1) The TMRM data in Extended data Fig 1i is contradictory to the TMRM data in Fig 1i.

We thank the reviewer for their comment. We would just like to highlight a minor misunderstanding of the data. The data in **Figure 1k** is the change in TMRM fluorescent signal normalised to mitochondrial mass using Mitotracker Green, which enables a more accurate measurement of membrane potential ($\Delta\Psi_m$). In contrast, the TMRM in **Extended data figure 2g** is the unnormalised data. This data shows a modest reduction in $\Delta\Psi_m$ or depolarisation of *m.5019A>G* mitochondria.

2) The representative images from WT and *m.5019A>G* BMDMs in Fig 3a and Fig 3c are not very different. The mitochondria in *m.5019A>G* BMDMs in Extended data figure 3d are even more fragmented than WT.

We thank the reviewer for their comment but respectfully disagree with this assessment. As described in the methods section, we performed both semi-quantitative and quantitative analysis of mitochondrial morphology using two established image analysis macros. For assessment of branching morphology—including mitochondrial number, network structure, junction points, and junctions per network—we employed the mitoMAPR macro, which was recently validated in bone marrow-derived macrophages (BMDMs) (PMID: 39972217). This approach involves selecting a standardised region of interest (ROI; $15 \times 15 \mu\text{m}$) in the peripheral cytoplasm, where mitochondrial networks are more readily visualised. This ROI selection reduces the number of mitochondrial objects and networks per cell, enhancing the reliability and comparability of the branching analysis across cells. Our data are consistent with previous reports using the same methodology in BMDMs (PMID: 39972217). Importantly, although mitochondrial number and overall network structure were unchanged between conditions, we observed a significant increase in both total junction points and junctions per network, indicating increased mitochondrial branching and fusion. To further strengthen our analysis, we have now included mitochondrial length measurements using the same mitoMAPR macro and ROI criteria (**Extended Data Figure 8c**).

Extended data figure 8:

c

In addition, we have now performed immunofluorescent staining of the mitochondrial network and the mitochondrial fission factor, DRP1 (**Figure 6e and 6f**). DRP1 associates with the outer mitochondrial membrane (OMM) to drive mitochondrial fission. Co-localisation of DRP1 and the OMM is an indication of a more punctate mitochondrial network (PMID: 36367943). Consistent with the elongation phenotype in *m.5019A>G* macrophages, there is less co-localisation between DRP1 and the mitochondrial network, which support our claims.

Figure 6:

In addition to confocal microscopy, we have included super resolution microscopy (please also refer to our response to Reviewer #3) and transmission electron microscopy (TEM). In addition to the already included semi-quantitative and quantitative TEM analysis, we have now also quantitatively measured the mitochondrial aspect ratio (Figure 7a and 7b), a prime measurement of mitochondrial elongation, which supports the observed phenotype in resting *m.5019A>G* macrophages. However, we also acknowledge that our new data in *m.5019A>G* macrophages treated with LPS for 24 h demonstrates a transition to a fragmented phenotype following prolonged stimulation (Figure 7d; Extended data figure 9c; as shown to a previous comment). Ultimately, the key take home from our mitochondrial analysis is the defect in cristae architecture.

Figure 7:

3) The image analysis of mitochondrial networks in Fig 3a-d and Extended data fig 3a should be described in the methods section instead of simply relying on citations for support.

We apologise for this oversight and have expanded the methods section under the heading **Confocal microscopy**.

We thank the reviewer for their thorough evaluation of our manuscript. We believe that the additional clarifications and new data have substantially strengthened the study.

Reviewer #3 (Remarks to the Author):

This study reports the impact of a mitochondrial tRNA mutation (m.5019A>G) on macrophage function. Leading to decreased mitochondrial translation and OXPHOS defects (CI, CIII, CIV), this mutation stimulates mitochondrial fusion, reductive glutamine metabolism and glycolysis.

We thank the reviewer for their summary, and we have now addressed your thoughtful questions and detailed remarks below.

QUESTIONS AND REMARKS

1. Do patients with a (homoplasmic) m.5019A>G mutation display altered inflammation-linked immune responses (SIRS)? Is there anything known?

We thank the reviewer for their inquiry. We would like to clarify that the mouse model employed in our study harbours a heteroplasmic mtDNA mutation (*m.5019A>G*) in the mitochondrial tRNA for alanine (mt-tRNA^{Ala}) and not a homoplasmic mtDNA mutation. This heteroplasmic model was recently described and validated (PMID: 36827974) and provides a tractable preclinical system to investigate the consequences of pathogenic heteroplasmic mt-tRNA mutations and mitochondrial dysfunction. This is of particular clinical relevance, as mutations in mt-tRNA genes are a well-established cause of human mitochondrial diseases. Specifically, mutations in mt-tRNA^{Ala} have been identified in patients with mitochondrial disorders (PMID: 25652200).

There is a growing body of reports that patients with mitochondrial disease suffer from severe and recurrent infections, systemic inflammatory response syndrome (SIRS), and mortality arising from sepsis and pneumonia (PMID: 3605636). Notably, a clinical study conducted at Massachusetts General Hospital over an eight-year period found that 42% of patients experienced severe or recurrent infections, 13% experienced at least one episode of SIRS, and 38% presented with clinical evidence of atopy and/or autoimmune disease (PMID: 25017538). All patients had confirmed oxidative phosphorylation (OxPhos) defects and included a range of mtDNA mutations, such as the heteroplasmic m.3243A>G mutation in MT-tRNA^{Leu}, as well as mutations in MT-tRNA^{Lys}, MT-tRNA^{Phe}, MT-ATP6, MT-ND1, MT-ND2, MT-ND6 and large scale mtDNA deletions.

Our study adds to this expanding body of evidence linking mtDNA defects to immune dysregulation. We hope that our findings will help draw further attention to this under-recognised aspect of mitochondrial disease and stimulate additional research into its clinical significance.

2. Line 70 states "complex I subunit", whereas line 80 states: CI, CIII and CIV. Please define the abbreviation first, and then use it.

We apologise for this oversight; we have now substantially restructured the manuscript and ensured the abbreviation is defined first.

3. Please avoid the commercial term "Mitostress test". Perhaps use "OCR" or "oxygen consumption analysis". Similarly for the term "Glycolysis stress test". I wonder how maximal OCR can be determined using the Seahorse system and a single FCCP addition? Was this strategy also used for the Oroboros analysis? It also is unclear to me for which figure(s) the Oroboros system was used.

We thank the reviewer for this suggestion and have now removed "Mitostress test" and "Glycolysis stress test" from the relevant figures. We have replaced this with "Oxygen consumption analysis" and "Proton efflux rate analysis", respectively. The XFe24 Seahorse analyser only allows for the injection of one concentration of FCCP when performing the assay, the optimal concentration was previously determined. However, in acknowledging this is not an ideal approach in isolation, we decided to perform additional respirometry analysis with the Oroboros system, which was used to generate the data presented in **Extended data figure 1d**. We also apologise for the lack of clarity here. Data that was generated from the XFe24 Seahorse analyser

and from the Oroboros system has now been indicated in the appropriate figure legends to make this clearer. In the Oroboros experimental set-up, FCCP was injected incrementally to increase the concentration until the maximum respiratory measurement was achieved. We have now included a line on this in the methods section under the heading **Respirometry analysis (Oroboros system)**.

4. The titles of the (Extended data) figures do not always properly reflect their contents.

We apologise for this oversight and have now amended the extended data figure titles to reflect their content.

5. In the Results section, figure data is not always explained (e.g. what is Ant A in Figure 1g and what does the obtained data mean)?

We apologise for the lack of clarity. Ant A is an abbreviation for antimycin A, an inhibitor for Complex III. This serves as a positive control for the assay by preventing Coenzyme Q (CoQ) oxidation. The method is published in the following papers (PMID: 31811922; PMID: 39972217). The Ant A abbreviation is found in the figure legend (**Figure 1f**), and we have now also added a line about the treatment and the abbreviation to the methods section under the heading **Coenzyme Q (CoQ) extraction and analysis**.

6. To avoid confusion, please use the terms "depolarized" (less negative deltaPSI) and "hyperpolarized" (more negative deltaPSI).

We apologise for any confusion caused and have now corrected this in the text.

7. More details are needed on the flowcytometry analyses. How was the gating performed, etc.? How does the TMRM flowcytometry assay work? TMRM accumulates in the cytosol (plasma membrane potential-dependent) and in the mitochondrial matrix (deltaPSI-dependent). With flowcytometry one measures the integrated (total) signal of each cell. How can specific statements be made on deltaPSI? Perhaps JC-1 could be used here (ratiometric)?

We thank the reviewer for their comment and apologise for the lack of clarity. Tetramethylrhodamine methyl ester (TMRM) is a cell-permeant, positively charged (cationic), lipophilic fluorescent dye that selectively accumulates in mitochondria due to the organelle's negative membrane potential ($\Delta\Psi_m$). In polarised mitochondria, the matrix is negatively charged relative to the cytosol, which drives TMRM uptake and concentration inside mitochondria according to the Nernst equation (i.e. the dye distributes across membranes in proportion to the voltage). As a result, cells with intact, high $\Delta\Psi_m$ sequester TMRM in their mitochondria and exhibit strong red-orange fluorescence signal localised to mitochondria and is a well-established approach in the field (PMID: 21486251). We have previously confirmed the mitochondrial origin of TMRM fluorescence using confocal microscopy (PMID: 39972217). We have now included the gating strategy for these experiments (**Extended data figure 2f**). A key feature of the TMRM assay is that mitochondrial depolarisation causes a loss of TMRM fluorescence. We have now repeated the flow cytometry assay in the presence of the uncoupler BAM15 as a positive control to confirm (**data not included in the revised manuscript; see below**). This shows that BAM15 significantly decreases TMRM fluorescence in both WT and *m.5019A>G* macrophages using flow cytometry. TMRM fluorescence is higher in *m.5019A>G* macrophages, as we have previously observed (**Extended data figure 2g**). However, this control experiment did not include Mitotracker Green to normalise for mitochondrial mass, which is increased in *m.5019A>G* macrophages (**Figure 1j**). TMRM has previously been shown to be superior to JC-1 for this type of analysis. JC-1 is better for 'yes' or 'no' assays, such as during apoptosis, which is not relevant in our case (PMID: 21486251).

Extended data figure 2:

For rebuttal:

8. What were the dilution factors of the used antibodies?

We apologise for the lack of clarity here. We have now noted the dilutions for all antibodies used in **Table S3**.

Antibody	Dilution	Source	Identifier
Pro-IL-1 β	1/1000	Cell Signalling Technology	12507; RRID: AB_2721117
COX2	1/1000	Cell Signalling Technology	12282S; RRID: AB_2571729
iNOS	1/1000	Cell Signalling Technology	13120S; RRID: AB_2687529
ISG15	1/1000	Cell Signalling Technology	89771S
IRF7	1/1000	Cell Signalling Technology	72073S; RRID: AB_3073735
IRF3	1/1000	Cell Signalling Technology	4302S
Phospho-IRF3	1/1000	Cell Signalling Technology	4947S; RRID: AB_823547
Total OXPHOS	1/250	Abcam	110413; RRID: AB_2629281
DLD	1/1000	Abcam	ab133551
β -actin	1/1000	Cell Signalling Technology	4970; RRID: AB_2223172
Cytochrome c	1/1000	BD Biosciences	556432; RRID: AB_396416
DRP1	1/1000	BD Biosciences	611113; RRID: AB_398424
TOM20	1/1000	Proteintech	11802-1-AP; RRID: AB_2207530

TOM20	1/500	Abcam	ab232589; RRID: AB_3065091
ATP synthase	1/500	Merck	MAB3494; RRID: AB_177597
DNA	1/1000	Merck	CBL186; RRID: AB_11213573
Vinculin	1/1000	Cell Signalling Technology	13901; RRID: AB_2728768
F4/80	1/100	ThermoFisher scientific	12-4801-80; 12-4801-80; RRID: AB_465922
Anti-rabbit IgG, HRP-linked	1/2000	Cell Signalling Technology	7074; RRID: AB_2099233
Anti-mouse IgG, HRP-linked	1/2000	Cell Signalling Technology	7076; RRID: AB_330924
Alexa Fluor 488, Anti-mouse IgG1	1/1000	ThermoFisher scientific	A21121; RRID: AB_2535764
Alexa Fluor 568, Anti-rabbit IgG	1/1000	ThermoFisher scientific	A11036; RRID: AB_10563566

9. Please provide magnifications of the four conditions in Figure 3A to illustrate the mitochondrial morphology changes. It is unclear whether the applied image analysis macros were specifically designed for BMDMs? If not, they should be properly validated first. Related to this, the number of mitochondrial objects per cell is only 15-20 and does not change (Extended data figure 3a). This number appears very low? It should be explained more clearly if/how the changes in various parameters are compatible with the conclusion that mitochondria are more elongated.

We thank the reviewer for their comment. We performed both semi-quantitative and quantitative analysis of mitochondrial morphology using two established image analysis macros. For assessment of branching morphology—including mitochondrial number, network structure, junction points, and junctions per network—we employed the mitoMAPR macro, which was recently validated in bone marrow-derived macrophages (BMDMs) (PMID: 39972217). This approach involves selecting a standardised region of interest (ROI; 15 × 15 µm) in the peripheral cytoplasm, where mitochondrial networks are more readily visualised. This ROI selection reduces the number of mitochondrial objects and networks per cell, enhancing the reliability and comparability of the branching analysis across cells. Our data are consistent with previous reports using the same methodology in BMDMs (PMID: 39972217). Importantly, although mitochondrial number and overall network structure were unchanged between conditions, we observed a significant increase in both total junction points and junctions per network, indicating increased mitochondrial branching and hyperfusion. To further strengthen our analysis, we have now included mitochondrial length measurements using the same mitoMAPR macro and ROI criteria (**Extended Data Figure 8c**). This data shows a significant increase in mitochondrial length in NS *m.5019A>G* macrophages, in agreement with the macro that analyses the whole cell mitochondrial network.

Extended data figure 8:

In addition, we have now performed immunofluorescent staining of the mitochondrial network and the mitochondrial fission factor, DRP1 (Figure 6e and 6f). DRP1 associates with the outer mitochondrial membrane (OMM) to drive mitochondrial fission. Co-localisation of DRP1 and the OMM is a sign of a more punctate mitochondrial network (PMID: 36367943). Consistent with the elongation phenotype in *m.5019A>G* macrophages, there is less co-localisation between DRP1 and the mitochondrial network further supporting our claims.

Figure 6:

In addition to confocal microscopy, we have performed quantitative super resolution microscopy, which is explained in more depth below, and transmission electron microscopy (TEM). In addition to the already included semi-quantitative and quantitative SEM analysis, we have now also quantitatively measured the mitochondrial aspect ratio as suggested (Figure 7a and 7b), which all support the observed phenotype in resting *m.5019A>G* macrophages. However, we also acknowledge that our new data in *m.5019A>G* macrophages treated with LPS for 24 h clearly demonstrates a transition to a fragmented phenotype following prolonged stimulation (Figure 7d; Extended data figure 9c).

Figure 7:

Extended data figure 9:

10. Along the same lines, for the SIM data, how are "oblate ellipticity" and "sphericity" exactly defined (equation?), and why was the aspect ratio not calculated (a prime measure of mitochondrial elongation)? How does the image processing ("standard deconvolution") and applied pipelines (e.g. "surfaces filter") affect the obtained numbers (no refs are provided on this strategy in the M&M)?

We thank the reviewer for their comment and apologise for the lack of clarity. Aspect ratio is a prime measure of mitochondrial elongation because most studies are quantifying 2D, rather than 3D, images. In reality, cells and mitochondria are three dimensional, so a punctate/fragmented mitochondrion would approach a sphere. We found that the stimulated WT mitochondria, together with both the NS and LPS mutant mitochondria, have less spherical shapes than WT NS, which is another way of saying they are less punctate/fragmented, i.e., more elongated and/or branched. These equations are explained below:

In mathematics, an ellipsoid is a type of quadratic that is a higher dimension analogue of an Ellipse. The equation of a standard ellipsoid in an x-y-z Cartesian coordinate system is:

$$\frac{x^2}{a^2} + \frac{y^2}{b^2} + \frac{z^2}{c^2} = 1$$

where a, b and c (the length of the three semi-axes) are fixed positive real numbers determining the shape of the Ellipsoid.

Ellipsoid

$$e_{oblate} = \frac{2b^2}{b^2 + c^2} \times \left(1 - \frac{2a^2}{b^2 + c^2}\right)$$

Greater oblate ellipticity indicates that the object is more flattened.

e_{oblate} = oblate Ellipsoid

Surface – Sphericity

Sphericity is a measure of how spherical an object is. Defined by Wadell in 1932. ψ , of a particle is the ratio of the surface area of a sphere (with the volume as the given particle) to the surface area of the particle:

$$\psi = \frac{\pi^{\frac{1}{3}}(6V_p)^{\frac{2}{3}}}{A_p}$$

1 voxel = $x*y*z = 0.0313 \mu\text{m} * 0.0313 \mu\text{m} * 0.0909 \mu\text{m} = 0.0000891 \mu\text{m}^3$

Surfaces filter = 10 voxels

Therefore, any objects below $0.000891 \mu\text{m}^3$ were excluded from analysis.

Mitochondria typically have diameters between 1 and $0.2 \mu\text{m}$.

Small mitochondria are unlikely to be smaller than a sphere with a diameter of $0.2 \mu\text{m}$, equal to a volume of $4.19 \times 10^{-3} \mu\text{m}^3 = 0.00419 \mu\text{m}^3$.

Therefore, objects excluded by this filter were approximately 5 times smaller than a very small mitochondrion. This means that any objects removed were likely to be background fluorescence and should not be included in the values for measuring mitochondrial shape.

The "standard deconvolution" function in Zen black is a default selection to convert the raw image, which contains the moiré-like diffraction patterns, into a super-resolution image. Analysing the raw images would involve analysing artifactual diffraction patterns, so it would not be informative for assessment of mitochondrial morphology. We have now included this additional explanation of the analysis in methods section under the heading **Super-resolution microscopy**.

However, to address this point we also performed aspect ratio calculations on our TEM data, which again supports an increase in mitochondrial elongation in *m.5019A>G* macrophages (**Figure 7b; as shown above**).

11. Did the TOMM20 staining (circular objects; Fig. 3C) interfere with the image quantification?

We thank the reviewer for their comment. We presume the reviewer is referring to the ATP synthase staining as opposed to the TOM20 staining. The TOM20 and ATP synthase staining involved different fluorescent secondary antibodies with different excitation maxima. These secondary antibodies were excited with different lasers and collected by different cameras, which prevented any bleed-through from one channel to another. Therefore, the segmentation and shape descriptions for the TOM20 and ATP synthase objects were independent of one another and would not interfere with the quantification.

12. Please state exactly what "n" means for each figure (biological replicates, number of cells analyzed?). Add the total number of cells analyzed when missing.

We thank the reviewer for highlighting this oversight and have now updated the relevant figures.

13. Please state exactly whether SEM or SD was used for each graph (e.g. Fig. 6c).

We thank the reviewer for their comment. Figure 6c in the original manuscript was a heatmap and so SEM and SD are not applicable. However, we have inspected each figure and ensured the relevant annotation is present in the figure legends. We have now updated the figures where required.

14. Please add the protein names to the proteomics data in Figure 4C.

We thank the reviewer for their comment. Due the text size constraints on figures required by Nature publishing group we had to remove the protein names from the proteomics data of Figure 4C, which is now part of **Figure 1**, due to issues of legibility. However, we will include a source data file with the paper for each figure. The source file for **Figure 1** will contain both the protein names and the data.

15. The used ATP, ADP and AMP analyses measure total cellular content. They are less informative on the free levels of these molecules, so may confound ratio calculations.

We presume the reviewer is referring to extracellular concentrations of ATP, ADP, and AMP in their use of the term "free levels." In our study, all measurements of ATP, ADP, and AMP were derived from total cellular extracts. Each metabolite was empirically quantified using mass spectrometry calibrated with external standard curves, as detailed in the Methods section. Under normal conditions, only minimal amounts of these nucleotides are released into the extracellular medium, unless there is substantial cell damage or death (PMID: 16784779), which was not observed in our experimental conditions. Therefore, extracellular leakage would not confound our measurements. This analytical approach allowed us to accurately assess intracellular bioenergetic ratios between WT and *m.5019A>G* macrophages in the basal state.

16. In the discussion it is stated that cristae architecture is disrupted (line 388). However, no quantitative data is presented?

We thank the reviewer for their comment. We have now performed an analysis of mitochondrial cristae architecture (**Figure 7b**). This is shown above to one of your previous comments.

We thank the reviewer for their thorough evaluation of our manuscript. We believe that the additional clarifications and new data have substantially strengthened the study.

Reviewer #3 (Remarks to the Author)

We thank the reviewer for acknowledging our efforts to revise the manuscript and hope they agree that the revised version reflects a significant improvement in quality and clarity. We have carefully considered the remaining points and respond below:

I thank the authors for properly answering most of my remarks. However, I still have some remaining suggestions/questions:

Previous remark 10: Detailed information on the SIM data analysis needs to be added to the manuscript.

We have reviewed the methods section **Super-resolution microscopy (see below)**, and the details of the SIM data analysis have been added (**highlighted in yellow**). We have updated this further by including information on raw phase image reconstruction into super resolution images and the equations that were used for the 3D mitochondrial morphology analysis. We believe this has now addressed this point but if there are any additional specific technical details the reviewer is referring to, we would be happy to incorporate them.

Super-resolution microscopy

BMDMs were plated at 0.5×10^6 cells/mL (0.5 mL total volume) on coverslips in a 24-well plate and left to adhere overnight at 37°C in a de-humidified incubator (21% O₂, 5% CO₂). Following indicated treatments, BMDMs were fixed in PFA (5%), pH 7.4 for 15 min at 37°C and then washed three times with PBS. After blocking with FBS (10%) for 30 mins, macrophages were incubated with ATP Synthase (MAB3494, Merck) and TOM20 (ab232589, Abcam) primary antibodies at 1:500 in FBS (5%) overnight at 4°C. Cells were then washed with FBS (5%) three times and incubated with fluorescent secondary antibodies at 1:1000 for 1 h at room temperature with shaking, protected from light, and then washed three times with PBS. Cells were then incubated with Hoechst 33342 (1:500 in PBS) for 1 h at room temperature, washed three times with PBS, mounted onto glass slides using mounting medium (ProLong Diamond), left to dry for 12 h at room temperature, and then stored at 4 °C until imaging. Fixed cell super-resolution images for analysis of mitochondrial morphology in 3D were obtained with the Zeiss Elyra7 lattice SIM, using the Plan-Apochromat 63x/1.4 Oil DIC M27 objective with 15 phases and 0.091 μm intervals. Images were acquired with 20 ms exposure time with 405 nm (20.0%), 488 nm (4.0%), and 561 nm (6.0%) lasers. Standard deconvolution was performed in Zen Black. The "standard deconvolution" function in Zen black is a default selection to convert the raw image, which contains the moiré-like diffraction patterns, into a super-resolution image. SR-Lattice SIM reconstruction: Raw phase images were reconstructed in ZEISS ZEN Black (v 2.3) using the 3D Lattice SIM module with a *linear SIM / generalised Wiener* approach (implementation proprietary to Zeiss). User-exposed settings were phases = 15, angles = 5, drift correction = on, OTF = system-measured beads, Wiener-regularisation = 0.003, noise/background suppression = default, apodisation = 0.85. The reconstructed volumes ($0.031 \times 0.031 \times 0.091$ μm voxels, 16-bit) were exported as TIFF stacks for analysis in Imaris. Reconstruction quality was verified by Fourier-space order separation and bead-based resolution checks. For the analysis of mitochondrial morphology in 3D and quantification of ATP synthase puncta from the obtained SIM images, individual cells were cropped in ImageJ (Fiji, NIH). Using Imaris 10.1.0, objects in separate channels were segmented and rendered in 3D with the Surfaces function, as detailed in the Imaris reference manual. Segmentation setup included smoothing with Surfaces detail of 0.0986 μm, and background subtraction (Local Contrast) of 0.370 and 0.2 μm were used for TOM20 and ATP Synthase channels, respectively. Mitochondria typically have diameters between 1 and 0.2 μm. Machine Learning Segmentation was used for Hoechst channel with smoothing and Surfaces detail of 1 μm. Manual thresholding was used for TOM20 and ATP Synthase channels. For ATP Synthase channel, Split touching Objects (Region Growing) was enabled with an Intensity Based Seed Points Diameter of 0.2 μm. A surfaces filter was then applied to select objects above 10 voxels. $1 \text{ voxel} = x \times y \times z = 0.0313 \text{ } \mu\text{m} \times 0.0313 \text{ } \mu\text{m} \times 0.0909 \text{ } \mu\text{m} = 0.0000891 \text{ } \mu\text{m}^3$. Small mitochondria are unlikely to be smaller than a sphere with a diameter of 0.2 μm, equal to a volume of $4.19 \times 10^{-3} \text{ } \mu\text{m}^3 = 0.00419 \text{ } \mu\text{m}^3$. Therefore, objects excluded by this filter were approximately 5 times smaller than a very small mitochondrion. This means that any objects removed were likely to be background fluorescence and should not be included in the values for measuring mitochondrial

shape (oblate ellipticity and sphericity). An ellipsoid is a type of quadratic that is a higher dimension analogue of an ellipse. The equation of a standard ellipsoid in an x-y-z Cartesian coordinate system is:

$$\frac{x^2}{a^2} + \frac{y^2}{b^2} + \frac{z^2}{c^2} = 1$$

where a, b and c (the length of the three semi-axes) are fixed positive real numbers determining the shape of the Ellipsoid.

Ellipsoid

$$e_{oblate} = \frac{2b^2}{b^2 + c^2} \times \left(1 - \frac{2a^2}{b^2 + c^2}\right)$$

oblate Spheroid

Greater oblate ellipticity indicates that the object is more flattened.

eoblate = oblate Ellipsoid

Surface – Sphericity

Sphericity is a measure of how spherical an object is. Defined by Wadell in 1932. ψ of a particle is the ratio of the surface area of a sphere (with the volume as the given particle) to the surface area of the particle:

$$\psi = \frac{\pi^{\frac{1}{3}}(6V_p)^{\frac{2}{3}}}{A_p}$$

Object statistics from Imaris surfaces were then recorded in Microsoft Excel.

Previous remark 15: My question was not addressed. I meant that the used assay measures "total" amounts (or cellular content). Within a cell, the free levels of ATP, ADP and AMP are the levels that are most relevant. However, within a cell a substantial fraction of ATP, ADP and AMP are protein-bound. This means that quantification of total amounts is less informative on the free amounts.

In this method, ATP, ADP, and AMP were extracted using an 80% methanol and 20% water solvent system under cold conditions. This approach constitutes a metabolite extraction with no protein fractionation, meaning that small, polar metabolites are isolated directly from the whole sample without separation of protein-bound components. The high methanol content rapidly quenches metabolism and precipitates proteins, while simultaneously extracting free, soluble nucleotides into the supernatant. To quantify protein-bound nucleotides, such as ATP, this typically requires heating (PMID: 15966056). As such, this method is selective for unbound (free) nucleotide pools present in the cytosol and mitochondria at the time of quenching. Importantly, nucleotides that are tightly or transiently bound to proteins, such as those engaged in enzyme active sites or structural complexes, are not effectively extracted under these conditions and remain associated with the protein pellet. Consequently, the measured ATP, ADP, and AMP levels do not represent total cellular nucleotide content, but instead reflect the free metabolite pool, which is most relevant for interpreting dynamic cellular energy status. We hope this now clarifies this point.

Previous remark 7: The provided explanation is still not clear to me. I'm perfectly aware of how TMRM (should) work(s). TMRM distributes according to the potential of the plasma membrane (~-70 mV) and deltaPSI (~-120mV). Given the Nernst equation, this will lead to TMRM accumulation in the mitochondrial matrix. This is all fine. However, flow cytometry measures the integrated (sum) fluorescence signal for each cell (so cytosol + mitochondria). This is the reason why Mitotracker Green is useful for normalization, since the mitochondrial content might change, thereby artificially suggesting that deltaPSI is changing. However, if TMRM is lost from the mitochondria (i.e. when deltaPSI depolarizes), it will show up in the cytosol. If the TMRM staining was performed correctly (i.e. there are no quenching-related phenomena occurring), this means that the total integrated (sum) fluorescence cellular signal of TMRM cannot change. I.e. the

fluorescent molecules are just moving from the mitochondria to the cytosol. Therefore, the fluorescence signal cannot change, unless the plasma membrane potential is depolarizing, which allows the TMRM to leave the cell. I guess the above needs to be considered in the interpretation of the results?

We apologise if our previous explanation wasn't sufficiently clear. We now provide a clarified and evidence-supported rationale for the observed decrease in TMRM signal. TMRM is used in non-quench mode for our flow cytometry experiments. Under these conditions and in accordance with the Nernst equation, fluorescence is dominated by TMRM accumulated in the mitochondrial matrix, while the cytosolic fluorescence at this concentration is negligible, as the reviewer has acknowledged. We agree with the reviewer that Mitotracker Green is useful for normalisation, and we used this approach to account for mitochondrial content in our flow cytometry experiments as detailed in the paper. Upon mitochondrial depolarisation, TMRM transiently redistributes from the matrix into the cytosol. However, the resulting cytosolic concentration is no longer in electrochemical equilibrium across the plasma membrane, and TMRM is rapidly lost from the cell down its concentration gradient. This leads to a net loss of total integrated cellular fluorescence, even in the absence of plasma membrane depolarisation. We confirmed this using BAM15, a mitochondrial uncoupler that does not affect plasma membrane potential (PMID: 24634817) but still results in a marked decrease in TMRM signal by flow cytometry. This demonstrates that the decline in signal is not dependent on plasma membrane depolarisation. Therefore, while we agree that TMRM first redistributes to the cytosol transiently, the subsequent efflux from the cell explains the observed decrease in total integrated fluorescence and supports the validity of using TMRM intensity as a readout of $\Delta\Psi_m$ under these experimental conditions. As such, the assertion that: *"the fluorescent molecules are just moving from the mitochondria to the cytosol. Therefore, the fluorescence signal cannot change, unless the plasma membrane potential is depolarizing, which allows the TMRM to leave the cell"* is not correct. We hope this now fully addresses the reviewer's concerns regarding TMRM interpretation. We thank the reviewer for their time and expertise in evaluating our manuscript.

Reviewer #3 (Remarks to the Author)

My questions have been adequately addressed.

We thank the reviewer for acknowledging our efforts.